# Implicit Bias of Gradient Descent for Two-layer ReLU and Leaky ReLU Networks on Nearly-orthogonal Data

**Yiwen Kou*  Zixiang Chen*  Quanquan Gu**
Department of Computer Science
University of California, Los Angeles
Los Angeles, CA 90095
{evankou,chenzx19,qgu}@cs.ucla.edu

## Abstract

The implicit bias towards solutions with favorable properties is believed to be a key reason why neural networks trained by gradient-based optimization can generalize well. While the implicit bias of gradient flow has been widely studied for homogeneous neural networks (including ReLU and leaky ReLU networks), the implicit bias of gradient descent is currently only understood for smooth neural networks. Therefore, implicit bias in non-smooth neural networks trained by gradient descent remains an open question. In this paper, we aim to answer this question by studying the implicit bias of gradient descent for training two-layer fully connected (leaky) ReLU neural networks. We showed that when the training data are nearly-orthogonal, for leaky ReLU activation function, gradient descent will find a network with a stable rank that converges to 1, whereas for ReLU activation function, gradient descent will find a neural network with a stable rank that is upper bounded by a constant. Additionally, we show that gradient descent will find a neural network such that all the training data points have the same normalized margin asymptotically. Experiments on both synthetic and real data backup our theoretical findings.

## 1 Introduction

Neural networks have achieved remarkable success in a variety of applications, such as image and speech recognition, natural language processing, and many others. Recent studies have revealed that the effectiveness of neural networks is attributed to their implicit bias towards particular solutions which enjoy favorable properties. Understanding how this bias is affected by factors such as network architecture, optimization algorithms and data used for training, has become an active research area in the field of deep learning theory.

The literature on the implicit bias in neural networks has expanded rapidly in recent years (Vardi, 2022), with numerous studies shedding light on the implicit bias of gradient flow (GF) with a wide range of neural network architecture, including deep linear networks (Ji and Telgarsky, 2018, 2020; Gunasekar et al., 2018), homogeneous networks (Lyu and Li, 2019; Vardi et al., 2022a) and more specific cases (Chizat and Bach, 2020; Lyu et al., 2021; Frei et al., 2022b; Safran et al., 2022). The implicit bias of gradient descent (GD), on the other hand, is better understood for linear predictors (Soudry et al., 2018) and smoothed neural networks (Lyu and Li, 2019; Frei et al., 2022b). Therefore, an open question still remains:

*What is the implicit bias of leaky ReLU and ReLU networks trained by gradient descent?*

In this paper, we will answer this question by investigating gradient descent for both two-layer leaky ReLU and ReLU neural networks on specific training data, where $\{\mathbf{x}_i\}_{i=1}^n$ are nearly-orthogonal

---

*Equal contribution

37th Conference on Neural Information Processing Systems (NeurIPS 2023).

(Frei et al., 2022b), i.e., $\|\mathbf{x}_i\|_2^2 \geq Cn \max_{k \neq i} |\langle \mathbf{x}_i, \mathbf{x}_k \rangle|$ with a constant $C$. Our main results are summarized as follows:

- For two-layer leaky ReLU networks trained by GD, we demonstrate that the neuron activation pattern reaches a stable state beyond a specific time threshold and provide rigorous proof of the convergence of the stable rank of the weight matrix to 1, matching the results of Frei et al. (2022b) regarding gradient flow.

- For two-layer ReLU networks trained by GD, we proved that the stable rank of weight matrix can be upper bounded by a constant. Moreover, we present an illustrative example using completely orthogonal training data, showing that the stable rank of the weight matrix converges to a value approximately equal to 2. To the best of our knowledge, this is the first implicit bias result for two-layer ReLU networks trained by gradient descent beyond the Karush–Kuhn–Tucker (KKT) point.

- For both ReLU and leaky ReLU networks, we show that weight norm increases at the rate of $\Theta(\log(t))$ and the training loss converges to zero at the rate of $\Theta(t^{-1})$, where $t$ is the number of gradient descent iterations. This improves upon the $O(t^{-1/2})$ rate proved in Frei et al. (2022b) for the case of a two-layer *smoothed* leaky ReLU network trained by gradient descent and aligns with the results by Lyu and Li (2019) for smooth homogeneous networks. Additionally, we prove that gradient descent will find a neural network such that all the training data points have the same normalized margin asymptotically.

## 2 Related Work

**Implicit bias in neural networks.** Recent years have witnessed significant progress on implicit bias in neural networks trained by gradient flow (GF). Lyu and Li (2019) and Ji and Telgarsky (2020) demonstrated that homogeneous neural networks trained with exponentially-tailed classification losses converge in direction to the KKT point of a maximum-margin problem. Lyu et al. (2021) studied the implicit bias in two-layer leaky ReLU networks trained on linearly separable and symmetric data, showing that GF converges to a linear classifier maximizing the $\ell_2$ margin. Frei et al. (2022b) showed that two-layer leaky ReLU networks trained by GF on nearly-orthogonal data produce a $\ell_2$-max-margin solution with a linear decision boundary and rank at most two. Other works studying the implicit bias of classification using GF in nonlinear two-layer networks include Chizat and Bach (2020); Phuong and Lampert (2021); Sarussi et al. (2021); Safran et al. (2022); Vardi et al. (2022a,b); Timor et al. (2023). Although implicit bias in neural networks trained by GF has been extensively studied, research on implicit bias in networks trained by gradient descent (GD) remains limited. Lyu and Li (2019) examined smoothed homogeneous neural network trained by GD with exponentially-tailed losses and proved a convergence to KKT points of a max-margin problem. Frei et al. (2022b) studied two-layer smoothed leaky ReLU trained by GD and revealed the implicit bias towards low-rank networks. Other works studying implicit bias towards rank minimization include Ji and Telgarsky (2018, 2020); Timor et al. (2023); Arora et al. (2019); Razin and Cohen (2020); Li et al. (2021). Lastly, Vardi (2022) provided a comprehensive literature survey on implicit bias.

**Benign overfitting and double descent in neural networks.** A parallel line of research aims to understand the benign overfitting phenomenon (Bartlett et al., 2020) of neural networks by considering a variety of models. For example, Allen-Zhu and Li (2020); Jelassi and Li (2022); Shen et al. (2022); Cao et al. (2022); Kou et al. (2023) studied the generalization performance of two-layer convolutional networks on patch-based data models. Several other papers studied high-dimensional mixture models (Chatterji and Long, 2021; Wang and Thrampoulidis, 2022; Cao et al., 2021; Frei et al., 2022a). Another thread of work Belkin et al. (2020); Hastie et al. (2022); Wu and Xu (2020); Mei and Montanari (2019); Liao et al. (2020) focuses on understanding the double descent phenomenon first empirically observed by Belkin et al. (2019).

## 3 Preliminaries

In this section, we introduce the notation, fully connected neural networks, the gradient descent-based training algorithm, and a data-coorrelated decomposition technique.

**Notation.** We use lower case letters, lower case bold face letters, and upper case bold face letters to denote scalars, vectors, and matrices respectively. For a vector $\mathbf{v} = (v_1, \cdots, v_d)^\top$, we denote by

$\|\mathbf{v}\|_2 := \left(\sum_{j=1}^d v_j^2\right)^{1/2}$ its $\ell_2$ norm. For a matrix $\mathbf{A} \in \mathbb{R}^{m \times n}$, we use $\|\mathbf{A}\|_F$ to denote its Frobenius norm and $\|\mathbf{A}\|_2$ its spectral norm. We use $\text{sign}(z)$ as the function that is 1 when $z > 0$ and $-1$ otherwise. For a vector $\mathbf{v} \in \mathbb{R}^d$, we use $[\mathbf{v}]_i \in \mathbb{R}$ to denote the $i$-th component of the vector. For two sequence $\{a_k\}$ and $\{b_k\}$, we denote $a_k = O(b_k)$ if $|a_k| \le C|b_k|$ for some absolute constant $C$, denote $a_k = \Omega(b_k)$ if $b_k = O(a_k)$, and denote $a_k = \Theta(b_k)$ if $a_k = O(b_k)$ and $a_k = \Omega(b_k)$. We also denote $a_k = o(b_k)$ if $\lim |a_k/b_k| = 0$.

**Two-layer fully connected neural newtork.** We consider a two-layer neural network described as follows: its first layer consists of $m$ positive neurons and $m$ negative neurons; its second layer parameters are fixed as $+1/m$ and $-1/m$ respectively for positive and negative neurons. Then the network can be written as $f(\mathbf{W}, \mathbf{x}) = F_{+1}(\mathbf{W}_{+1}, \mathbf{x}) - F_{-1}(\mathbf{W}_{-1}, \mathbf{x})$, where the partial network function of positive and negative neurons, i.e., $F_{+1}(\mathbf{W}_{+1}, \mathbf{x})$, $F_{-1}(\mathbf{W}_{-1}, \mathbf{x})$, are defined as:

$$F_j(\mathbf{W}_j, \mathbf{x}) = \frac{1}{m}\sum_{r=1}^m \sigma(\langle \mathbf{w}_{j,r}, \mathbf{x}\rangle) \tag{3.1}$$

for $j \in \{\pm 1\}$. Here, $\sigma(z)$ represents the activation function. For ReLU, $\sigma(z) = \max\{0, z\}$, and for leaky ReLU, $\sigma(z) = \max\{\gamma z, z\}$, where $\gamma \in (0, 1)$. $\mathbf{W}_j \in \mathbb{R}^{m \times d}$ is the collection of model weights associated with $F_j$, and $\mathbf{w}_{j,r} \in \mathbb{R}^d$ denotes the weight vector for the $r$-th neuron in $\mathbf{W}_j$. We use $\mathbf{W}$ to denote the collection of all model weights.

**Gradient Descent.** Given a training data set $\mathcal{S} = \{(\mathbf{x}_i, y_i)\}_{i=1}^n \subseteq \mathbb{R}^d \times \{\pm 1\}$, instead of considering the gradient flow (GF) that is commonly studied in prior work on the implicit bias, we use gradient descent (GD) to optimize the empirical loss on the training data

$$L_S(\mathbf{W}) = \frac{1}{n}\sum_{i=1}^n \ell(y_i \cdot f(\mathbf{W}, \mathbf{x}_i)),$$

where $\ell(z) = \log(1 + \exp(-z))$ is the logistic loss, and $S = \{(\mathbf{x}_i, y_i)\}_{i=1}^n$ is the training data set. The gradient descent update rule of each neuron in the two-layer neural network can be written as

$$\mathbf{w}_{j,r}^{(t+1)} = \mathbf{w}_{j,r}^{(t)} - \eta \cdot \nabla_{\mathbf{w}_{j,r}} L_S(\mathbf{W}^{(t)}) = \mathbf{w}_{j,r}^{(t)} - \frac{\eta}{nm}\sum_{i=1}^n \ell_i'^{(t)} \cdot \sigma'(\langle \mathbf{w}_{j,r}^{(t)}, \mathbf{x}_i\rangle) \cdot j y_i \mathbf{x}_i \tag{3.2}$$

for all $j \in \{\pm 1\}$ and $r \in [m]$, where we introduce a shorthand notation $\ell_i'^{(t)} = \ell'[y_i \cdot f(\mathbf{W}^{(t)}, \mathbf{x}_i)]$ and assume the derivative of the ReLU activation function at 0 is $\sigma'(0) = 1$ without loss of generality. Here $\eta > 0$ is the learning rate. We initialize the gradient descent by Gaussian initialization, where all the entries of $\mathbf{W}^{(0)}$ are sampled from i.i.d. Gaussian distributions $\mathcal{N}(0, \sigma_0^2)$ with $\sigma_0^2$ being the variance.

# 4 Main Results

In this section, we present our main theoretical results. For the training data set $\mathcal{S} = \{(\mathbf{x}_i, y_i)\}_{i=1}^n \subseteq \mathbb{R}^d \times \{\pm 1\}$, let $R_{\min} = \min_i \|\mathbf{x}_i\|_2$, $R_{\max} = \max_i \|\mathbf{x}_i\|_2$, $p = \max_{i \ne k} |\langle \mathbf{x}_i, \mathbf{x}_k\rangle|$, and suppose $R = R_{\max}/R_{\min}$ is at most an absolute constant. For simplicity, we only consider the dependency on $t$ when characterizing the convergence rates of the weight matrix related quantities and the training loss, omitting the dependency on other parameters such as $m, n, \sigma_0, R_{\min}, R_{\max}$.

**Theorem 4.1** (Leaky ReLU Networks)**.** For two-layer neural network defined in (3.1) with leaky ReLU activation $\sigma(z) = \max\{\gamma z, z\}, \gamma \in (0, 1)$. Assume the training data satisfy $R_{\min}^2 \ge CR^2\gamma^{-4}np$ for some sufficiently large constant $C$. For any $\delta \in (0, 1)$, if the learning rate $\eta \le (CR_{\max}^2/nm)^{-1}$ and the initialization scale $\sigma_0 \le \gamma\big(CR_{\max}\sqrt{\log(mn/\delta)}\big)^{-1}$, then with probability at least $1 - \delta$ over the random initialization of gradient descent, the trained network satisfies:

- The $\ell_2$ norm of each neuron increases to infinity at a logarithmic rate: $\|\mathbf{w}_{j,r}^{(t)}\|_2 = \Theta(\log(t))$ for all $j \in \{\pm 1\}$ and $r \in [m]$.

- Throughout the gradient descent trajectory, the stable rank of the weights $\mathbf{W}_j^{(t)}$ for all $j \in \{\pm 1\}$ satisfies

$$\lim_{t \to \infty} \|\mathbf{W}_j^{(t)}\|_F^2 / \|\mathbf{W}_j^{(t)}\|_2^2 = 1,$$

  with a convergence rate of $O(1/\log(t))$.

- Gradient descent will find $\mathbf{W}^{(t)}$ such that all the training data points possess the same normalized margin asymptotically:

$$\lim_{t \to \infty} \left| y_i f(\mathbf{W}^{(t)}/\|\mathbf{W}^{(t)}\|_F, \mathbf{x}_i) - y_k f(\mathbf{W}^{(t)}/\|\mathbf{W}^{(t)}\|_F, \mathbf{x}_k) \right| = 0, \ \forall i, k \in [n].$$

  If we assume that $\mathbf{W}^{(t)}$ converges in direction, i.e., the limit of $\mathbf{W}^{(t)}/\|\mathbf{W}^{(t)}\|_F$ exists, denoted by $\bar{\mathbf{W}}$, then there exists a scaling factor $\alpha > 0$ such that $\alpha \bar{\mathbf{W}}$ satisfies the Karush-Kuhn-Tucker (KKT) conditions for the following max-margin problem:

$$\min_{\mathbf{W}} \frac{1}{2}\|\mathbf{W}\|_F^2, \qquad \text{s.t.} \qquad y_i f(\mathbf{W}, \mathbf{x}_i) \geq 1, \ \forall i \in [n]. \tag{4.1}$$

- The empirical loss converges to zero at the following rate: $L_S(\mathbf{W}^{(t)}) = \Theta(t^{-1})$.

**Remark 4.2.** In Theorem 4.1, we show that when using the leaky ReLU activation function on nearly orthogonal training data, gradient descent asymptotically finds a network with a stable rank of $\mathbf{W}_j$ equal to 1. Additionally, we demonstrate that gradient descent will find a network by which all the training data points share the same normalized margin asymptotically. Moreover, if we assume the weight matrix converges in direction, then its limit will satisfy the KKT conditions of the max-margin problem (4.1). Furthermore, we analyze the rate of weight norm increase and the convergence rate of the stable rank for gradient descent, both of which exhibit a logarithmic dependency in $t$.

**Theorem 4.3** (ReLU Networks). For two-layer neural network defined in (3.1) with ReLU activation $\sigma(z) = \max\{0, z\}$. Assume the training data satisfy $R_{\min}^2 \geq CR^2 np$ for some sufficiently large constant $C$. For any $\delta \in (0, 1)$, if the neural network width $m \geq C\log(n/\delta)$, learning rate $\eta \leq (CR_{\max}^2/nm)^{-1}$ and initialization scale $\sigma_0 \leq \left(CR_{\max}\sqrt{\log(mn/\delta)}\right)^{-1}$, then with probability at least $1 - \delta$ over the random initialization of gradient descent, the trained network satisfies:

- The Frobenious norm and the spectral norm of weight matrix increase to infinity at a logarithmic rate: $\|\mathbf{W}_j^{(t)}\|_F = \Theta(\log(t))$ and $\|\mathbf{W}_j^{(t)}\|_2 = \Theta(\log(t))$ for all $j \in \{\pm 1\}$.

- Throughout the gradient descent trajectory, the stable rank of the weights $\mathbf{W}_j^{(t)}$ for all $j \in \{\pm 1\}$ satisfies,

$$\limsup_{t \to \infty} \|\mathbf{W}_j^{(t)}\|_F^2 / \|\mathbf{W}_j^{(t)}\|_2^2 \leq c,$$

  where $c$ is an absolute constant.

- Gradient descent will find a $\mathbf{W}^{(t)}$ such that all the training data points possess the same normalized margin asymptotically:

$$\lim_{t \to \infty} \left| y_i f(\mathbf{W}^{(t)}/\|\mathbf{W}^{(t)}\|_F, \mathbf{x}_i) - y_k f(\mathbf{W}^{(t)}/\|\mathbf{W}^{(t)}\|_F, \mathbf{x}_k) \right| = 0, \ \forall i, k \in [n].$$

- The empirical loss converges to zero at the following rate: $L_S(\mathbf{W}^{(t)}) = \Theta(t^{-1})$.

**Remark 4.4.** For ReLU networks, we provide an example in the appendix concerning fully orthogonal training data and prove that the activation pattern during training depends solely on the initial activation state. Specifically, when training a two-layer ReLU network with gradient descent using such data, the stable rank of the network's weight matrix $\mathbf{W}_j$ converges to approximately 2. It is worth noting that this stable rank value is higher than the stable rank achieved by leaky ReLU networks, which is 1.

**Comparison with previous work.** One notable related work is Lyu et al. (2021), which also investigates the implicit bias of two-layer leaky ReLU networks. The main distinction between our work and Lyu et al. (2021) is the optimization method employed. We utilize gradient descent, whereas they utilize gradient flow. Additionally, our assumption is that the training data is nearly-orthogonal,

while they assume the training data is symmetric. Our findings are more closely related to the work by Frei et al. (2022b), which investigates both gradient flow and gradient decent. In both our study and Frei et al. (2022b), we examine two-layer neural networks with leaky ReLU activations. However, they focus on networks trained via gradient flow, while we investigate networks trained using gradient descent. For the gradient descent approach, Frei et al. (2022b) provide a constant stable rank upper bound for smoothed leaky ReLU. In contrast, we prove that the stable rank of leaky ReLU networks converges to 1, aligning with the implicit bias of gradient flow proved in Frei et al. (2022b). Furthermore, they presented an $O(t^{-1/2})$ convergence rate for the empirical loss, whereas our convergence rate is $\Theta(t^{-1})$. Another related work is Lyu and Li (2019), which studied smooth homogeneous networks trained by gradient descent. Our results on the rate of weight norm increase and the convergence rate of training loss match those in Lyu and Li (2019), despite the fact that we study non-smooth homogeneous networks. It is worth noting that Lyu and Li (2019); Lyu et al. (2021); Frei et al. (2022b) demonstrated that neural networks trained by gradient flow converge to a Karush-Kuhn-Tucker (KKT) point of the max-margin problem. We do not have such a result unless we assume the directional convergence of the weight matrix.

# 5 Overview of Proof Techniques

In this section, we discuss the key techniques we invent in our proofs to analyze the implicit bias of ReLU and leaky ReLU networks.

## 5.1 Refined Analysis of Decomposition Coefficient

*Signal-noise decomposition*, a technique initially introduced by Cao et al. (2022), is used to analyze the learning dynamics of two-layer convolutional networks. This method decomposes the convolutional filters into a linear combination of initial filters, signal vectors, and noise vectors, converting the neural network learning into a dynamical system of coefficients derived from the decomposition. In this work, we extend the signal-noise decomposition to *data-correlated decomposition* to facilitate the analysis of the training dynamic for two-layer fully connected neural networks.

**Definition 5.1** (Data-correlated Decomposition). Let $\mathbf{w}_{j,r}^{(t)}$, $j \in \{\pm 1\}$, $r \in [m]$ be the weights of first-layer neurons at the $t$-th iteration of gradient descent. There exist unique coefficients $\rho_{j,r,i}^{(t)}$ such that

$$\mathbf{w}_{j,r}^{(t)} = \mathbf{w}_{j,r}^{(0)} + \sum_{i=1}^{n} \rho_{j,r,i}^{(t)} \cdot \|\mathbf{x}_i\|_2^{-2} \cdot \mathbf{x}_i. \tag{5.1}$$

By defining $\overline{\rho}_{j,r,i}^{(t)} := \rho_{j,r,i}^{(t)} \mathbb{1}(\rho_{j,r,i}^{(t)} \geq 0)$, $\underline{\rho}_{j,r,i}^{(t)} := \rho_{j,r,i}^{(t)} \mathbb{1}(\rho_{j,r,i}^{(t)} \leq 0)$, (5.1) can be further written as

$$\mathbf{w}_{j,r}^{(t)} = \mathbf{w}_{j,r}^{(0)} + \sum_{i=1}^{n} \overline{\rho}_{j,r,i}^{(t)} \cdot \|\mathbf{x}_i\|_2^{-2} \cdot \mathbf{x}_i + \sum_{i=1}^{n} \underline{\rho}_{j,r,i}^{(t)} \cdot \|\mathbf{x}_i\|_2^{-2} \cdot \mathbf{x}_i. \tag{5.2}$$

As an extension of the signal-noise decomposition first proposed in Cao et al. (2022) for analyzing two-layer convolutional networks, *data-correlated decomposition* defined in Definition 5.1 can be used to analyze two-layer fully-connected network, where the normalization factors $\|\mathbf{x}_i\|_2^{-2}$ are introduced to ensure that $\rho_{j,r,i}^{(t)} \approx \langle \mathbf{w}_{j,r}^{(t)}, \mathbf{x}_i \rangle$. This is also inspired by previous works by Lyu and Li (2019); Frei et al. (2022b), which demonstrate that $\mathbf{W}$ converges to a KKT point of the max-margin problem. This implies that $\mathbf{w}_{j,r}^{(\infty)}/\|\mathbf{w}_{j,r}^{(\infty)}\|_2$ can be expressed as a linear combination of the training data $\{\mathbf{x}_i\}_{i=1}^{n}$, with the coefficient $\lambda_i$ corresponding to $\rho_{j,r,i}^{(t)}$ in our analysis. This technique does not rely on the strictly increasing and smoothness properties of the activation function and will serve as the foundation for our analysis. Let us first investigate the update rule of the coefficient $\overline{\rho}_{j,r,i}^{(t)}, \underline{\rho}_{j,r,i}^{(t)}$:

**Lemma 5.2.** The coefficients $\overline{\rho}_{j,r,i}^{(t)}, \underline{\rho}_{j,r,i}^{(t)}$ defined in Definition 5.1 satisfy the following iterative equations:

$$\overline{\rho}_{j,r,i}^{(0)}, \underline{\rho}_{j,r,i}^{(0)} = 0, \tag{5.3}$$

$$\overline{\rho}_{j,r,i}^{(t+1)} = \overline{\rho}_{j,r,i}^{(t)} - \frac{\eta}{nm} \cdot \ell_i'^{(t)} \cdot \sigma'(\langle \mathbf{w}_{j,r}^{(t)}, \mathbf{x}_i \rangle) \cdot \|\mathbf{x}_i\|_2^2 \cdot \mathbb{1}(y_i = j), \tag{5.4}$$

$$\underline{\rho}_{j,r,i}^{(t+1)} = \underline{\rho}_{j,r,i}^{(t)} + \frac{\eta}{nm} \cdot \ell_i'^{(t)} \cdot \sigma'(\langle \mathbf{w}_{j,r}^{(t)}, \mathbf{x}_i \rangle) \cdot \|\mathbf{x}_i\|_2^2 \cdot \mathbb{1}(y_i = -j), \tag{5.5}$$

for all $r \in [m]$, $j \in \{\pm 1\}$ and $i \in [n]$.

To study implicit bias, the first main challenge is to generalize the decomposition coefficient analysis to infinite time. The signal-noise decomposition used in Cao et al. (2022); Kou et al. (2023) requires early stopping with threshold $T^*$ to facilitate their analysis. They only provided upper bounds of $4\log(T^*)$ for $\overline{\rho}_{j,r,i}^{(t)}, |\underline{\rho}_{j,r,i}^{(t)}|$ (See Proposition 5.3 in Cao et al. (2022), Proposition 5.2 in Kou et al. (2023)), and then carried out a two-stage analysis. To obtain upper bounds for $\overline{\rho}_{j,r,i}^{(t)}, |\underline{\rho}_{j,r,i}^{(t)}|$, they used an upper bound for $|\ell_i'^{(t)}|$ and directly plugged it into (5.4) and (5.5) to demonstrate that $\overline{\rho}_{j,r,i}^{(t)}$ and $|\underline{\rho}_{j,r,i}^{(t)}|$ would not exceed $4\log(T^*)$, which is a fixed value related to the early stopping threshold. Therefore, dealing with infinite time requires new techniques. To overcome this difficulty, we propose a *refined analysis of decomposition coefficients* which generalizes Cao et al. (2022)'s technique. We first give the following key lemma.

**Lemma 5.3.** For non-negative real number sequence $\{x_t\}_{t=0}^{\infty}$ satisfying

$$C_1 \exp(-x_t) \le x_{t+1} - x_t \le C_2 \exp(-x_t), \tag{5.6}$$

it holds that

$$\log(\exp(-x_0) + C_1 \cdot t) \le x_t \le \log(\exp(-x_0) + C_2 \exp(C_2) \cdot t). \tag{5.7}$$

We can establish the relationship between (5.4), (5.5) and inequality (5.6) if we are able to express $|\ell_i'^{(t)}|$ using coefficients $\overline{\rho}_{j,r,i}^{(t)}$ and $|\underline{\rho}_{j,r,i}^{(t)}|$. To achieve this, we can first approximate $\ell_i^{(t)}$ using the margin $y_i f(\mathbf{W}^{(t)}, \mathbf{x}_i)$ and then approximate $F_j(\mathbf{W}_j^{(t)}, \mathbf{x}_i)$ using the coefficients $\rho_{j,r,i}^{(t)}$. The approximation is given as follows:

$$\ell_i^{(t)} = \Theta(\exp(-y_i f(\mathbf{W}^{(t)}, \mathbf{x}_i))) = \Theta\big(\exp\big(F_{-y_i}(\mathbf{W}_{-y_i}^{(t)}, \mathbf{x}_i) - F_{y_i}(\mathbf{W}_{y_i}^{(t)}, \mathbf{x}_i)\big)\big), \tag{5.8}$$

$$\left| F_j(\mathbf{W}_j^{(t)}, \mathbf{x}_i) - \frac{1}{m} \sum_{r=1}^{m} \rho_{j,r,i}^{(t)} \right| \le \sum_{i' \ne i} \left( \frac{1}{m} \sum_{r=1}^{m} |\rho_{j,r,i'}^{(t)}| R_{\min}^{-2} p \right), \tag{5.9}$$

From (5.9), one can see that we need to decouple $\ell_i^{(t)}$ from $|\rho_{j,r,i'}^{(t)}|(i' \ne i)$. In order to accomplish this, we also prove the following lemma, which demonstrates that the ratio between $\sum_{r=1}^{m} |\rho_{j,r,i}^{(t)}|$ and $\sum_{r=1}^{m} |\rho_{j,r,i'}^{(t)}|(i' \ne i)$ will maintain a constant order throughout the training process. Here, we present the lemma for leaky ReLU networks.

**Lemma 5.4** (leaky ReLU automatic balance). For two-layer leaky ReLU network defined in (3.1), for any $t \ge 0$, we have $\sum_{r=1}^{m} |\rho_{j,r,i}^{(t)}| \ge c\gamma^2 \sum_{r=1}^{m} |\rho_{j,r,i'}^{(t)}|$ for any $j \in \{\pm 1\}$ and $i, i' \in [n]$, where $c$ is a constant.

By Lemma 5.4, we can approximate the neural network output using (5.9). This approximation expresses the output $F_j(\mathbf{W}_j^{(t)}, \mathbf{x}_i)$ as a sum of the coefficients $\rho_{j,r,i}^{(t)}$:

$$F_j(\mathbf{W}_j^{(t)}, \mathbf{x}_i) \approx \frac{1 \pm c\gamma^2 R_{\min}^{-2} pn}{m} \sum_{r=1}^{m} \rho_{j,r,i}^{(t)}. \tag{5.10}$$

By combining (5.4), (5.5), (5.8), and (5.10), we obtain the following relationship:

$$\frac{1}{m} \sum_{r=1}^{m} |\rho_{j,r,i}^{(t+1)}| - \frac{1}{m} \sum_{r=1}^{m} |\rho_{j,r,i}^{(t)}| = \Theta\Big(\frac{\eta\|\mathbf{x}_i\|_2^2}{nm}\Big) \cdot \exp\Big(-\frac{1 \pm c\gamma^2 R_{\min}^{-2} pn}{m} \sum_{r=1}^{m} |\rho_{j,r,i}^{(t)}|\Big).$$

This relationship aligns with the form of (5.6), if we set $x_t = \frac{1 \pm c\gamma^2 R_{\min}^{-2} pn}{m} \sum_{r=1}^{m} |\rho_{j,r,i}^{(t)}|$. Thus, we can directly apply Lemma 5.3 to gain insights into the logarithmic rate of increase for the average

magnitudes of the coefficients $\frac{1}{m}\sum_{r=1}^{m}|\rho_{j,r,i}^{(t)}|$, which in turn implies that $\|\mathbf{w}_{j,r}^{(t)}\|_2 = \Theta(\log t)$ and $\|\mathbf{W}^{(t)}\|_F = \Theta(\log t)$. In the case of ReLU networks, we have the following lemma that provides automatic balance:

**Lemma 5.5** (ReLU automatic balance). For two-layer ReLU network defined in (3.1), there exists a constant $c$ such that for any $t \geq 0$, we have $|\rho_{y_i,r,i}^{(t)}| \geq c|\rho_{j,r',i'}^{(t)}|$ for any $j \in \{\pm 1\}$, $r \in S_i^{(0)} := \{r \in [m] : \langle \mathbf{w}_{y_i,r}^{(0)}, \mathbf{x}_i \rangle \geq 0\}$, $r' \in [m]$ and $i, i' \in [n]$.

The automatic balance lemma guarantees that the magnitudes of coefficients related to the neurons of class $y_i$, which are activated by $\mathbf{x}_i$ during initialization, dominate those of other classes. With the help of Lemma 5.5, we can get the following approximation for the margin $y_i f(\mathbf{W}^{(t)}, \mathbf{x}_i)$:

$$F_{y_i}(\mathbf{W}_{y_i}^{(t)}, \mathbf{x}_i) - F_{-y_i}(\mathbf{W}_{-y_i}^{(t)}, \mathbf{x}_i) \approx \frac{1 \pm cR_{\min}^{-2}pn}{m} \sum_{r \in S_i^{(0)}} \rho_{y_i,r,i}^{(t)}. \tag{5.11}$$

By combining (5.4), (5.5), (5.8) and (5.11), we obtain the following relationship:

$$\sum_{r \in S_i^{(0)}} |\rho_{y_i,r,i}^{(t+1)}| - \sum_{r \in S_i^{(0)}} |\rho_{y_i,r,i}^{(t)}| = \Theta\Big(\frac{\eta\|\mathbf{x}_i\|_2^2|S_i^{(0)}|}{nm}\Big) \cdot \exp\Big(-\frac{1 \pm cR_{\min}^{-2}pn}{m} \sum_{r \in S_i^{(0)}} \rho_{y_i,r,i}^{(t)}\Big),$$

which precisely matches the form of (5.6) by setting $x_t = \frac{1 \pm cR_{\min}^{-2}pn}{m}\sum_{r \in S_i^{(0)}} \rho_{y_i,r,i}^{(t)}$. Therefore, we can directly apply Lemma 5.3 and obtain the logarithmic increasing rate of $|\rho_{y_i,r,i}^{(t)}|$ for $r \in S_i^{(0)}$. Consequently, this implies that $\|\mathbf{W}^{(t)}\|_F = \Theta(\log t)$.

## 5.2 Analysis of Activation Pattern

One notable previous work (Frei et al., 2022b) provided a constant upper bound for the stable rank of two-layer smoothed leaky ReLU networks trained by gradient descent in their Theorem 4.2. To achieve a better stable rank bound, we characterize the activation pattern of leaky ReLU network neurons after a certain threshold time $T$ in the following lemma.

**Lemma 5.6** (leaky ReLU activation pattern). Let $T = C\eta^{-1}nmR_{\max}^{-2}$. For two-layer leaky ReLU network defined in (3.1), for any $t \geq T$, it holds that $\text{sign}(\langle \mathbf{w}_{j,r}^{(t)}, \mathbf{x}_i \rangle) = jy_i$ for any $j \in \{\pm 1\}$ and $r \in [m]$.

Lemma 5.6 indicates that the activation pattern will not change after time $T$. Given Lemma 5.6, we can get $\sigma'(\langle \mathbf{w}_{j,r}^{(t)}, \mathbf{x}_i \rangle) = \gamma$ for $j \neq y_i$ and $\sigma'(\langle \mathbf{w}_{j,r}^{(t)}, \mathbf{x}_i \rangle) = 1$ for $j = y_i$. Plugging this into (5.4) and (5.5) can give the following useful lemma.

**Lemma 5.7.** Let $T$ be defined in Lemma 5.6. For $t \geq T$, it holds that

$$\overline{\rho}_{y_i,r,i}^{(t)} - \overline{\rho}_{y_i,r,i}^{(T)} = \overline{\rho}_{y_i,r',i}^{(t)} - \overline{\rho}_{y_i,r',i}^{(T)}, \underline{\rho}_{-y_i,r,i}^{(t)} - \underline{\rho}_{-y_i,r,i}^{(T)} = \underline{\rho}_{-y_i,r',i}^{(t)} - \underline{\rho}_{-y_i,r',i}^{(T)},$$
$$\overline{\rho}_{y_i,r,i}^{(t)} - \overline{\rho}_{y_i,r,i}^{(T)} = (\underline{\rho}_{-y_i,r',i}^{(t)} - \underline{\rho}_{-y_i,r',i}^{(T)})/\gamma,$$

for any $i \in [n]$ and $r, r' \in [m]$.

This lemma reveals that beyond a certain time threshold $T$, the increase in $\rho_{j,r,i}^{(t)}$ is consistent across neurons within the same positive or negative class. However, for neurons belonging to the oppose class, this increment in $\rho_{j,r,i}^{(t)}$ is scaled by a factor equivalent to the slope of the leaky ReLU function $\gamma$. From this and (5.1), we can demonstrate that $\|\mathbf{w}_{j,r}^{(t)} - \mathbf{w}_{j,r'}^{(t)}\|_2 (r \neq r')$ can be upper bounded by a constant, leading to the following inequalities:

$$\|\mathbf{W}_j^{(t)}\|_F^2 \leq m\|\mathbf{w}_{j,1}^{(t)}\|_2^2 + mC_1\|\mathbf{w}_{j,1}^{(t)}\|_2 + mC_2, \|\mathbf{W}_j^{(t)}\|_2^2 \geq m\|\mathbf{w}_{j,1}^{(t)}\|_2^2 - mC_3\|\mathbf{w}_{j,1}^{(t)}\|_2 - mC_4.$$

Considering that $\|\mathbf{w}_{j,r}^{(t)}\|_2 = \Theta(\log t)$, the stable rank of $\mathbf{W}_j^{(t)}$ naturally converges to a value of 1. For ReLU networks, we can partially characterize the activation pattern as illustrated in the following lemma.

**Lemma 5.8.** (ReLU activation pattern) For two-layer ReLU networks defined in (3.1), for any $i \in [n]$, we have $S_i^{(t)} \subseteq S_i^{(t+1)}$ for any $t \geq 0$, where $S_i^{(t)} := \{r \in [m] : \langle \mathbf{w}_{y_i,r}^{(t)}, \mathbf{x}_i \rangle \geq 0\}$.

Lemma 5.8 suggests that once the neuron of class $y_i$ is activated by $\mathbf{x}_i$, it will remain activated throughout the training process. Leveraging such an activation pattern, we can establish a lower bound for $\|\mathbf{W}_j^{(t)}\|_2$ as $\Omega(\log(t))$. Together with the trivial upper bound for $\|\mathbf{W}_j^{(t)}\|_F$ of order $O(\log(t))$, it provides a constant upper bound for the stable rank of ReLU network weight.

## 5.3 Analysis of Margin and Training Loss

Notably, Lyu and Li (2019) established in their Theorem 4.4 that any limit point of smooth homogeneous neural networks $f(\mathbf{W}, \mathbf{x})$ trained by gradient descent is along the direction of a KKT point for the max-margin problem (4.1). Additionally, Lyu and Li (2019) provided precise bounds on the training loss and weight norm for smooth homogeneous neural networks in their Theorem 4.3 as follows:

$$L_S(\mathbf{W}^{(t)}) = \Theta\Big(\frac{1}{t(\log t)^{2-2/L}}\Big), \qquad \|\mathbf{W}^{(t)}\|_F = \Theta\big((\log t)^{1/L}\big),$$

where $L$ is the order of the homogeneous network satisfying the property $f(c\mathbf{W}, \mathbf{x}) = c^L f(\mathbf{W}, \mathbf{x})$ for all $c > 0$, $\mathbf{W}$, and $\mathbf{x}$. It is worth noting that the two-layer (leaky) ReLU neural network analyzed in this paper is 1-homogeneous but not smooth. In Section 5.1, we have already demonstrated that $\|\mathbf{W}^{(t)}\|_F = \Theta(\log t)$, and in this subsection, we will discuss the proof technique employed to show a convergence rate of $\Theta(t^{-1})$ for the loss and establish the same normalized margin for all the training data points asymptotically. These results align with those presented by Lyu and Li (2019) regarding smooth homogeneous networks.

By the nearly orthogonal property, we can bound the increment of margin as follows:

$$\frac{\eta}{5nm} \cdot |\ell_i'^{(t)}| \cdot \|\mathbf{x}_i\|_2^2 \leq y_i f(\mathbf{W}^{(t+1)}, \mathbf{x}_i) - y_i f(\mathbf{W}^{(t)}, \mathbf{x}_i) \leq \frac{3\eta}{nm} \cdot |\ell_i'^{(t)}| \|\mathbf{x}_i\|_2^2. \qquad (5.12)$$

Given (5.8) and (5.12), we can apply Lemma 5.3 and obtain

$$\Big| y_i f(\mathbf{W}^{(t)}, \mathbf{x}_i) - \log t - \log(\eta \|\mathbf{x}_i\|_2^2/nm) \Big| \leq C_3, \qquad (5.13)$$

where $C_3$ is a constant. Utilizing (5.13) and the inequality $z - z^2/2 \leq \log(1 + z) \leq z$ for $z \geq 0$, we can derive:

$$L_S(\mathbf{W}^{(t)}) \leq \frac{1}{n} \sum_{i=1}^n \exp\big(-y_i f(\mathbf{W}^{(t)}, \mathbf{x}_i)\big)$$

$$\leq \frac{1}{n} \sum_{i=1}^n \exp\big(-\log t - \log(\eta \|\mathbf{x}_i\|_2^2/nm) + C_3\big) = O(t^{-1}),$$

$$L_S(\mathbf{W}^{(t)}) \geq \frac{1}{n} \sum_{i=1}^n \exp\big(-y_i f(\mathbf{W}^{(t)}, \mathbf{x}_i)\big) - \exp\big(-2y_i f(\mathbf{W}^{(t)}, \mathbf{x}_i)\big) = \Omega(t^{-1}).$$

To demonstrate that all the training data points attain the same normalized margin as $t$ goes to infinity, we first observe that (5.12) provides the following bounds for the increment of margin difference:

$$y_k f(\mathbf{W}^{(t+1)}, \mathbf{x}_k) - y_i f(\mathbf{W}^{(t+1)}, \mathbf{x}_i)$$
$$\leq y_k f(\mathbf{W}^{(t)}, \mathbf{x}_k) - y_i f(\mathbf{W}^{(t)}, \mathbf{x}_i) + \frac{3\eta}{nm} \cdot |\ell_k'^{(t)}| \cdot \|\mathbf{x}_k\|_2^2 - \frac{\eta}{5nm} \cdot |\ell_i'^{(t)}| \cdot \|\mathbf{x}_i\|_2^2. \qquad (5.14)$$

Now, we consider two cases:

- If the ratio $|\ell_i'^{(t)}|/|\ell_k'^{(t)}|$ is relatively large, then $y_k f(\mathbf{W}^{(t)}, \mathbf{x}_k) - y_i f(\mathbf{W}^{(t)}, \mathbf{x}_i)$ will not increase.
- If the ratio $|\ell_i'^{(t)}|/|\ell_k'^{(t)}|$ is relatively small, then $y_k f(\mathbf{W}^{(t)}, \mathbf{x}_k) - y_i f(\mathbf{W}^{(t)}, \mathbf{x}_i)$ will also be relatively small. In fact, it can be bounded by a constant due to the fact that $|\ell_i'^{(t)}|/|\ell_k'^{(t)}|$ can be approximated by $\exp(y_k f(\mathbf{W}^{(t)}, \mathbf{x}_k) - y_i f(\mathbf{W}^{(t)}, \mathbf{x}_i))$. By (5.14), we can show that

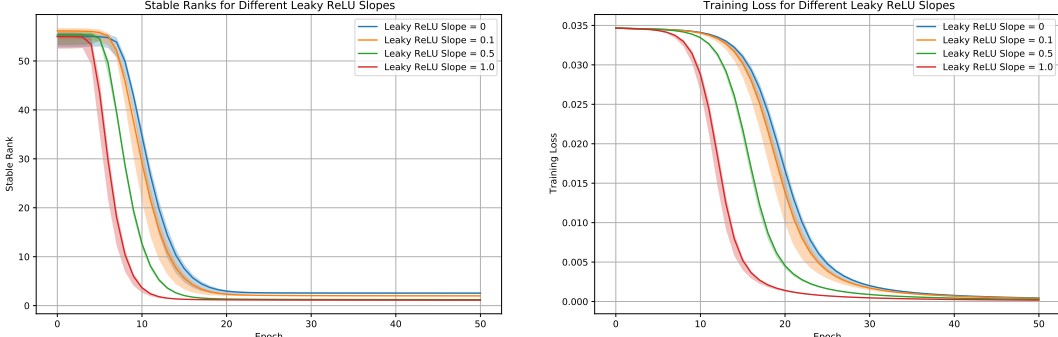

Figure 1: Stable ranks and training loss for different leaky ReLU slopes $\gamma$ across multiple runs. A slope of 1 corresponds to linear activation, while a slope of 0 corresponds to ReLU activation. Each line represents the mean stable rank or training loss for a given leaky ReLU slope, while the shaded regions indicate the variability of the values ($\pm 3$ times the standard deviation) across the 5 runs.

$y_k f(\mathbf{W}^{(t+1)}, \mathbf{x}_k) - y_i f(\mathbf{W}^{(t+1)}, \mathbf{x}_i)$ can also be bounded by a constant, provided that the learning rate $\eta$ is sufficiently small.

By combining both cases, we can conclude that both $|\ell_i'^{(t)}|/|\ell_k'^{(t)}|$ and $y_k f(\mathbf{W}^{(t)}, \mathbf{x}_k) - y_i f(\mathbf{W}^{(t)}, \mathbf{x}_i)$ can be bounded by constants. This result is formally stated in the following lemma.

**Lemma 5.9.** For two-layer neural networks defined in (3.1) with (leaky) ReLU activation, the following bounds hold for any $t \geq 0$:

$$y_i f(\mathbf{W}^{(t)}, \mathbf{x}_i) - y_k f(\mathbf{W}^{(t)}, \mathbf{x}_k) \leq C_1, \quad \ell_i'^{(t)}/\ell_k'^{(t)} \leq C_2, \tag{5.15}$$

for any $i, k \in [n]$, where $C_1, C_2$ are positive constants.

By Lemma 5.9, which shows that the difference between the margins of any two data points can be bounded by a constant, and taking into account that $\|\mathbf{W}^{(t)}\|_F = \Theta(\log t)$, we can deduce the following result:

$$\lim_{t \to \infty} \left| y_i f(\mathbf{W}^{(t)}/\|\mathbf{W}^{(t)}\|_F, \mathbf{x}_i) - y_k f(\mathbf{W}^{(t)}/\|\mathbf{W}^{(t)}\|_F, \mathbf{x}_k) \right| = 0, \forall i, k \in [n].$$

This demonstrates that gradient descent will asymptotically find a neural network in which all the training data points achieve the same normalized margin.

## 6 Experiments

In this section, we present simulations of both synthetic and real data to back up our theoretical analysis in the previous section.

**Synthetic-data experiments.** Here we generate a synthetic mixture of Gaussian data as follows: Let $\boldsymbol{\mu} \in \mathbb{R}^d$ be a fixed vector representing the signal contained in each data point. Each data point $(\mathbf{x}, y)$ with predictor $\mathbf{x} \in \mathbb{R}^d$ and label $y \in \{-1, 1\}$ is generated from a distribution $\mathcal{D}$, which we specify as follows:

1. The label $y$ is generated as a Rademacher random variable, i.e. $\mathbb{P}[y = 1] = \mathbb{P}[y = -1] = 1/2$.

2. A noise vector $\boldsymbol{\xi}$ is generated from the Gaussian distribution $\mathcal{N}(\mathbf{0}, \sigma_p^2 \mathbf{I}_d)$. And $\mathbf{x}$ is assigned as $y \cdot \boldsymbol{\mu} + \boldsymbol{\xi}$ where $\boldsymbol{\mu}$ is a fixed feature vector.

Specifically, we set training data size $n = 10, d = 784$ and train the NN with gradient descent using learning rate 0.1 for 50 epochs. We set $\boldsymbol{\mu}$ to be a feature randomly drawn from $\mathcal{N}(0, 10^{-4}\mathbf{I}_d)$. We then generate the noise vector $\boldsymbol{\xi}$ from the Gaussian distribution $\mathcal{N}(\mathbf{0}, \sigma_p^2 \mathbf{I})$ with fixed standard deviation $\sigma_p = 1$. We train the FNN model defined in Section 3 with ReLU (or leaky-RelU) activation function and width $m = 100$. As we can infer from Figure 1, the stable rank will decrease faster for larger leaky ReLU slopes and have a smaller value when epoch $t \to \infty$.

**Real-data experiments on MNIST dataset.** Here we train a two-layer feed-forward neural network defined in Section 3 with ReLU (or leaky-ReLU) functions. The number of widths is

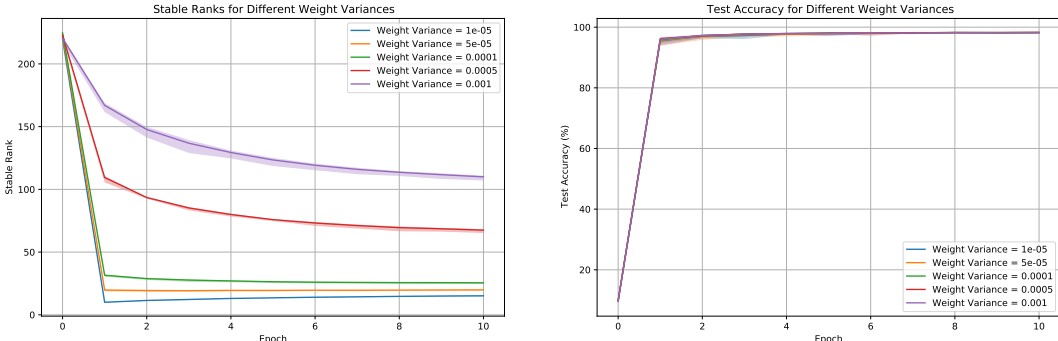

Figure 2: Stable ranks and test errors for different weight variances across multiple runs (ReLU Activation Function). Each line represents the mean stable rank or test accuracy for a given weight variance, while the shaded regions indicate the variability of the values ($\pm 3$ times the standard deviation) across the 5 runs.

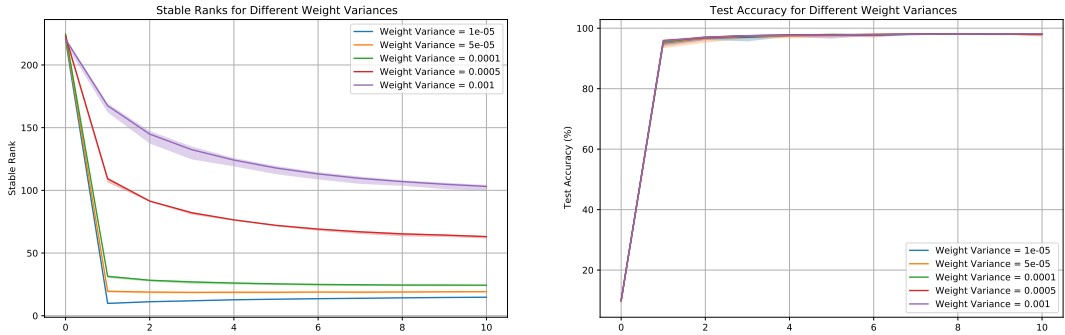

Figure 3: Stable ranks and test errors for different weight variances across multiple runs (leaky-ReLU Activation Function with slope $0.1$). Each line represents the mean stable rank or test accuracy for a given weight variance, while the shaded regions indicate the variability of the values ($\pm 3$ times the standard deviation) across the 5 runs.

set as $m = 1000$. We use the Gaussian initialization and consider different weight variance $\sigma_0 \in \{0.00001, 0.00005, 0.0001, 0.0005, 0.001\}$. We train the NN with stochastic gradient descent with batch size $64$ and learning rate $0.1$ for $10$ epochs. As we can infer from Figures 2 and 3, the stable rank of ReLU or leaky ReLU networks will largely depend on the initialization and the training time. When initialization is sufficiently small, the stable rank will quickly decrease to a small value compared to its initialization values.

## 7 Conclusion and Future Work

This paper employs a data-correlated decomposition technique to examine the implicit bias of two-layer ReLU and Leaky ReLU networks trained using gradient descent. By analyzing the training dynamics, we provide precise characterizations of the weight matrix stable rank limits for both ReLU and Leaky ReLU cases, demonstrating that both scenarios will yield a network with a low stable rank. Additionally, we present an analysis for the convergence rate of the loss function. An important future work is to investigate the directional convergence of the weight matrix in neural networks trained via gradient descent, which is essential to prove the convergence to a KKT point of the max-margin problem. Furthermore, it is important to extend our analysis to fully understand the neuron activation patterns in ReLU networks. Specifically, we will explore whether certain neurons will switch their activation patterns by an infinite number of times throughout the training or if the activation patterns stabilize after a certain number of gradient descent iterations.

## Acknowledgements

We thank the anonymous reviewers and area chair for their helpful comments. YK, ZC, and QG are supported in part by the National Science Foundation CAREER Award 1906169 and IIS-2008981, and the Sloan Research Fellowship. The views and conclusions contained in this paper are those of the authors and should not be interpreted as representing any funding agencies.

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

# A  Additional Experiments

In this section, we conduct additional experiments on the nearly orthogonal and MINIST datasets.

## A.1  Additional Experiment on Nearly Orthogonal Dataset

In this subsection, we conduct additional experiments on a nearly orthogonal dataset for long epochs to support our main Theorems 4.1 and 4.3.

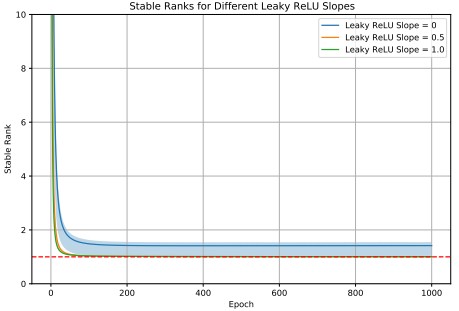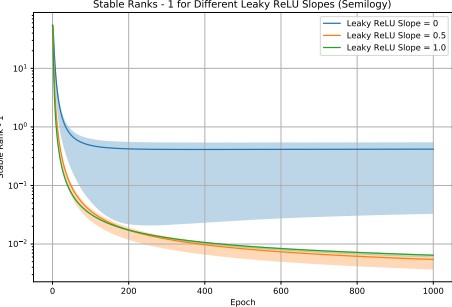

Figure 4: (Left) Stable ranks for different leaky ReLU slopes $\gamma$ across multiple runs. A slope of 1 corresponds to linear activation, while a slope of 0 corresponds to ReLU activation. Each line represents the mean stable rank for a given leaky ReLU slope, while the shaded regions indicate the variability of the values ($\pm 3$ times the standard deviation) across the 20 runs. The red dashed line indicates a stable rank of 1. (Right) The difference between stable rank and 1 from the left figure is visualized on a semilog y-axis.

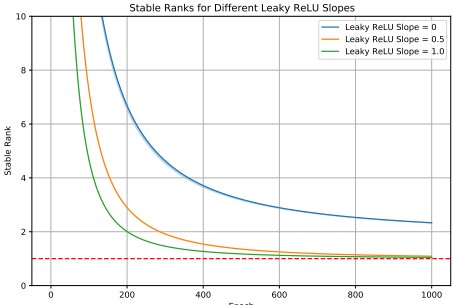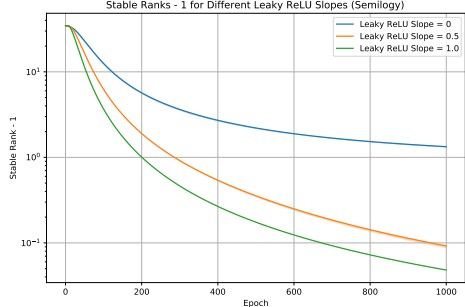

Figure 5: **fully orthogonal data**: (Left) Stable ranks for different leaky ReLU slopes $\gamma$ across multiple runs. A slope of 1 corresponds to linear activation, while a slope of 0 corresponds to ReLU activation. Each line represents the mean stable rank for a given leaky ReLU slope, while the shaded regions indicate the variability of the values ($\pm 3$ times the standard deviation) across the 20 runs. The red dashed line indicates a stable rank of 1. (Right) The difference between stable rank and 1 from the left figure is visualized on a semilog y-axis.

**Figure 4:** Under the same setting of the synthetic data introduced in Section 6, we train the NN with full batch gradient descent with a learning rate 0.1 for 1000 epochs. We set $\boldsymbol{\mu}$ to be a feature randomly drawn from $\mathcal{N}(0, 10^{-4}\mathbf{I}_d)$. We then generate the noise vector $\boldsymbol{\xi}$ from the Gaussian distribution $\mathcal{N}(\mathbf{0}, \sigma_p^2\mathbf{I}_d)$ with fixed standard deviation $\sigma_p = 1$. We train the FNN model defined in Section 3 with ReLU (or leaky-ReLU) activation function and width $m = 100$. As we can see from Figure 4, the stable rank for the leaky ReLU network with large slopes $\gamma$ will converge to 1 when epoch $t \to \infty$. In comparison, the stable rank for the ReLU network will not converge to 1.

**Figure 5:** To further illustrate the behavior of the ReLU network, we generate the synthetic training data with fully orthogonal input. Each data point $(\mathbf{x}, y)$ with input $\mathbf{x} \in \mathbb{R}^d$ and label $y \in \{-1, 1\}$ is generated from a distribution $\mathcal{D}$, which we specify as follows:

1. The label $y$ is generated as a Rademacher random variable, i.e., $\mathbb{P}[y = 1] = \mathbb{P}[y = -1] = 1/2$.

2. Input $\mathbf{x}$ is randomly generated from the basis $\{\mathbf{e}_1, \mathbf{e}_2, \ldots, \mathbf{e}_d\}$.

Specifically, we set training data size $n = 20, d = 40$ and train the NN with full batch gradient descent with a learning rate 0.1 for 1000 epochs. We train the FNN model defined in Section 3 with ReLU (or leaky-ReLU) activation function and width $m = 10000$. As we can observe from Figure 5,

the stable rank for the leaky ReLU network with large slopes $\gamma$ will converge to 1 when epoch $t \to \infty$. In comparison, the stable rank for the large-width ReLU network will not converge to 1 but to 2.

## A.2 Additional Experiment on MINIST

Our focus is the training of a two-layer feed-forward neural network, as discussed in Section 3, utilizing either ReLU or leaky-ReLU activation functions. We examine different widths, specifically choosing from $\{10, 50, 100, 500, 1000\}$.

The network initialization process follows a Gaussian distribution, with a variance of $\sigma_0 = 0.00001$. Training is executed using stochastic gradient descent, a batch size of $64$, and a learning rate of $0.1$, for a total of 10 epochs. As discerned from Figures 6 and 7, the stable rank of networks utilizing either ReLU or leaky ReLU is weakly influenced by the width. For an exceedingly small width such as 10, the weight matrix is low rank with a correspondingly small stable rank. However, this also results in low test accuracy as the network cannot effectively learn all necessary features. As the width increases, the test accuracy and final stable rank will increase. However, for sufficiently large widths, an increase in width no longer corresponds to stable rank or test accuracy increases.

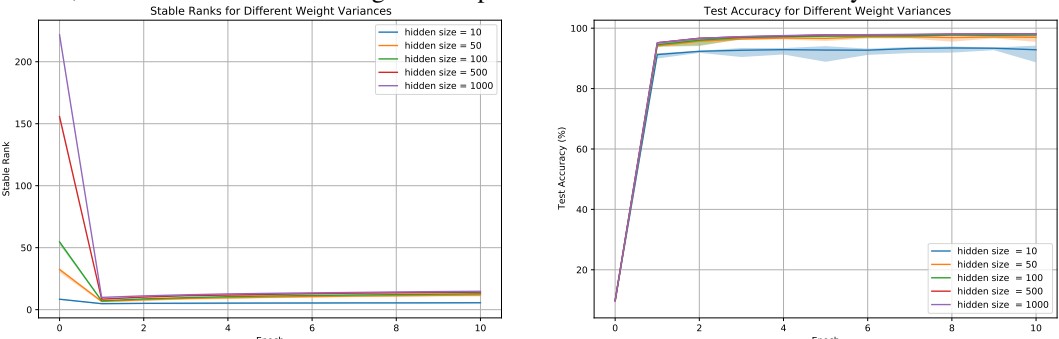

Figure 6: Stable ranks and test errors for different width across multiple runs (ReLU Activation Function). Each line represents the mean stable rank or test error for a given weight variance, while the shaded regions indicate the variability of the values (±3 times the standard deviation) across the 5 runs.

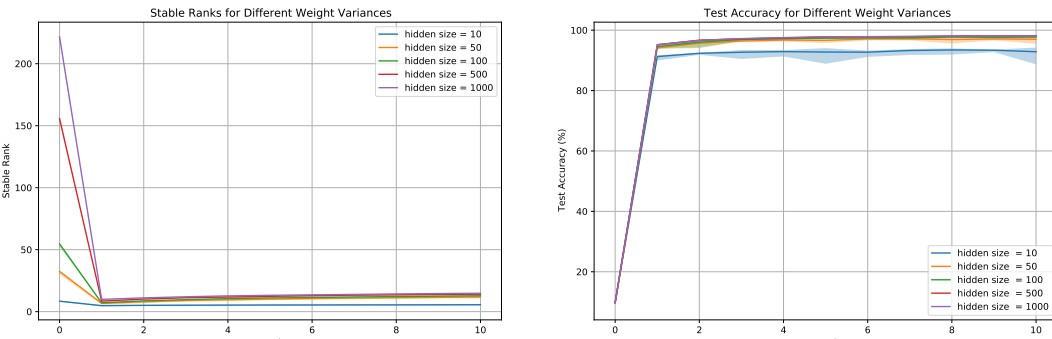

Figure 7: Stable ranks and test errors for different width across multiple runs (leaky-ReLU Activation Function). Each line represents the mean stable rank or test error for a given weight variance, while the shaded regions indicate the variability of the values (±3 times the standard deviation) across the 5 runs.

## B Preliminary Lemmas

In this section, we present some pivotal lemmas that illustrate some important properties of the data and neural network parameters at their random initialization and provide the update rule of coefficients from data-correlated decomposition.

Now turning to network initialization, the following lemma studies the inner product between a randomly initialized neural network neuron $\mathbf{w}_{j,r}^{(0)}$ ($j \in \{\pm 1\}$ and $r \in [m]$) and the training data. The calculations characterize how the neural network at initialization randomly captures the information in training data.

**Lemma B.1.** Suppose that $d = \Omega(\log(mn/\delta))$, $m = \Omega(\log(1/\delta))$. Then with probability at least $1 - \delta$,

$$\sigma_0^2 d/2 \leq \|\mathbf{w}_{j,r}^{(0)}\|_2^2 \leq 3\sigma_0^2 d/2,$$

$$|\langle \mathbf{w}_{j,r}^{(0)}, \mathbf{x}_i \rangle| \leq \sqrt{2\log(8mn/\delta)} \cdot \sigma_0 R_{\max}$$

for all $r \in [m]$, $j \in \{\pm 1\}$ and $i \in [n]$.

*Proof of Lemma B.1.* First of all, the initial weights $\mathbf{w}_{j,r}^{(0)} \sim \mathcal{N}(\mathbf{0}, \sigma_0 \mathbf{I})$. By Bernstein's inequality, with probability at least $1 - \delta/(4m)$ we have

$$\left| \|\mathbf{w}_{j,r}^{(0)}\|_2^2 - \sigma_0^2 d \right| = O(\sigma_0^2 \cdot \sqrt{d\log(8m/\delta)}).$$

Therefore, if we set appropriately $d = \Omega(\log(m/\delta))$, we have with probability at least $1 - \delta/2$, for all $j \in \{\pm 1\}$ and $r \in [m]$,

$$\sigma_0^2 d/2 \leq \|\mathbf{w}_{j,r}^{(0)}\|_2^2 \leq 3\sigma_0^2 d/2.$$

Under definition, we have $\|\mathbf{x}_i\|_2 \leq R_{\max}$ for all $i \in [n]$. It is clear that for each $j, r$, $\langle \mathbf{w}_{j,r}^{(0)}, \boldsymbol{\mu} \rangle$ is a Gaussian random variable with mean zero and variance $\sigma_0^2 \|\mathbf{x}_i\|_2^2$. Therefore, by Gaussian tail bound and union bound, with probability at least $1 - \delta/2$,

$$|\langle \mathbf{w}_{j,r}^{(0)}, \mathbf{x}_i \rangle| \leq \sqrt{2\log(8mn/\delta)} \cdot \sigma_0 R_{\max}.$$

$\square$

Next, we denote $S_i^{(0)}$ as $\{r \in [m] : \langle \mathbf{w}_{y_i,r}^{(0)}, \mathbf{x}_i \rangle > 0\}$. We give a lower bound of $|S_i^{(0)}|$ in the following two lemmas.

**Lemma B.2.** Suppose that $\delta > 0$ and $m \geq 50\log(2n/\delta)$. Then with probability at least $1 - \delta$,

$$0.4m \leq |S_i^{(0)}| \leq 0.6m, \ \forall i \in [n].$$

*Proof of Lemma B.2.* Note that $|S_i^{(0)}| = \sum_{r=1}^m \mathbb{1}[\langle \mathbf{w}_{y_i,r}^{(0)}, \mathbf{x}_i \rangle > 0]$ and $P(\langle \mathbf{w}_{y_i,r}^{(0)}, \mathbf{x}_i \rangle > 0) = 1/2$, then by Hoeffding's inequality, with probability at least $1 - \delta/n$, we have

$$\left| \frac{|S_i^{(0)}|}{m} - \frac{1}{2} \right| \leq \sqrt{\frac{\log(2n/\delta)}{2m}}.$$

Therefore, as long as $m \geq 50\log(2n/\delta)$, by applying union bound, with probability at least $1 - \delta$, we have

$$0.4m \leq |S_i^{(0)}| \leq 0.6m, \ \forall i \in [n].$$

$\square$

Now we give the update rule of coefficients from data-correlated decomposition. We will begin by analyzing the coefficients in the data-correlated decomposition in Definition 5.1. The following lemma presents an iterative expression for the coefficients.

**Lemma B.3.** (Restatement of Lemma 5.2) The coefficients $\overline{\rho}_{j,r,i}^{(t)}, \underline{\rho}_{j,r,i}^{(t)}$ defined in Definition 5.1 satisfy the following iterative equations:

$$\overline{\rho}_{j,r,i}^{(0)}, \underline{\rho}_{j,r,i}^{(0)} = 0,$$

$$\overline{\rho}_{j,r,i}^{(t+1)} = \overline{\rho}_{j,r,i}^{(t)} - \frac{\eta}{nm} \cdot \ell_i'^{(t)} \cdot \sigma'(\langle \mathbf{w}_{j,r}^{(t)}, \mathbf{x}_i \rangle) \cdot \|\mathbf{x}_i\|_2^2 \cdot \mathbb{1}(y_i = j),$$

$$\underline{\rho}_{j,r,i}^{(t+1)} = \underline{\rho}_{j,r,i}^{(t)} + \frac{\eta}{nm} \cdot \ell_i'^{(t)} \cdot \sigma'(\langle \mathbf{w}_{j,r}^{(t)}, \mathbf{x}_i \rangle) \cdot \|\mathbf{x}_i\|_2^2 \cdot \mathbb{1}(y_i = -j),$$

for all $r \in [m]$, $j \in \{\pm 1\}$ and $i \in [n]$.

*Proof of Lemma B.3.* First, we iterate the gradient descent update rule (3.2) $t$ times and get

$$\mathbf{w}_{j,r}^{(t+1)} = \mathbf{w}_{j,r}^{(0)} - \frac{\eta}{nm} \sum_{s=0}^{t} \sum_{i=1}^{n} \ell_i'^{(s)} \cdot \sigma'(\langle \mathbf{w}_{j,r}^{(s)}, \mathbf{x}_i \rangle) \cdot jy_i \mathbf{x}_i.$$

According to the definition of $\rho_{j,r,i}^{(t)}$, we have

$$\mathbf{w}_{j,r}^{(t)} = \mathbf{w}_{j,r}^{(0)} + \sum_{i=1}^{n} \rho_{j,r,i}^{(t)} \cdot \|\mathbf{x}_i\|_2^{-2} \cdot \mathbf{x}_i.$$

Therefore, we have the unique representation

$$\rho_{j,r,i}^{(t)} = -\frac{\eta}{nm} \sum_{s=0}^{t} \ell_i'^{(s)} \cdot \sigma'(\langle \mathbf{w}_{j,r}^{(s)}, \mathbf{x}_i \rangle) \cdot \|\mathbf{x}_i\|_2^2 \cdot jy_i.$$

Now with the notation $\overline{\rho}_{j,r,i}^{(t)} := \rho_{j,r,i}^{(t)} \mathbb{1}(\rho_{j,r,i}^{(t)} \geq 0)$, $\underline{\rho}_{j,r,i}^{(t)} := \rho_{j,r,i}^{(t)} \mathbb{1}(\rho_{j,r,i}^{(t)} \leq 0)$ and the fact $\ell_i'^{(s)} < 0$, we get

$$\overline{\rho}_{j,r,i}^{(t)} = -\frac{\eta}{nm} \sum_{s=0}^{t} \ell_i'^{(s)} \cdot \sigma'(\langle \mathbf{w}_{j,r}^{(s)}, \mathbf{x}_i \rangle) \cdot \|\mathbf{x}_i\|_2^2 \cdot \mathbb{1}(y_i = j), \tag{B.1}$$

$$\underline{\rho}_{j,r,i}^{(t)} = \frac{\eta}{nm} \sum_{s=0}^{t} \ell_i'^{(s)} \cdot \sigma'(\langle \mathbf{w}_{j,r}^{(s)}, \mathbf{x}_i \rangle) \cdot \|\mathbf{x}_i\|_2^2 \cdot \mathbb{1}(y_i = -j). \tag{B.2}$$

Writing out the iterative versions of (B.1) and (B.2) completes the proof. $\qquad\square$

# C   Coefficient Analysis of Leaky ReLU

In this section, we establish a series of results on the data-correlated decomposition for two-layer leaky ReLU network defined as

$$f(\mathbf{W}^{(t)}, \mathbf{x}) = F_{+1}(\mathbf{W}_{+1}^{(t)}, \mathbf{x}) - F_{-1}(\mathbf{W}_{-1}^{(t)}, \mathbf{x})$$
$$= \frac{1}{m} \sum_{r=1}^{m} \sigma(\langle \mathbf{w}_{+1,r}^{(t)}, \mathbf{x} \rangle) - \frac{1}{m} \sum_{r=1}^{m} \sigma(\langle \mathbf{w}_{-1,r}^{(t)}, \mathbf{x} \rangle), \tag{C.1}$$
$$\sigma(z) = \max\{\gamma z, z\}, \gamma \in (0, 1).$$

The results in Section C, D and G are based on Lemma B.1, which hold with high probability. Denote by $\mathcal{E}_{\text{prelim}}$ the event that Lemma B.1 in Section B holds (for a given $\delta$, we see $\mathbb{P}(\mathcal{E}_{\text{prelim}}) \geq 1 - \delta$). For simplicity and clarity, we state all the results in Section C, D and G conditional on $\mathcal{E}_{\text{prelim}}$.

Denote $\beta = \max_{i,j,r}\{|\langle \mathbf{w}_{j,r}^{(0)}, \mathbf{x}_i \rangle|\}$, $R_{\max} = \max_{i \in [n]} \|\mathbf{x}_i\|_2$, $R_{\min} = \min_{i \in [n]} \|\mathbf{x}_i\|_2$, $p = \max_{i \neq k} |\langle \mathbf{x}_i, \mathbf{x}_k \rangle|$ and suppose $R = R_{\max}/R_{\min}$ is at most an absolute constant. Here we list the exact conditions for $\eta, \sigma_0, R_{\min}, R_{\max}, p$ required by the proofs in this section.

$$\sigma_0 \leq \gamma \big( C R_{\max} \sqrt{\log(mn/\delta)} \big)^{-1}, \tag{C.2}$$
$$\eta \leq (C R_{\max}^2 / nm)^{-1}, \tag{C.3}$$
$$R_{\min}^2 \geq C r^{-4} R^2 np, \tag{C.4}$$

where $C$ is a large enough constant. By Lemma B.1, we can upper bound $\beta$ by $2\sqrt{\log(12mn/\delta)} \cdot \sigma_0 R_{\max}$. Then, by (C.2) and (C.4), it is straightforward to verify the following inequality:

$$\beta \leq c\gamma, \tag{C.5}$$
$$\gamma^{-4} R_{\min}^{-2} np \leq c, \tag{C.6}$$
$$\gamma^{-4} R_{\min}^{-2} R^2 np \leq c, \tag{C.7}$$

where $c$ is a sufficiently small constant.

Suppose the conditions listed in (C.2) and (C.4) hold, we claim that for any $t \geq 0$ the following property holds.

**Lemma C.1.** Under the same conditions as Theorem 4.1, for any $t \geq 0$, we have that

$$\sum_{r=1}^{m} |\rho_{j,r,i}^{(t)}| \geq c_1 \gamma^2 \sum_{r=1}^{m} |\rho_{j',r,i'}^{(t)}|, \forall j, j' \in \{\pm 1\}, \forall i, i' \in [n], \tag{C.8}$$

where $c_1$ is a constant.

To prove Lemma C.1, we divide it into two lemmas, each addressing a specific case: $0 \leq t \leq T_1$ (Lemma C.2) when the logit $|\ell_i^{(t)}| = \Theta(1)$, and $t \geq T_1$ (Lemma C.3) when the logit $|\ell_i^{(t)}|$ is smaller than constant order. Here, $T_1 = C'\eta^{-1}nmR_{\max}^{-2}$, and $C'$ is a constant. For each case, we apply different techniques to establish the proof.

**Lemma C.2** ($0 \leq t \leq T_1$). Under the same conditions as Theorem 4.1, for any $0 \leq t \leq T_1 = C'\eta^{-1}nmR_{\max}^{-2}$, where $C'$ is a constant, we have that

$$|\rho_{j,r,i}^{(t)}| \geq c_2 \gamma |\rho_{j',r',i'}^{(t)}|, \forall j, j' \in \{\pm 1\}, \forall r, r' \in [m], \forall i, i' \in [n], \tag{C.9}$$

where $c_2$ is a constant.

*Proof of Lemma C.2.* In this lemma, we first show that (C.8) hold for $t \leq T_1 = C'\eta^{-1}nmR_{\max}^{-2}$ where $C' = \Theta(1)$ is a constant. Recall from Lemma B.3 that

$$\overline{\rho}_{j,r,i}^{(t+1)} = \overline{\rho}_{j,r,i}^{(t)} - \frac{\eta}{nm} \cdot \ell_i^{\prime(t)} \cdot \sigma'(\langle \mathbf{w}_{j,r}^{(t)}, \mathbf{x}_i \rangle) \cdot \|\mathbf{x}_i\|_2^2 \cdot \mathbb{1}(y_i = j),$$

$$\underline{\rho}_{j,r,i}^{(t+1)} = \underline{\rho}_{j,r,i}^{(t)} + \frac{\eta}{nm} \cdot \ell_i^{\prime(t)} \cdot \sigma'(\langle \mathbf{w}_{j,r}^{(t)}, \mathbf{x}_i \rangle) \cdot \|\mathbf{x}_i\|_2^2 \cdot \mathbb{1}(y_i = -j),$$

we can get

$$\overline{\rho}_{-y_i,r,i}^{(t)}, \underline{\rho}_{y_i,r,i}^{(t)} = 0, \tag{C.10}$$

and

$$\overline{\rho}_{y_i,r,i}^{(t+1)} \leq \overline{\rho}_{y_i,r,i}^{(t)} + \frac{\eta}{nm} \cdot \|\mathbf{x}_i\|_2^2 \leq \overline{\rho}_{y_i,r,i}^{(t)} + \frac{\eta R_{\max}^2}{nm}, \tag{C.11}$$

$$|\underline{\rho}_{-y_i,r,i}^{(t+1)}| \leq |\underline{\rho}_{-y_i,r,i}^{(t)}| + \frac{\eta}{nm} \cdot \|\mathbf{x}_i\|_2^2 \leq |\underline{\rho}_{-y_i,r,i}^{(t)}| + \frac{\eta R_{\max}^2}{nm}. \tag{C.12}$$

Therefore, we have $\max_{j,r,i}\{\overline{\rho}_{j,r,i}^{(t)}, |\underline{\rho}_{j,r,i}^{(t)}|\} = O(1)$ for any $t \leq T_1$ and hence $\max_i\{F_{+1}(\mathbf{W}_{+1}^{(t)}, \mathbf{x}_i), F_{-1}(\mathbf{W}_{-1}^{(t)}, \mathbf{x}_i)\} = O(1)$ for any $t \leq T_1$. Thus there exists a positive constant $\widetilde{c}$ such that $|\ell_i^{\prime(t)}| \geq \widetilde{c}$ for any $t \leq T_1$. And it follows for any $j \in \{\pm 1\}, r \in [m], i \in [n]$ that

$$|\rho_{j,r,i}^{(t+1)}| \geq |\rho_{j,r,i}^{(t)}| + \frac{\gamma\eta}{nm} \cdot |\ell_i^{\prime(t)}| \cdot \|\mathbf{x}_i\|_2^2 \geq |\rho_{j,r,i}^{(t)}| + \frac{\widetilde{c}\gamma\eta}{nm} \cdot \|\mathbf{x}_i\|_2^2, \forall 0 \leq t \leq T_1,$$

$$|\rho_{j,r,i}^{(t)}| \geq \frac{\widetilde{c}\gamma\eta t}{nm} \cdot \|\mathbf{x}_i\|_2^2 \geq \frac{\widetilde{c}\gamma\eta R_{\min}^2 t}{nm}, \forall 0 \leq t \leq T_1. \tag{C.13}$$

On the other hand, by (C.10), (C.11) and (C.12), we have for any $j' \in \{\pm 1\}, r' \in [m], i' \in [n]$ that

$$|\rho_{j',r',i'}^{(t)}| \leq \frac{\eta R_{\max}^2 t}{nm}, \forall 0 \leq t \leq T_1. \tag{C.14}$$

Dividing (C.14) by (C.13), we can get for any $j, j' \in \{\pm 1\}, r, r' \in [m], i, i' \in [n]$ that

$$|\rho_{j,r,i}^{(t)}| \geq \frac{\widetilde{c}\gamma R_{\min}^2}{R_{\max}^2} |\rho_{j',r',i'}^{(t)}|,$$

which indicates that the first bullet holds for time $t \leq T_1$ as long as $c_2 \leq \widetilde{c} R_{\min}^2 R_{\max}^{-2}$. $\square$

**Lemma C.3** ($t \geq T_1$). Let $T_1$ be defined in Lemma C.2. Under the same conditions as Theorem 4.1, for any $t \geq T_1$, we have that

$$\sum_{r=1}^{m} |\rho_{j,r,i}^{(t)}| \geq c_3 \gamma^2 \sum_{r=1}^{m} |\rho_{j',r,i'}^{(t)}|, \forall j, j' \in \{\pm 1\}, \forall i, i' \in [n], \tag{C.15}$$

where $c_3 = \Theta(1)$ is a constant. Moreover, we also have the following increasing rate estimation of $|\rho_{y_i,r,i}^{(t)}|, |\rho_{-y_i,r,i}^{(t)}|$:

- $\frac{1}{m} \sum_{r=1}^{m} \rho_{y_i,r,i}^{(t)} \leq c_4^{-1} \log \left( 1 + \frac{\eta \|\mathbf{x}_i\|_2^2 c_4 e^{2\beta}}{nm} \cdot t \right),$

- $\frac{1}{m} \sum_{r=1}^{m} |\rho_{-y_i,r,i}^{(t)}| \leq c_5^{-1} \gamma^{-1} \log \left( 1 + \frac{\gamma \eta \|\mathbf{x}_i\|_2^2 c_5 e^{2\beta}}{nm} \cdot t \right),$

- $\frac{1}{m} \sum_{r=1}^{m} \rho_{y_i,r,i}^{(t)} \geq c_6^{-1} \log \left( 1 + \frac{\gamma \eta \|\mathbf{x}_i\|_2^2 c_6 e^{-(\gamma+1)\beta}}{nm} \cdot t \right),$

- $\frac{1}{m} \sum_{r=1}^{m} |\rho_{-y_i,r,i}^{(t)}| \geq c_6^{-1} \gamma \log \left( 1 + \frac{\eta \|\mathbf{x}_i\|_2^2 c_6 e^{-(\gamma+1)\beta}}{nm} \cdot t \right),$

where $c_4, c_5, c_6$ are constants.

*Proof of Lemma C.3.* We prove this lemma by induction. By Lemma C.2, we know that (C.15) holds for time $t = T_1$ as long as $c_3 \leq c_2$. Suppose that there exists $\widetilde{t} > T_1$ such that (C.15) holds for all time $0 \leq t \leq \widetilde{t} - 1$. We aim to prove that they also hold for $t = \widetilde{t}$. For any $0 \leq t \leq \widetilde{t} - 1$, we have

$$
\begin{aligned}
F_{y_i}(\mathbf{W}_{y_i}^{(t)}, \mathbf{x}_i) &= \frac{1}{m} \sum_{r=1}^{m} \sigma(\langle \mathbf{w}_{y_i,r}^{(t)}, \mathbf{x}_i \rangle) \\
&\geq \frac{1}{m} \sum_{r=1}^{m} \langle \mathbf{w}_{y_i,r}^{(t)}, \mathbf{x}_i \rangle \\
&= \frac{1}{m} \sum_{r=1}^{m} \left( \langle \mathbf{w}_{y_i,r}^{(0)}, \mathbf{x}_i \rangle + \sum_{i'=1}^{n} \rho_{y_i,r,i'}^{(t)} \|\mathbf{x}_{i'}\|_2^{-2} \cdot \langle \mathbf{x}_{i'}, \mathbf{x}_i \rangle \right) \\
&\geq \frac{1}{m} \sum_{r=1}^{m} \left( \rho_{y_i,r,i}^{(t)} - \sum_{i' \neq i} |\rho_{y_i,r,i'}^{(t)}| R_{\min}^{-2} p \right) - \beta \\
&= \frac{1}{m} \sum_{r=1}^{m} \rho_{y_i,r,i}^{(t)} - \sum_{i' \neq i} \left( \frac{1}{m} \sum_{r=1}^{m} \rho_{y_i,r,i'}^{(t)} \right) R_{\min}^{-2} p - \beta \\
&\geq \frac{1 - \gamma^{-2} c_3^{-1} R_{\min}^{-2} pn}{m} \sum_{r=1}^{m} \rho_{y_i,r,i}^{(t)} - \beta,
\end{aligned}
\tag{C.16}
$$

where the first inequality is by $\sigma(z) \geq z$; the second equality is by (5.1); the third inequality is by triangle inequality and the definition of $\beta, p, R_{\min}$; the fourth inequality is by the induction hypothesis (C.15). Besides, for any $0 \leq t \leq \widetilde{t} - 1$, we also have the following upper bound of

$F_{y_i}(\mathbf{W}_{y_i}^{(t)}, \mathbf{x}_i)$:

$$
\begin{aligned}
F_{y_i}(\mathbf{W}_{y_i}^{(t)}, \mathbf{x}_i) &= \frac{1}{m} \sum_{r=1}^{m} \sigma(\langle \mathbf{w}_{y_i,r}^{(t)}, \mathbf{x}_i \rangle) \\
&= \frac{1}{m} \sum_{r=1}^{m} \sigma\left( \langle \mathbf{w}_{y_i,r}^{(0)}, \mathbf{x}_i \rangle + \sum_{i'=1}^{n} \rho_{y_i,r,i'}^{(t)} \|\mathbf{x}_{i'}\|_2^{-2} \cdot \langle \mathbf{x}_{i'}, \mathbf{x}_i \rangle \right) \\
&\leq \frac{1}{m} \sum_{r=1}^{m} \sigma\left( \rho_{y_i,r,i}^{(t)} + \sum_{i' \neq i} |\rho_{y_i,r,i'}^{(t)}| R_{\min}^{-2} p + \beta \right) \\
&= \frac{1}{m} \sum_{r=1}^{m} \left( \rho_{y_i,r,i}^{(t)} + \sum_{i' \neq i} |\rho_{y_i,r,i'}^{(t)}| R_{\min}^{-2} p + \beta \right) \\
&= \frac{1}{m} \sum_{r=1}^{m} \rho_{y_i,r,i}^{(t)} + \sum_{i' \neq i} \left( \frac{1}{m} \sum_{r=1}^{m} \rho_{y_i,r,i'}^{(t)} \right) R_{\min}^{-2} p + \beta \\
&\leq \frac{1 + \gamma^{-2} c_3^{-1} R_{\min}^{-2} p n}{m} \sum_{r=1}^{m} \rho_{y_i,r,i}^{(t)} + \beta,
\end{aligned}
\tag{C.17}
$$

where the first inequality is by triangle inequality and the definition of $\beta, p, R_{min}$; the second inequality is by the induction hypothesis (C.15). On the other hand, for any $0 \leq t \leq \tilde{t}$, we can give following upper and lower bounds for $F_{-y_i}(\mathbf{W}_{-y_i}^{(t)}, \mathbf{x}_i)$ by applying similar arguments like (C.16) and (C.17):

$$
\begin{aligned}
F_{-y_i}(\mathbf{W}_{-y_i}^{(t)}, \mathbf{x}_i) &\geq \frac{\gamma}{m} \sum_{r=1}^{m} \langle \mathbf{w}_{-y_i,r}^{(t)}, \mathbf{x}_i \rangle \\
&\geq \frac{\gamma}{m} \sum_{r=1}^{m} \left( \rho_{-y_i,r,i}^{(t)} - \sum_{i' \neq i} |\rho_{-y_i,r,i'}^{(t)}| R_{\min}^{-2} p - \beta \right), \\
&\geq \frac{\gamma(1 + \gamma^{-2} c_3^{-1} R_{\min}^{-2} p n)}{m} \sum_{r=1}^{m} \rho_{-y_i,r,i}^{(t)} - \gamma\beta,
\end{aligned}
\tag{C.18}
$$

and

$$
\begin{aligned}
F_{-y_i}(\mathbf{W}_{-y_i}^{(t)}, \mathbf{x}_i) &\leq \frac{1}{m} \sum_{r=1}^{m} \sigma\left( \rho_{-y_i,r,i}^{(t)} + \sum_{i' \neq i} |\rho_{-y_i,r,i'}^{(t)}| R_{\min}^{-2} p + \beta \right) \\
&\leq \frac{1}{m} \sum_{r=1}^{m} \left[ \sigma(\rho_{-y_i,r,i}^{(t)}) + \sigma\left( \sum_{i' \neq i} |\rho_{-y_i,r,i'}^{(t)}| R_{\min}^{-2} p \right) + \sigma(\beta) \right] \\
&= \frac{\gamma}{m} \sum_{r=1}^{m} \rho_{-y_i,r,i}^{(t)} + \sum_{i' \neq i} \left( \frac{1}{m} \sum_{r=1}^{m} |\rho_{-y_i,r,i}^{(t)}| \right) + \beta \\
&= \frac{\gamma(1 - \gamma^{-3} c_3^{-1} R_{\min}^{-2} p n)}{m} \sum_{r=1}^{m} \rho_{-y_i,r,i}^{(t)} + \beta,
\end{aligned}
\tag{C.19}
$$

where the second inequality is by a property of leaky ReLU function that $\sigma(a + b) \leq \sigma(a) + \sigma(b), \forall a, b \in \mathbb{R}$.

Next, we can bound $|\ell_i'^{(t)}|$ for $0 \le t \le \widetilde{t} - 1$:

$$
\begin{aligned}
&|\ell_i'^{(t)}| \\
&= \frac{1}{1 + \exp\{F_{y_i}(\mathbf{W}_{y_i}^{(t)}, \mathbf{x}_i) - F_{-y_i}(\mathbf{W}_{-y_i}^{(t)}, \mathbf{x}_i)\}} \\
&\le \exp\{-F_{y_i}(\mathbf{W}_{y_i}^{(t)}, \mathbf{x}_i) + F_{-y_i}(\mathbf{W}_{-y_i}^{(t)}, \mathbf{x}_i)\} \\
&\le \exp\left\{ -\frac{1 - \gamma^{-2}c_3^{-1}R_{\min}^{-2}pn}{m} \sum_{r=1}^m \rho_{y_i, r, i}^{(t)} + \frac{\gamma(1 - \gamma^{-3}c_3^{-1}R_{\min}^{-2}pn)}{m} \sum_{r=1}^m \rho_{-y_i, r, i}^{(t)} + 2\beta \right\},
\end{aligned}
$$
(C.20)

where the second inequality is by (C.16) and (C.19). And

$$
\begin{aligned}
&|\ell_i'^{(t)}| \\
&= \frac{1}{1 + \exp\{F_{y_i}(\mathbf{W}_{y_i}^{(t)}, \mathbf{x}_i) - F_{-y_i}(\mathbf{W}_{-y_i}^{(t)}, \mathbf{x}_i)\}} \\
&\ge \frac{1}{1 + \exp\left\{ \frac{1 + \gamma^{-2}c_3^{-1}R_{\min}^{-2}pn}{m} \sum_{r=1}^m \rho_{y_i, r, i}^{(t)} - \frac{\gamma(1 + \gamma^{-2}c_3^{-1}R_{\min}^{-2}pn)}{m} \sum_{r=1}^m \rho_{-y_i, r, i}^{(t)} + (\gamma + 1)\beta \right\}} \\
&\ge \frac{1}{2} \exp\left\{ -\frac{1 + \gamma^{-2}c_3^{-1}R_{\min}^{-2}pn}{m} \sum_{r=1}^m \rho_{y_i, r, i}^{(t)} + \frac{\gamma(1 + \gamma^{-2}c_3^{-1}R_{\min}^{-2}pn)}{m} \sum_{r=1}^m \rho_{-y_i, r, i}^{(t)} - (\gamma + 1)\beta \right\},
\end{aligned}
$$
(C.21)

where the first inequality is by (C.17) and (C.18); the last inequality is by $1/(1 + \exp(z)) \ge \exp(-z)/2$ if $z \ge 0$. By (C.20), we can get for $0 \le t \le \widetilde{t} - 1$ that

$$
|\ell_i'^{(t)}| \le \exp\left\{ -\frac{1 - \gamma^{-2}c_3^{-1}R_{\min}^{-2}pn}{m} \sum_{r=1}^m \rho_{y_i, r, i}^{(t)} + 2\beta \right\},
$$
(C.22)

$$
|\ell_i'^{(t)}| \le \exp\left\{ \frac{\gamma(1 - \gamma^{-3}c_3^{-1}R_{\min}^{-2}pn)}{m} \sum_{r=1}^m \rho_{-y_i, r, i}^{(t)} + 2\beta \right\}.
$$
(C.23)

By (C.21) and $\gamma\rho_{y_i, r, i}^{(t)} \le |\rho_{-y_i, r, i}^{(t)}| \le \gamma^{-4}\rho_{y_i, r, i}^{(t)}$, we can get for $0 \le t \le \widetilde{t} - 1$ that

$$
|\ell_i'^{(t)}| \ge \frac{1}{2} \exp\left\{ -\frac{2(1 + \gamma^{-2}c_3^{-1}R_{\min}^{-2}pn)}{m} \sum_{r=1}^m \rho_{y_i, r, i}^{(t)} - (\gamma + 1)\beta \right\},
$$
(C.24)

$$
\begin{aligned}
|\ell_i'^{(t)}| &\ge \frac{1}{2} \exp\left\{ \frac{(\gamma^{-1} + \gamma)(1 + \gamma^{-2}c_3^{-1}R_{\min}^{-2}pn)}{m} \sum_{r=1}^m \rho_{-y_i, r, i}^{(t)} - (\gamma + 1)\beta \right\} \\
&\ge \frac{1}{2} \exp\left\{ \frac{2(1 + \gamma^{-2}c_3^{-1}R_{\min}^{-2}pn)}{\gamma m} \sum_{r=1}^m \rho_{-y_i, r, i}^{(t)} - (\gamma + 1)\beta \right\}.
\end{aligned}
$$
(C.25)

By (5.4), (5.5) and $\sigma' \in [\gamma, 1]$, we have for $0 \le t \le \widetilde{t} - 1$ that

$$
\begin{aligned}
\rho_{y_i, r, i}^{(t+1)} &\le \rho_{y_i, r, i}^{(t)} + \frac{\eta}{nm} \cdot |\ell_i'^{(t)}| \cdot \|\mathbf{x}_i\|_2^2, \\
\rho_{y_i, r, i}^{(t+1)} &\ge \rho_{y_i, r, i}^{(t)} + \frac{\gamma\eta}{nm} \cdot |\ell_i'^{(t)}| \cdot \|\mathbf{x}_i\|_2^2, \\
|\rho_{-y_i, r, i}^{(t+1)}| &\le |\rho_{-y_i, r, i}^{(t)}| + \frac{\eta}{nm} \cdot |\ell_i'^{(t)}| \cdot \|\mathbf{x}_i\|_2^2, \\
|\rho_{-y_i, r, i}^{(t+1)}| &\ge |\rho_{-y_i, r, i}^{(t)}| + \frac{\gamma\eta}{nm} \cdot |\ell_i'^{(t)}| \cdot \|\mathbf{x}_i\|_2^2.
\end{aligned}
$$
(C.26)

By plugging (C.22), (C.24), (C.23) and (C.25) into (C.26), we have for $0 \leq t \leq \tilde{t} - 1$ that

$$\sum_{r=1}^{m} \rho_{y_i,r,i}^{(t+1)} \leq \sum_{r=1}^{m} \rho_{y_i,r,i}^{(t)} + \frac{\eta \|\mathbf{x}_i\|_2^2 e^{2\beta}}{n} \cdot \exp\left\{ -\frac{1 - \gamma^{-2} c_3^{-1} R_{\min}^{-2} pn}{m} \sum_{r=1}^{m} \rho_{y_i,r,i}^{(t)} \right\}, \tag{C.27}$$

$$\sum_{r=1}^{m} |\rho_{-y_i,r,i}^{(t+1)}| \leq \sum_{r=1}^{m} |\rho_{-y_i,r,i}^{(t)}| + \frac{\eta \|\mathbf{x}_i\|_2^2 e^{2\beta}}{n} \cdot \exp\left\{ -\frac{\gamma(1 - \gamma^{-3} c_3^{-1} R_{\min}^{-2} pn)}{m} \sum_{r=1}^{m} |\rho_{-y_i,r,i}^{(t)}| \right\}, \tag{C.28}$$

$$\sum_{r=1}^{m} \rho_{y_i,r,i}^{(t+1)} \geq \sum_{r=1}^{m} \rho_{y_i,r,i}^{(t)} + \frac{\gamma\eta \|\mathbf{x}_i\|_2^2 e^{-(\gamma+1)\beta}}{2n} \cdot \exp\left\{ -\frac{2(1 + \gamma^{-2} c_3^{-1} R_{\min}^{-2} pn)}{m} \sum_{r=1}^{m} \rho_{y_i,r,i}^{(t)} \right\}, \tag{C.29}$$

$$\sum_{r=1}^{m} |\rho_{-y_i,r,i}^{(t+1)}| \geq \sum_{r=1}^{m} |\rho_{-y_i,r,i}^{(t)}| + \frac{\gamma\eta \|\mathbf{x}_i\|_2^2 e^{-(\gamma+1)\beta}}{2n} \cdot \exp\left\{ -\frac{2(1 + \gamma^{-2} c_3^{-1} R_{\min}^{-2} pn)}{\gamma m} \sum_{r=1}^{m} |\rho_{-y_i,r,i}^{(t)}| \right\}. \tag{C.30}$$

By applying Lemma H.1 to (C.27) and taking

$$x_t = \frac{1 - \gamma^{-2} c_3^{-1} R_{\min}^{-2} pn}{m} \sum_{r=1}^{m} \rho_{y_i,r,i}^{(t)},$$

we can get for $0 \leq t \leq \tilde{t}$ that

$$\frac{1}{m} \sum_{r=1}^{m} \rho_{y_i,r,i}^{(t)} \leq c_4^{-1} \log\left( 1 + \frac{\eta \|\mathbf{x}_i\|_2^2 c_4 e^{2\beta}}{nm} \exp\left\{ \frac{\eta \|\mathbf{x}_i\|_2^2 c_4 e^{2\beta}}{nm} \right\} \cdot t \right)$$

$$\leq c_4^{-1} \log\left( 1 + \frac{\eta \|\mathbf{x}_i\|_2^2 c_4 e^{2\beta}}{nm} \cdot t \right), \tag{C.31}$$

where $c_4 := 1 - \gamma^{-2} c_3^{-1} R_{\min}^{-2} pn$ and the last inequality is by $\eta \leq (CR_{\max}^2/nm)^{-1}$ and $C$ is a sufficiently large constant.

By applying Lemma H.1 to (C.28) and taking

$$x_t = \frac{\gamma(1 - \gamma^{-3} c_3^{-1} R_{\min}^{-2} pn)}{m} \sum_{r=1}^{m} |\rho_{-y_i,r,i}^{(t)}|,$$

we can get for $0 \leq t \leq \tilde{t}$ that

$$\frac{1}{m} \sum_{r=1}^{m} |\rho_{-y_i,r,i}^{(t)}| \leq c_5^{-1} \gamma^{-1} \log\left( 1 + \frac{\gamma\eta \|\mathbf{x}_i\|_2^2 c_5 e^{2\beta}}{nm} \exp\left\{ \frac{\gamma\eta \|\mathbf{x}_i\|_2^2 c_5 e^{2\beta}}{nm} \right\} \cdot t \right)$$

$$\leq c_5^{-1} \gamma^{-1} \log\left( 1 + \frac{\gamma\eta \|\mathbf{x}_i\|_2^2 c_5 e^{2\beta}}{nm} \cdot t \right), \tag{C.32}$$

where $c_5 := 1 - \gamma^{-3} c_3^{-1} R_{\min}^{-2} pn$ and the last inequality is by $\eta \leq (CR_{\max}^2/nm)^{-1}$ and $C$ is a sufficiently large constant.

By applying Lemma H.2 to (C.29) and taking

$$x_t = \frac{2(1 + \gamma^{-2} c_3^{-1} R_{\min}^{-2} pn)}{m} \sum_{r=1}^{m} \rho_{y_i,r,i}^{(t)},$$

we can get

$$\frac{1}{m} \sum_{r=1}^{m} \rho_{y_i,r,i}^{(t)} \geq (2c_6)^{-1} \log\left( 1 + \frac{\gamma\eta \|\mathbf{x}_i\|_2^2 c_6 e^{-(\gamma+1)\beta}}{nm} \cdot t \right), \tag{C.33}$$

where $c_6 := 1 + \gamma^{-2} c_3^{-1} R_{\min}^{-2} pn$.

By applying Lemma H.2 to (C.30) and taking

$$x_t = \frac{2(1 + \gamma^{-4}c_3^{-1}R_{\min}^{-2}pn)}{\gamma m} \sum_{r=1}^{m} |\rho_{-y_i,r,i}^{(t)}|,$$

we can get

$$\frac{1}{m} \sum_{r=1}^{m} |\rho_{-y_i,r,i}^{(t)}| \geq (2c_6)^{-1}\gamma \log\left(1 + \frac{\eta\|\mathbf{x}_i\|_2^2 c_6 e^{-(\gamma+1)\beta}}{nm} \cdot t\right), \tag{C.34}$$

where $c_6 := 1 + \gamma^{-2}c_3^{-1}R_{\min}^{-2}pn$.

In order to apply Lemma H.4 (requiring $b > a$), we loosen the bounds in (C.31), (C.32), (C.33) and (C.34) as follows:

$$\frac{1}{m} \sum_{r=1}^{m} \rho_{y_i,r,i}^{(t)} \leq c_4^{-1}\gamma^{-1} \log\left(1 + \frac{\gamma\eta R_{\max}^2 c_5 e^{2\beta}}{nm} \cdot t\right), \forall 0 \leq t \leq \widetilde{t}, \tag{C.35}$$

$$\frac{1}{m} \sum_{r=1}^{m} |\rho_{-y_i,r,i}^{(t)}| \leq c_4^{-1}\gamma^{-1} \log\left(1 + \frac{\gamma\eta R_{\max}^2 c_5 e^{2\beta}}{nm} \cdot t\right), \forall 0 \leq t \leq \widetilde{t}, \tag{C.36}$$

$$\frac{1}{m} \sum_{r=1}^{m} \rho_{y_i,r,i}^{(t)} \geq (2c_6)^{-1}\gamma \log\left(1 + \frac{\eta R_{\min}^2 c_6 e^{-(\gamma+1)\beta}}{nm} \cdot t\right), \forall 0 \leq t \leq \widetilde{t}, \tag{C.37}$$

$$\frac{1}{m} \sum_{r=1}^{m} |\rho_{-y_i,r,i}^{(t)}| \geq (2c_6)^{-1}\gamma \log\left(1 + \frac{\eta R_{\min}^2 c_6 e^{-(\gamma+1)\beta}}{nm} \cdot t\right), \forall 0 \leq t \leq \widetilde{t}, \tag{C.38}$$

where (C.35) is by Bernoulli's inequality that $1+\gamma^{-1}x \leq (1+x)^{\gamma^{-1}}$ for every real number $0 \leq r \leq 1$ and $x \geq -1$; (C.37) is by Bernoulli's inequality that $1 + \gamma x \geq (1+x)^\gamma$ for every real number $0 \leq r \leq 1$ and $x \geq -1$. If $R_{\min}^2 c_6 e^{-(\gamma+1)\beta} \geq \gamma R_{\max}^2 c_5 e^{2\beta}$, we have

$$\frac{1}{m} \sum_{r=1}^{m} |\rho_{j,r,i}^{(t)}| \geq \frac{\gamma^2(2c_6)^{-1}c_4}{m} \sum_{r=1}^{m} |\rho_{j',r,i'}^{(t)}|. \tag{C.39}$$

If $R_{\min}^2 c_6 e^{-(\gamma+1)\beta} < \gamma R_{\max}^2 c_5 e^{2\beta}$, by Lemma H.4, we have

$$\frac{\min\{\frac{1}{m}\sum_{r=1}^{m}\rho_{y_i,r,i}^{(t)}, \frac{1}{m}\sum_{r=1}^{m}|\rho_{-y_i,r,i}^{(t)}|\}}{\max\{\frac{1}{m}\sum_{r=1}^{m}\rho_{y_{i'},r,i'}^{(t)}, \frac{1}{m}\sum_{r=1}^{m}|\rho_{-y_{i'},r,i'}^{(t)}|\}}$$

$$\geq \gamma^2(2c_6)^{-1}c_4 \cdot \frac{\log\left(1 + \frac{\eta R_{\min}^2 c_6 e^{-(\gamma+1)\beta}}{nm} \cdot t\right)}{\log\left(1 + \frac{\gamma\eta R_{\max}^2 c_5 e^{2\beta}}{nm} \cdot t\right)}$$

$$\geq \gamma^2(2c_6)^{-1}c_4 \cdot \frac{\log\left(1 + \frac{\eta R_{\min}^2 c_6 e^{-(\gamma+1)\beta}}{nm} \cdot T_1\right)}{\log\left(1 + \frac{\gamma\eta R_{\max}^2 c_5 e^{2\beta}}{nm} \cdot T_1\right)}$$

$$\geq \gamma^2(2c_6)^{-1}c_4 \cdot \frac{\log(1 + R^{-2}c_6 e^{-(\gamma+1)\beta}C')}{\log(1 + \gamma c_5 e^{2\beta}C')}.$$

Therefore, we can get for $0 \leq t \leq \widetilde{t}$ that

$$\sum_{r=1}^{m} |\rho_{j,r,i}^{(t)}| \geq \gamma^2 c_3 \sum_{r=1}^{m} |\rho_{j',r,i'}^{(t)}|, \forall j, j' \in \{\pm 1\}, \forall i, i' \in [n], \tag{C.40}$$

as long as

$$c_3 \leq (2c_6)^{-1}c_4 \cdot \min\left\{1, \frac{\log(1 + R^{-2}c_6 e^{-(\gamma+1)\beta}C')}{\log(1 + \gamma c_5 e^{2\beta}C')}\right\}.$$

This condition holds under the following conditions:

$$\gamma^{-3}c_3^{-1}R_{\min}^{-2}pn \le \frac{1}{2} \quad \Longrightarrow \quad c_4, c_5 \ge \frac{1}{2}, c_6 \le \frac{3}{2},$$

$$c_3 = \frac{1}{6}\min\left\{1, \frac{\log(1 + R^{-2}e^{-(\gamma+1)\beta}C')}{\log(1 + \gamma e^{2\beta}C')}\right\}.$$

This implies that induction hypothesis (C.15) holds for $t = \tilde{t}$. $\qquad\square$

**Lemma C.4** (Implication of Lemma C.1). Under the same condition as Theorem 4.1, if (C.8) hold for time $t$, then we have that

$$|\rho_{j,r,i}^{(t)}| \ge c_1\gamma^4|\rho_{j',r',i'}^{(t)}|,$$

where $c_1$ is the same constant as defined in Lemma C.1.

*Proof of Lemma C.4.* By $\sigma' \in [\gamma, 1]$, (5.4) and (5.5), we have

$$|\rho_{j,r,i}^{(t+1)}| \ge |\rho_{j,r,i}^{(t)}| + \frac{\gamma\eta}{nm}\cdot|\ell_i'^{(t)}|\cdot\|\mathbf{x}_i\|_2^2, \forall j \in \{\pm1\}, \forall r \in [m], \forall i \in [n],$$

$$|\rho_{j,r,i}^{(t+1)}| \le |\rho_{j,r,i}^{(t)}| + \frac{\eta}{nm}\cdot|\ell_i'^{(t)}|\cdot\|\mathbf{x}_i\|_2^2, \forall j \in \{\pm1\}, \forall r \in [m], \forall i \in [n].$$

Thus, we have

$$|\rho_{j,r,i}^{(t)}| \ge \frac{\gamma\eta\|\mathbf{x}_i\|_2^2}{nm}\cdot\sum_{s=1}^{t-1}|\ell_i'^{(t)}|, \forall j \in \{\pm1\}, \forall r \in [m], \forall i \in [n],$$

$$|\rho_{j,r,i}^{(t)}| \le \frac{\eta\|\mathbf{x}_i\|_2^2}{nm}\cdot\sum_{s=1}^{t-1}|\ell_i'^{(t)}|, \forall j \in \{\pm1\}, \forall r \in [m], \forall i \in [n].$$

Therefore, $|\rho_{j,r,i}^{(t)}| \ge \gamma|\rho_{j',r',i}^{(t)}|$ for any $j, j' \in \{\pm1\}$, $r', r \in [m]$ and $i \in [n]$, and hence

$$m|\rho_{j,r,i}^{(t)}| \ge \gamma\sum_{r=1}^{m}|\rho_{j,r,i}^{(t)}|,$$

$$\sum_{r=1}^{m}|\rho_{j',r,i'}^{(t)}| \ge m\gamma|\rho_{j',r',i'}^{(t)}|. \tag{C.41}$$

Plugging (C.41) back into (C.8) completes the proof. $\qquad\square$

**Lemma C.5.** Let $T_1$ be defined in Lemma C.2. Every neuron will never change its activation pattern after time $T_1$, i.e.,

$$\mathrm{sign}(\langle\mathbf{w}_{j,r}^{(t)}, \mathbf{x}_i\rangle) = \mathrm{sign}(\langle\mathbf{w}_{j,r}^{(T_1)}, \mathbf{x}_i\rangle),$$

for any $t \ge T_1$, $j \in \{\pm1\}$ and $r \in [m]$. Moreover, it holds that

$$\mathrm{sign}(\langle\mathbf{w}_{j,r}^{(t)}, \mathbf{x}_i\rangle) = jy_i, \tag{C.42}$$

for any $t \ge T_1$, $j \in \{\pm1\}$ and $r \in [m]$.

*Proof of Lemma C.5.* For $j = y_i$ and $t \ge 0$, we have $\rho_{-j,r,i}^{(t)} = 0$, and so

$$\langle\mathbf{w}_{j,r}^{(t)}, \mathbf{x}_i\rangle = \langle\mathbf{w}_{j,r}^{(0)}, \mathbf{x}_i\rangle + \sum_{i'=1}^{n}\rho_{j,r,i'}^{(t)}\|\mathbf{x}_{i'}\|_2^{-2}\cdot\langle\mathbf{x}_{i'}, \mathbf{x}_i\rangle$$

$$= \langle\mathbf{w}_{j,r}^{(0)}, \mathbf{x}_i\rangle + \overline{\rho}_{j,r,i}^{(t)} + \sum_{i'\neq i}\rho_{j,r,i'}^{(t)}\|\mathbf{x}_{i'}\|_2^{-2}\cdot\langle\mathbf{x}_{i'}, \mathbf{x}_i\rangle$$

$$\ge \overline{\rho}_{j,r,i}^{(t)} - \sum_{i'\neq i}|\rho_{j,r,i'}^{(t)}|R_{\min}^{-2}p - \beta$$

$$\geq \overline{\rho}_{j,r,i}^{(t)} - \gamma^{-4}c_1^{-1}\overline{\rho}_{j,r,i}^{(t)}R_{\min}^{-2}pn - \beta$$
$$= (1 - \gamma^{-4}c_1^{-1}R_{\min}^{-2}pn) \cdot \overline{\rho}_{j,r,i}^{(t)} - \beta,$$

where the first inequality is by triangle inequality; the second inequality is by $|\rho_{j,r,i'}^{(t)}| \leq \gamma^{-4}c_1^{-1}\overline{\rho}_{y_i,r,i}^{(t)}$ from Lemma C.1 and Lemma C.4.

By (C.13), we have for $t \geq T_1$ that

$$\overline{\rho}_{y_i,r,i}^{(t)} \geq \frac{\widetilde{c}\gamma\eta R_{\min}^2 T_1}{nm} = C'\widetilde{c}\gamma R_{\min}^2 R_{\max}^{-2}. \tag{C.43}$$

Therefore, by (C.5), (C.6) and (C.43), we know that

$$(1 - \gamma^{-1}c_1^{-4}R_{\min}^{-2}pn) \cdot \overline{\rho}_{y_i,r,i}^{(t)} > \beta, \forall\, r \in [m], i \in [n].$$

and thus $\text{sign}(\langle \mathbf{w}_{j,r}^{(t)}, \mathbf{x}_i \rangle) = 1$ for any $r \in [m], i \in [n], j = y_i$.

For $j \neq y_i$ and any $t \geq 0$, we have $\overline{\rho}_{j,r,i}^{(t)} = 0$, and so

$$\langle \mathbf{w}_{j,r}^{(t)}, \mathbf{x}_i \rangle = \langle \mathbf{w}_{j,r}^{(0)}, \mathbf{x}_i \rangle + \sum_{i'=1}^n \rho_{j,r,i'}^{(t)}\|\mathbf{x}_{i'}\|_2^{-2} \cdot \langle \mathbf{x}_{i'}, \mathbf{x}_i \rangle$$

$$= \underline{\rho}_{j,r,i}^{(t)} + \sum_{i' \neq i} \rho_{j,r,i'}^{(t)}\|\mathbf{x}_{i'}\|_2^{-2} \cdot \langle \mathbf{x}_{i'}, \mathbf{x}_i \rangle$$

$$\leq \underline{\rho}_{j,r,i}^{(t)} + \sum_{i' \neq i} |\rho_{j,r,i'}^{(t)}|R_{\min}^{-2}p + \beta$$

$$\leq \underline{\rho}_{j,r,i}^{(t)} - \gamma^{-1}c_2^{-1}\underline{\rho}_{j,r,i}^{(t)}R_{\min}^{-2}pn - \beta$$

$$= (1 - \gamma^{-1}c_2^{-1}R_{\min}^{-2}pn)\underline{\rho}_{j,r,i}^{(t)} - \beta,$$

where the first inequality is by triangle inequality; the second inequality is by $|\rho_{j,r,i'}^{(t)}| \leq \gamma^{-4}c_1^{-1}|\underline{\rho}_{-y_i,r,i}^{(t)}|$ from Lemma C.1 and Lemma C.4.

By (C.13), we have

$$|\underline{\rho}_{-y_i,r,i}^{(t)}| \geq \frac{\widetilde{c}\gamma\eta R_{\min}^2 T_1}{nm} = C'\widetilde{c}\gamma R_{\min}^2 R_{\max}^{-2}. \tag{C.44}$$

Therefore, by (C.5), (C.6) and (C.44), we know that

$$(1 - \gamma^{-4}c_1^{-1}R_{\min}^{-2}pn) \cdot |\underline{\rho}_{-y_i,r,i}^{(t)}| > \beta, \forall\, r \in [m], i \in [n],$$

and thus $\text{sign}(\langle \mathbf{w}_{j,r}^{(t)}, \mathbf{x}_i \rangle) = -1$ for $j \neq y_i$, which completes the proof. $\qquad\square$

# D  Stable Rank of Leaky ReLU Network

In this section, we consider the properties of stable rank of the weight matrix $\mathbf{W}^{(t)}$ found by gradient descent at time $t$, defined as $\|\mathbf{W}^{(t)}\|_F^2/\|\mathbf{W}^{(t)}\|_2^2$, the square of the ratio of the Frobenius norm to the spectral norm of $\mathbf{W}^{(t)}$. Given Lemma C.5, we have following coefficient update rule for $t \geq T_1$ where $T_1$ is defined in Lemma C.2:

$$\overline{\rho}_{y_i,r,i}^{(t+1)} = \overline{\rho}_{y_i,r,i}^{(t)} + \frac{\eta}{nm} \cdot |\ell_i'^{(t)}| \cdot \|\mathbf{x}_i\|_2^2, \tag{D.1}$$

$$\underline{\rho}_{-y_i,r,i}^{(t+1)} = \underline{\rho}_{-y_i,r,i}^{(t)} - \frac{\gamma\eta}{nm} \cdot |\ell_i'^{(t)}| \cdot \|\mathbf{x}_i\|_2^2, \tag{D.2}$$

where

$$|\ell_i'^{(t)}| = \frac{1}{1 + \exp\{F_{y_i}(\mathbf{W}_{y_i}^{(t)}, \mathbf{x}_i) - F_{-y_i}(\mathbf{W}_{-y_i}^{(t)}, \mathbf{x}_i)\}}.$$

Based on (D.1) and (D.2), we first introduce the following helpful lemmas.

**Lemma D.1.** Let $T_1$ be defined in Lemma C.3. For any $r, r' \in [m]$, $i \in [n]$ and $t \leq T_1$,

$$|\overline{\rho}_{y_i,r,i}^{(t)} - \overline{\rho}_{y_i,r',i}^{(t)}| \leq C', |\underline{\rho}_{-y_i,r,i}^{(t)} - \underline{\rho}_{-y_i,r',i}^{(t)}| \leq C'. \tag{D.3}$$

*Proof of Lemma D.1.* By (C.14), we can get

$$|\rho_{j,r,i}^{(t)}| \leq \frac{\eta R_{\max}^2 T_1}{nm} = C',$$

for $t \leq T_1$. Notice that

$$|\overline{\rho}_{y_i,r,i}^{(t)} - \overline{\rho}_{y_i,r',i}^{(t)}| \leq \max\{|\overline{\rho}_{y_i,r,i}^{(t)}|, |\overline{\rho}_{y_i,r',i}^{(t)}|\},$$
$$|\underline{\rho}_{-y_i,r,i}^{(t)} - \underline{\rho}_{-y_i,r',i}^{(t)}| \leq \max\{|\underline{\rho}_{-y_i,r,i}^{(t)}|, |\underline{\rho}_{-y_i,r',i}^{(t)}|\},$$

which completes the proof. $\qquad\square$

**Lemma D.2.** Let $T_1$ be defined in Lemma C.3. For any $r, r' \in [m]$, $i \in [n]$ and $t \geq T_1$,

$$|\overline{\rho}_{y_i,r,i}^{(t)} - \overline{\rho}_{y_i,r',i}^{(t)}| \leq C', |\underline{\rho}_{-y_i,r,i}^{(t)} - \underline{\rho}_{-y_i,r',i}^{(t)}| \leq C'.$$

*Proof of Lemma D.2.* By (D.1) and (D.2), we can get for any $r \in [m]$, $i \in [n]$ and $t \geq T_1$ that

$$\overline{\rho}_{y_i,r,i}^{(t)} = \overline{\rho}_{y_i,r,i}^{(T_1)} + \frac{\eta}{nm} \sum_{s=T_1}^{t-1} |\ell_i'^{(t)}| \cdot \|\mathbf{x}_i\|_2^2,$$

$$\underline{\rho}_{-y_i,r,i}^{(t)} = \underline{\rho}_{-y_i,r,i}^{(T_1)} + \frac{\eta}{nm} \sum_{s=T_1}^{t-1} |\ell_i'^{(t)}| \cdot \|\mathbf{x}_i\|_2^2.$$

Since $\overline{\rho}_{y_i,r,i}^{(t)}, \overline{\rho}_{y_i,r',i}^{(t)}$ possess the same increment and $\underline{\rho}_{-y_i,r,i}^{(t)}, \underline{\rho}_{-y_i,r',i}^{(t)}$ possess the same increment, we have

$$\overline{\rho}_{y_i,r,i}^{(t)} - \overline{\rho}_{y_i,r',i}^{(t)} = \overline{\rho}_{y_i,r,i}^{(T_1)} - \overline{\rho}_{y_i,r',i}^{(T_1)},$$
$$\underline{\rho}_{-y_i,r,i}^{(t)} - \underline{\rho}_{-y_i,r',i}^{(t)} = \underline{\rho}_{-y_i,r,i}^{(T_1)} - \underline{\rho}_{-y_i,r',i}^{(T_1)}.$$

Notice that

$$\max_{i,r,r'}\{|\overline{\rho}_{y_i,r,i}^{(T_1)} - \overline{\rho}_{y_i,r',i}^{(T_1)}|, |\underline{\rho}_{-y_i,r,i}^{(T_1)} - \underline{\rho}_{-y_i,r',i}^{(T_1)}|\} \leq \max_{i,r,r'}\{|\overline{\rho}_{y_i,r,i}^{(T_1)}|, |\underline{\rho}_{-y_i,r,i}^{(T_1)}|\} \leq C',$$

which completes the proof. $\qquad\square$

**Lemma D.3.** Let $T_1$ be defined in Lemma C.3. For $t \geq T_1$, it holds that

$$\overline{\rho}_{y_i,r,i}^{(t)} - \overline{\rho}_{y_i,r,i}^{(T_1)} = \overline{\rho}_{y_i,r',i}^{(t)} - \overline{\rho}_{y_i,r',i}^{(T_1)},$$
$$\underline{\rho}_{-y_i,r,i}^{(t)} - \underline{\rho}_{-y_i,r,i}^{(T_1)} = \underline{\rho}_{-y_i,r',i}^{(t)} - \underline{\rho}_{-y_i,r',i}^{(T_1)},$$
$$\overline{\rho}_{y_i,r,i}^{(t)} - \overline{\rho}_{y_i,r,i}^{(T_1)} = \left(|\underline{\rho}_{-y_i,r',i}^{(t)}| - |\underline{\rho}_{-y_i,r',i}^{(T_1)}|\right)/\gamma,$$

for any $i \in [n]$ and $r, r' \in [m]$.

*Proof of Lemma D.3.* By Lemma C.3 about the activation pattern after time $T_1$, we can get

$$\overline{\rho}_{y_i,r,i}^{(t+1)} = \overline{\rho}_{y_i,r,i}^{(t)} + \frac{\eta}{nm} \cdot |\ell_i'^{(t)}| \cdot \|\mathbf{x}_i\|_2^2, \tag{D.4}$$

$$\underline{\rho}_{-y_i,r,i}^{(t+1)} = \underline{\rho}_{-y_i,r,i}^{(t)} - \frac{\gamma\eta}{nm} \cdot |\ell_i'^{(t)}| \cdot \|\mathbf{x}_i\|_2^2, \tag{D.5}$$

for $t \geq T_1$. Recursively using (D.4) and (D.5) $t - T_1$ times, we can get

$$\overline{\rho}_{y_i,r,i}^{(t)} - \overline{\rho}_{y_i,r,i}^{(T_1)} = \frac{\eta\|\mathbf{x}_i\|_2^2}{nm} \sum_{s=T_1}^{t-1} |\ell_i'^{(s)}|,$$

$$|\underline{\rho}^{(t)}_{-y_i,r,i}| - |\underline{\rho}^{(T_1)}_{-y_i,r,i}| = \frac{\gamma\eta\|\mathbf{x}_i\|_2^2}{nm} \sum_{s=T_1}^{t-1} |\ell_i^{\prime(s)}|.$$

This indicates that for different $r, r' \in [m]$, $\overline{\rho}^{(t)}_{y_i,r,i} - \overline{\rho}^{(T_1)}_{y_i,r,i}$ and $\overline{\rho}^{(t)}_{y_i,r',i} - \overline{\rho}^{(T_1)}_{y_i,r',i}$ are the same, whereas $\gamma(|\underline{\rho}^{(t)}_{-y_i,r,i}| - |\underline{\rho}^{(T_1)}_{-y_i,r,i}|)$ and $\overline{\rho}^{(t)}_{y_i,r',i} - \overline{\rho}^{(T_1)}_{y_i,r',i}$ are the same, which completes the proof. $\qquad\square$

Now we are ready to prove the second bullet of Theorem 4.1.

**Lemma D.4.** Throughout the gradient descent trajectory, the stable rank of the weights $\mathbf{W}_j$ satisfies,

$$\lim_{t\to\infty} \frac{\|\mathbf{W}_j\|_F^2}{\|\mathbf{W}_j\|_2^2} = 1, \forall j \in \{\pm 1\},$$

with a decreasing rate of $O\big(1/\log(t)\big)$.

*Proof of Lemma D.4.* By Definition 5.1, we have

$$\mathbf{w}^{(t)}_{j,r} = \mathbf{w}^{(0)}_{j,r} + \underbrace{\sum_{i=1}^{n} \rho^{(t)}_{j,r,i} \cdot \|\mathbf{x}_i\|_2^{-2} \cdot \mathbf{x}_i}_{:=\mathbf{v}^{(t)}_{j,r}}.$$

We first show that $\|\mathbf{v}^{(t)}_{j,r}\|_2 = \Theta(\log t)$.

$$\|\mathbf{v}^{(t)}_{j,r}\|_2^2 = \left( \sum_{i=1}^{n} \rho^{(t)}_{j,r,i} \cdot \|\mathbf{x}_i\|_2^{-2} \cdot \mathbf{x}_i \right)^2$$

$$= \sum_{i=1}^{n} (\rho^{(t)}_{j,r,i})^2 \cdot \|\mathbf{x}_i\|_2^{-2} + \sum_{i\neq i'} \rho^{(t)}_{j,r,i}\rho^{(t)}_{j,r,i'} \|\mathbf{x}_i\|_2^{-2}\|\mathbf{x}_{i'}\|_2^{-2}\langle\mathbf{x}_i, \mathbf{x}_{i'}\rangle$$

$$\geq \sum_{i=1}^{n} (\rho^{(t)}_{j,r,i})^2 \cdot R_{\max}^{-2} - \sum_{i\neq i'} |\rho^{(t)}_{j,r,i}||\rho^{(t)}_{j,r,i'}| \cdot R_{\min}^{-4}p$$

$$\geq R_{\max}^{-2}(1 - R^2 R_{\min}^{-2}c_1^{-1}\gamma^{-4}np) \sum_{i=1}^{n} (\rho^{(t)}_{j,r,i})^2$$

$$= \Theta(nR_{\max}^{-2}\log^2(t)),$$

where the second last inequality is by triangle inequality; the last inequality is by $|\rho^{(t)}_{j,r,i}| \leq \gamma^{-4}c_1^{-1}|\rho^{(t)}_{j,r,i'}|$ from Lemma C.1 and Lemma C.4.
By the definition of $\mathbf{v}^{(t)}_{j,r}$, we have

$$\|\mathbf{v}^{(t)}_{j,r} - \mathbf{v}^{(t)}_{j,r'}\|_2^2 = \left\| \sum_{i=1}^{n} \rho^{(t)}_{j,r,i} \cdot \|\mathbf{x}_i\|_2^{-2} \cdot \mathbf{x}_i - \sum_{i=1}^{n} \rho^{(t)}_{j,r',i} \cdot \|\mathbf{x}_i\|_2^{-2} \cdot \mathbf{x}_i \right\|_2^2$$

$$= \left\| \sum_{i=1}^{n} (\rho^{(t)}_{j,r,i} - \rho^{(t)}_{j,r',i}) \cdot \|\mathbf{x}_i\|_2^{-2} \cdot \mathbf{x}_i \right\|_2^2$$

$$= \sum_{i=1}^{n} (\rho^{(t)}_{j,r,i} - \rho^{(t)}_{j,r',i})^2 \cdot \|\mathbf{x}_i\|_2^{-2} + \sum_{i\neq i'} (\rho^{(t)}_{j,r,i} - \rho^{(t)}_{j,r',i})(\rho^{(t)}_{j,r,i'} - \rho^{(t)}_{j,r',i'}) \frac{\langle\mathbf{x}_i, \mathbf{x}_{i'}\rangle}{\|\mathbf{x}_i\|_2^2\|\mathbf{x}_{i'}\|_2^2}$$

$$\leq (C')^2 nR_{\min}^{-2} + (C')^2 n^2 R_{\min}^{-4}p$$

$$\leq 2(C')^2 nR_{\min}^{-2},$$

where the first inequality is by Lemma D.1 and Lemma D.2.

Now, we are ready to estimate the stable rank of $\mathbf{W}^{(t)}$. On the one hand, for $\|\mathbf{W}_j^{(t)}\|_F^2$, we have

$$
\begin{aligned}
\|\mathbf{W}_j^{(t)}\|_F^2 &= \sum_r \|\mathbf{w}_{j,r}^{(t)}\|_2^2 \\
&= \sum_r \|\mathbf{w}_{j,r}^{(0)} + \mathbf{v}_{j,r}^{(t)}\|_2^2 \\
&= \sum_r \|\mathbf{w}_{j,r}^{(0)}\|_2^2 + \|\mathbf{v}_{j,r}^{(t)}\|_2^2 + 2\langle \mathbf{w}_{j,r}^{(0)}, \mathbf{v}_{j,r}^{(t)} \rangle \\
&\leq \sum_r \|\mathbf{w}_{j,r}^{(0)}\|_2^2 + (\|\mathbf{v}_{j,1}^{(t)}\|_2 + \|\mathbf{v}_{j,r}^{(t)} - \mathbf{v}_{j,1}^{(t)}\|_2)^2 + 2\|\mathbf{w}_{j,r}^{(0)}\|_2(\|\mathbf{v}_{j,1}^{(t)}\|_2 + \|\mathbf{v}_{j,r}^{(t)} - \mathbf{v}_{j,1}^{(t)}\|_2) \\
&= m\|\mathbf{v}_{j,1}^{(t)}\|_2^2 + 2\Big(\sum_r \|\mathbf{v}_{j,r}^{(t)} - \mathbf{v}_{j,1}^{(t)}\|_2 + \|\mathbf{w}_{j,r}^{(0)}\|_2\Big)\|\mathbf{v}_{j,1}^{(t)}\|_2 \\
&\quad + \Big(\sum_r \|\mathbf{w}_{j,r}^{(0)}\|_2^2 + \|\mathbf{v}_{j,r}^{(t)} - \mathbf{v}_{j,1}^{(t)}\|_2^2 + 2\|\mathbf{w}_{j,r}^{(0)}\|_2\|\mathbf{v}_{j,r}^{(t)} - \mathbf{v}_{j,1}^{(t)}\|_2\Big) \\
&\leq m\|\mathbf{v}_{j,1}^{(t)}\|_2^2 + mC_1\|\mathbf{v}_{j,1}^{(t)}\|_2 + mC_2.
\end{aligned}
$$

where the first inequality is by triangle inequality and Cauchy inequality; the last inequality is by Lemma D.1, Lemma D.2 and taking

$$
\begin{aligned}
C_1 &= 3(\sigma_0\sqrt{d} + C'\sqrt{n}R_{\min}^{-1}), \\
C_2 &= 2(\sigma_0\sqrt{d} + C'\sqrt{n}R_{\min}^{-1})^2.
\end{aligned}
$$

On the other hand, for $\|\mathbf{W}_j^{(t)}\|_2^2$, we have

$$
\begin{aligned}
\|\mathbf{W}_j^{(t)}\|_2^2 &= \max_{\mathbf{x} \in S^{d-1}} \|\mathbf{W}_j^{(0)}\mathbf{x} + \mathbf{V}_j^{(t)}\mathbf{x}\|_2^2 \\
&= \max_{\mathbf{x} \in S^{d-1}} \|\mathbf{W}_j^{(0)}\mathbf{x}\|_2^2 + \|\mathbf{V}_j^{(t)}\mathbf{x}\|_2^2 + 2\langle \mathbf{W}_j^{(0)}\mathbf{x}, \mathbf{V}_j^{(t)}\mathbf{x} \rangle \\
&= \max_{\mathbf{x} \in S^{d-1}} \sum_r \langle \mathbf{w}_{j,r}^{(0)}, \mathbf{x} \rangle^2 + \sum_r \langle \mathbf{v}_{j,r}^{(t)}, \mathbf{x} \rangle^2 + \sum_r \langle \mathbf{w}_{j,r}^{(0)}, \mathbf{x} \rangle \cdot \langle \mathbf{v}_{j,r}^{(t)}, \mathbf{x} \rangle \\
&\geq \sum_r \Big\langle \mathbf{w}_{j,r}^{(0)}, \frac{\mathbf{v}_{j,1}^{(t)}}{\|\mathbf{v}_{j,1}^{(t)}\|_2} \Big\rangle^2 + \sum_r \Big\langle \mathbf{v}_{j,r}^{(t)}, \frac{\mathbf{v}_{j,1}^{(t)}}{\|\mathbf{v}_{j,1}^{(t)}\|_2} \Big\rangle^2 \\
&\quad + \sum_r \Big\langle \mathbf{w}_{j,r}^{(0)}, \frac{\mathbf{v}_{j,1}^{(t)}}{\|\mathbf{v}_{j,1}^{(t)}\|_2} \Big\rangle \cdot \Big\langle \mathbf{v}_{j,r}^{(t)}, \frac{\mathbf{v}_{j,1}^{(t)}}{\|\mathbf{v}_{j,1}^{(t)}\|_2} \Big\rangle \\
&\geq \sum_r \Big\langle \mathbf{v}_{j,r}^{(t)}, \frac{\mathbf{v}_{j,1}^{(t)}}{\|\mathbf{v}_{j,1}^{(t)}\|_2} \Big\rangle^2 - \sum_r \|\mathbf{w}_{j,r}^{(0)}\|_2^2 - \sum_r \|\mathbf{w}_{j,r}^{(0)}\|_2\|\mathbf{v}_{j,r}^{(t)}\|_2 \\
&\geq m\|\mathbf{v}_{j,1}^{(t)}\|_2^2 + 2\sum_r \|\mathbf{v}_{j,1}^{(t)}\|_2 \cdot \Big\langle \mathbf{v}_{j,r}^{(t)} - \mathbf{v}_{j,1}^{(t)}, \frac{\mathbf{v}_{j,1}^{(t)}}{\|\mathbf{v}_{j,1}^{(t)}\|_2} \Big\rangle + \sum_r \Big\langle \mathbf{v}_{j,r}^{(t)} - \mathbf{v}_{j,1}^{(t)}, \frac{\mathbf{v}_{j,1}^{(t)}}{\|\mathbf{v}_{j,1}^{(t)}\|_2} \Big\rangle^2 \\
&\quad - \sum_r \|\mathbf{w}_{j,r}^{(0)}\|_2^2 - \sum_r \|\mathbf{w}_{j,r}^{(0)}\|_2\big(\|\mathbf{v}_{j,1}^{(t)}\|_2 + \|\mathbf{v}_{j,r}^{(t)} - \mathbf{v}_{j,1}^{(t)}\|_2\big) \\
&\geq m\|\mathbf{v}_{j,1}^{(t)}\|_2^2 - \Big(\sum_r 2\|\mathbf{v}_{j,r}^{(t)} - \mathbf{v}_{j,1}^{(t)}\|_2 + \|\mathbf{w}_{j,r}^{(0)}\|_2\Big) \cdot \|\mathbf{v}_{j,1}^{(t)}\|_2 \\
&\quad - \Big(\sum_r \|\mathbf{w}_{j,r}^{(0)}\|_2^2 + \|\mathbf{w}_{j,r}^{(0)}\|_2\|\mathbf{v}_{j,r}^{(t)} - \mathbf{v}_{j,1}^{(t)}\|_2\Big) \\
&\geq m\|\mathbf{v}_{j,1}^{(t)}\|_2^2 - mC_3\|\mathbf{v}_{j,1}^{(t)}\|_2 - mC_4
\end{aligned}
$$

where the first inequality is by taking $\mathbf{x} = \mathbf{v}_{j,1}^{(t)}/\|\mathbf{v}_{j,1}^{(t)}\|_2$; the second inequality is by Cauchy inequality; the third inequality by breaking $\mathbf{v}_{j,r}^{(t)}$ down into $\mathbf{v}_{j,1}^{(t)} + \mathbf{v}_{j,r}^{(t)} - \mathbf{v}_{j,1}^{(t)}$ and then expanding the

first term as well as applying triangle inequality to the last term; the fourth inequality is by Cauchy inequality; the last inequality is by Lemma D.1, Lemma D.2 and taking

$$C_3 = 1.5\sigma_0\sqrt{d} + 3C'\sqrt{n}R_{\min}^{-1},$$
$$C_4 = 1.5\sigma_0^2 d + 3C'\sigma_0\sqrt{d}\sqrt{n}R_{\min}^{-1}.$$

By leverage the upper bound of $\|\mathbf{W}_j^{(t)}\|_F^2$ as well as the lower bound of $\|\mathbf{W}_j^{(t)}\|_2^2$, we can get

$$\frac{\|\mathbf{W}_j^{(t)}\|_F^2}{\|\mathbf{W}_j^{(t)}\|_2^2} \leq \frac{\|\mathbf{v}_{j,1}^{(t)}\|_2^2 + C_1\|\mathbf{v}_{j,1}^{(t)}\|_2 + C_2}{\|\mathbf{v}_{j,1}^{(t)}\|_2^2 - C_3\|\mathbf{v}_{j,1}^{(t)}\|_2 - C_4}.$$

Since $\|\mathbf{W}_j^{(t)}\|_F^2/\|\mathbf{W}_j^{(t)}\|_2^2 \geq 1$, $\|\mathbf{v}_{j,1}^{(t)}\|_2 = \Theta(\log t)$ and $C_1, C_2, C_3, C_4$ are constants, it follow that

$$\lim_{t\to\infty} \frac{\|\mathbf{W}_j^{(t)}\|_F^2}{\|\mathbf{W}_j^{(t)}\|_2^2} = 1,$$

and

$$\begin{aligned}
\frac{\|\mathbf{W}_j^{(t)}\|_F^2}{\|\mathbf{W}_j^{(t)}\|_2^2} - 1 &\leq \frac{(C_1 + C_3)\|\mathbf{v}_{j,1}^{(t)}\|_2 + (C_2 + C_4)}{\|\mathbf{v}_{j,1}^{(t)}\|_2^2 - C_3\|\mathbf{v}_{j,1}^{(t)}\|_2 - C_4} \\
&\preceq \frac{C_1 + C_3}{\|\mathbf{v}_{j,1}^{(t)}\|_2} \\
&\leq \frac{6(C'\sqrt{n}R_{\min}^{-1} + \sigma_0\sqrt{d})}{\|\mathbf{v}_{j,1}^{(t)}\|_2} \\
&= \Theta\Big(\frac{\sqrt{n}R_{\min}^{-1} + \sigma_0\sqrt{d}}{\sqrt{n}R_{\max}^{-1}\log(t)}\Big) = \Theta\Big(\frac{1 + \sigma_0\sqrt{d/n}R_{\max}}{\log(t)}\Big),
\end{aligned}$$

which completes the proof. $\qquad\square$

# E Coefficient Analysis of ReLU

In this section, we discuss the stable rank of two-layer ReLU neural network, which is defined as

$$\begin{aligned}
f(\mathbf{W}, \mathbf{x}) &= F_{+1}(\mathbf{W}_{+1}, \mathbf{x}) - F_{+1}(\mathbf{W}_{+1}, \mathbf{x}), \\
F_j(\mathbf{W}_j, \mathbf{x}) &= \frac{1}{m}\sum_{r=1}^m \sigma(\langle\mathbf{w}_{j,r}, \mathbf{x}\rangle),
\end{aligned} \tag{E.1}$$

where $\sigma(z) = \max\{0, z\}$ is ReLU activation function.

These results are based on the conclusions in Section B, which hold with high probability. Denote by $\mathcal{E}'_{\mathrm{prelim}}$ the event that all the results in Section B hold (for a given $\delta$, we see $\mathbb{P}(\mathcal{E}'_{\mathrm{prelim}}) \geq 1 - 2\delta$ by a union bound). For simplicity and clarity, we state all the results in this and the following sections conditional on $\mathcal{E}'_{\mathrm{prelim}}$.

Denote $\beta = \max_{i,j,r}\{|\langle\mathbf{w}_{j,r}^{(0)}, \mathbf{x}_i\rangle|\}$, $R_{\max} = \max_{i\in[n]}\|\mathbf{x}_i\|_2$, $R_{\min} = \min_{i\in[n]}\|\mathbf{x}_i\|_2$, $p = \max_{i\neq k}|\langle\mathbf{x}_i, \mathbf{x}_k\rangle|$ and suppose $R = R_{\max}/R_{\min}$ is at most an absolute constant. Here we list the exact conditions for $\eta, \sigma_0, R_{\min}, R_{\max}, p$ required by the proofs in this section:

$$\sigma_0 \leq \big(CR_{\max}\sqrt{\log(mn/\delta)}\big)^{-1}, \tag{E.2}$$

$$\eta \leq (CR_{\max}^2/nm)^{-1}, \tag{E.3}$$

$$R_{\min}^2 \geq CR^2 np, \tag{E.4}$$

where $C$ is a large enough constant. By Lemma B.1, we can upper bound $\beta$ by $2\sqrt{\log(12mn/\delta)} \cdot \sigma_0 R_{\max}$. Then, by (C.2) and (C.4), it is straightforward to verify the following inequality:

$$\beta \leq c, \tag{E.5}$$

$$R_{\min}^{-2} np \leq c, \tag{E.6}$$

$$R_{\min}^{-2} R^2 np \leq c, \tag{E.7}$$

where $c$ is a sufficiently small constant.

We first introduce the following lemma which characterizes the increasing rate of coefficients $\rho_{j,r,i}^{(t)}$.

**Lemma E.1.** For two-layer ReLU neural network defined in (E.1), under the same condition as Theorem 4.3, the decomposition coefficients $\rho_{j,r,i}^{(t)}$ satisfy following properties:

- $\overline{\rho}_{y_i,r,i}^{(t)} \geq c_1 |\rho_{j,r',i'}^{(t)}|$ for any $r \in S_i^{(0)}$, $r' \in [m]$, $j \in \{\pm 1\}$ and $i, i' \in [n]$,

- $\overline{\rho}_{y_i,r,i}^{(t)} \geq c_2 \log\left(1 + \frac{\eta |S_i^{(0)}| \|\mathbf{x}_i\|_2^2 e^{-\beta}}{2nm^2} \cdot t\right)$ for any $r \in S_i^{(0)}$ and $i \in [n]$,

- $\overline{\rho}_{y_i,r,i}^{(t)} \leq c_3 \log\left(1 + \frac{2\eta |S_i^{(0)}| \|\mathbf{x}_i\|_2^2 e^{2\beta}}{nm^2} \cdot t\right)$ for any $r \in S_i^{(0)}$ and $i \in [n]$,

where $c_1, c_2, c_3$ are constants. And the following activation pattern is also observed: $S_i^{(0)} \subseteq S_i^{(t)}$ where $S_i^{(t)} := \{r \in [m] : \langle \mathbf{w}_{y_i,r}^{(t)}, \mathbf{x}_i \rangle \geq 0\}$, i.e., the on-diagonal neuron activated at initialization will remain activated throughout the training.

*Proof of Lemma E.1.* We first show that the first bullet and $S_i^{(0)} \subseteq S_i^{(t)}$ hold for $t \leq T_1 = C\eta^{-1}nmR_{\max}^{-2}$ where $C = \Theta(1)$ is a constant. Now we prove this by induction. When $t = 0$, the two hypotheses hold naturally. Suppose that there exists time $\widetilde{t} \leq T_1$ such that the two hypotheses hold for all time $t \leq \widetilde{t} - 1$. We aim to prove they also hold for $t = \widetilde{t}$. Recall from Lemma B.3 that

$$\overline{\rho}_{j,r,i}^{(t+1)} = \overline{\rho}_{j,r,i}^{(t)} - \frac{\eta}{nm} \cdot \ell_i'^{(t)} \cdot \sigma'(\langle \mathbf{w}_{j,r}^{(t)}, \mathbf{x}_i \rangle) \cdot \|\mathbf{x}_i\|_2^2 \cdot \mathbb{1}(y_i = j),$$

$$\underline{\rho}_{j,r,i}^{(t+1)} = \underline{\rho}_{j,r,i}^{(t)} + \frac{\eta}{nm} \cdot \ell_i'^{(t)} \cdot \sigma'(\langle \mathbf{w}_{j,r}^{(t)}, \mathbf{x}_i \rangle) \cdot \|\mathbf{x}_i\|_2^2 \cdot \mathbb{1}(y_i = -j),$$

we can get

$$\overline{\rho}_{j,r,i}^{(t+1)} \leq \overline{\rho}_{j,r,i}^{(t)} + \frac{\eta}{nm} \cdot \|\mathbf{x}_i\|_2^2 \leq \overline{\rho}_{j,r,i}^{(t)} + \frac{\eta R_{\max}^2}{nm}, \tag{E.8}$$

$$|\underline{\rho}_{j,r,i}^{(t+1)}| \leq |\underline{\rho}_{j,r,i}^{(t)}| + \frac{\eta}{nm} \cdot \|\mathbf{x}_i\|_2^2 \leq |\underline{\rho}_{j,r,i}^{(t)}| + \frac{\eta R_{\max}^2}{nm}. \tag{E.9}$$

Therefore, $\max_{j,r,i}\{\overline{\rho}_{j,r,i}^{(t)}, |\underline{\rho}_{j,r,i}^{(t)}|\} = O(1)$ for any $t \leq T_1$ and hence $\max_i\{F_{+1}(\mathbf{W}_{+1}^{(t)}, \mathbf{x}_i), F_{-1}(\mathbf{W}_{-1}^{(t)}, \mathbf{x}_i)\} = O(1)$ for any $t \leq T_1$. Thus there exists a positive constant $c$ such that $|\ell_i'^{(t)}| \geq c$ for any $t \leq T_1$. By induction hypothesis, we have $S_i^{(0)} \subseteq S_i^{(t)}$ for all $0 \leq t \leq \widetilde{t} - 1$ and hence $\sigma'(\langle \mathbf{w}_{y_i,r}^{(t)}, \mathbf{x}_i \rangle) = 1$ for all $0 \leq t \leq \widetilde{t} - 1$. And it follows that for $r \in S_i^{(0)}$

$$\overline{\rho}_{y_i,r,i}^{(t+1)} = \overline{\rho}_{y_i,r,i}^{(t)} + \frac{\eta}{nm} \cdot |\ell_i'^{(t)}| \cdot \|\mathbf{x}_i\|_2^2 \geq \overline{\rho}_{y_i,r,i}^{(t)} + \frac{c\eta}{nm} \cdot \|\mathbf{x}_i\|_2^2, \forall 0 \leq t \leq \widetilde{t} - 1,$$

$$\overline{\rho}_{y_i,r,i}^{(\widetilde{t})} \geq \frac{c\eta \widetilde{t}}{nm} \cdot \|\mathbf{x}_i\|_2^2 \geq \frac{c\eta R_{\min}^2 \widetilde{t}}{nm}. \tag{E.10}$$

On the other hand, by (E.8) and (E.9), we have

$$\overline{\rho}_{j,r',i'}^{(\widetilde{t})} \leq \frac{\eta R_{\max}^2 \widetilde{t}}{nm}, |\underline{\rho}_{j,r',i'}^{(\widetilde{t})}| \leq \frac{\eta R_{\max}^2 \widetilde{t}}{nm} \implies |\rho_{j,r',i'}^{(\widetilde{t})}| \leq \frac{\eta R_{\max}^2 \widetilde{t}}{nm}. \tag{E.11}$$

Dividing (E.10) by (E.11), we can get

$$\frac{\overline{\rho}_{y_i,r,i}^{(\widetilde{t})}}{|\rho_{j,r',i'}^{(\widetilde{t})}|} \geq \frac{cR_{\min}^2}{R_{\max}^2}, \forall\, r \in S_i^{(0)}, j \in \{\pm 1\}, i, i' \in [n],$$ (E.12)

which indicates that the first bullet holds for time $t = \widetilde{t}$ as long as $c_1 \leq (cR_{\min}^2)/R_{\max}^2$. For $r \in S_i^{(0)}$, we have

$$\langle \mathbf{w}_{y_i,r}^{(\widetilde{t})}, \mathbf{x}_i \rangle = \langle \mathbf{w}_{y_i,r}^{(0)}, \mathbf{x}_i \rangle + \sum_{i'=1}^{n} \rho_{y_i,r,i'}^{(\widetilde{t})} \|\mathbf{x}_{i'}\|_2^{-2} \cdot \langle \mathbf{x}_{i'}, \mathbf{x}_i \rangle$$

$$= \langle \mathbf{w}_{y_i,r}^{(0)}, \mathbf{x}_i \rangle + \overline{\rho}_{y_i,r,i}^{(\widetilde{t})} + \sum_{i' \neq i} \rho_{y_i,r,i'}^{(\widetilde{t})} \|\mathbf{x}_{i'}\|_2^{-2} \cdot \langle \mathbf{x}_{i'}, \mathbf{x}_i \rangle$$

$$\geq \overline{\rho}_{y_i,r,i}^{(\widetilde{t})} - \sum_{i' \neq i} |\rho_{y_i,r,i'}^{(\widetilde{t})}| R_{\min}^{-2} p$$

$$\geq \overline{\rho}_{y_i,r,i}^{(\widetilde{t})} - \sum_{i' \neq i} \frac{R_{\max}^2}{cR_{\min}^2} \overline{\rho}_{y_i,r,i}^{(\widetilde{t})} \cdot R_{\min}^{-2} p$$

$$\geq \left(1 - \frac{R_{\max}^2}{cR_{\min}^4} pn\right) \cdot \overline{\rho}_{y_i,r,i}^{(\widetilde{t})} \geq 0,$$

where the second inequality is by (E.12). This implies that $S_i^{(0)} \subseteq S_i^{(t)}$ holds for time $t = \widetilde{t}$, which completes the induction. By then, we have already proved that the first bullet and $S_i^{(0)} \subseteq S_i^{(t)}$ hold for $t \leq T_1 = C\eta^{-1} nm R_{\max}^{-2}$.

Next, we will prove by induction that the three bullets as well as $S_i^{(0)} \subseteq S_i^{(t)}$ hold for any time $t \geq 0$. The second and third bullets are obvious at $t = 0$ as all the coefficients are zero. Suppose there exists $\widetilde{t}$ such that the three bullets as well as $S_i^{(0)} \subseteq S_i^{(t)}$ hold for all time $0 \leq t \leq \widetilde{t} - 1$. We aim to prove that they also hold for $t = \widetilde{t}$. We first prove that the second and third bullets hold for $t = \widetilde{t}$. To prove this, we first provide more precise upper and lower bounds for $|\ell_i'^{(t)}|$. For lower bound, we have

$$|\ell_i'^{(t)}| = \frac{1}{1 + \exp\left\{F_{y_i}(\mathbf{W}_{y_i}^{(t)}, \mathbf{x}_i) - F_{-y_i}(\mathbf{W}_{-y_i}^{(t)}, \mathbf{x}_i)\right\}}$$

$$\geq \frac{1}{1 + \exp\left\{F_{y_i}(\mathbf{W}_{y_i}^{(t)}, \mathbf{x}_i)\right\}}$$

$$= \frac{1}{1 + \exp\left\{\frac{1}{m} \sum_{r \in S_i^{(t)}} \langle \mathbf{w}_{y_i,r}^{(t)}, \mathbf{x}_i \rangle\right\}}$$ (E.13)

and

$$\sum_{r \in S_i^{(t)}} \langle \mathbf{w}_{y_i,r}^{(t)}, \mathbf{x}_i \rangle = \sum_{r \in S_i^{(t)}} \left( \langle \mathbf{w}_{y_i,r}^{(0)}, \mathbf{x}_i \rangle + \overline{\rho}_{y_i,r,i}^{(t)} + \sum_{i' \neq i} \rho_{y_i,r,i'}^{(t)} \|\mathbf{x}_{i'}\|_2^{-2} \cdot \langle \mathbf{x}_{i'}, \mathbf{x}_i \rangle \right)$$

$$\leq \sum_{r \in S_i^{(t)}} \overline{\rho}_{y_i,r,i}^{(t)} + \sum_{r \in S_i^{(t)}} \sum_{i' \neq i} |\rho_{y_i,r,i'}^{(t)}| R_{\min}^{-2} p + |S_i^{(t)}| \cdot \beta$$

$$\leq \sum_{r \in S_i^{(t)}} \overline{\rho}_{y_i,r,i}^{(t)} + \frac{|S_i^{(t)}|}{c_1 |S_i^{(0)}|} \sum_{r \in S_i^{(0)}} \overline{\rho}_{y_i,r,i'}^{(t)} R_{\min}^{-2} pn + |S_i^{(t)}| \cdot \beta$$

$$\leq \frac{|S_i^{(t)}|}{|S_i^{(0)}|} \sum_{r \in S_i^{(0)}} \overline{\rho}_{y_i,r,i}^{(t)} + \frac{|S_i^{(t)}|}{c_1 |S_i^{(0)}|} \sum_{r \in S_i^{(0)}} \overline{\rho}_{y_i,r,i'}^{(t)} R_{\min}^{-2} pn + |S_i^{(t)}| \cdot \beta$$

$$\leq c'(1 + R_{\min}^{-2} pn/c_1) \sum_{r \in S_i^{(0)}} \overline{\rho}_{y_i,r,i}^{(t)} + |S_i^{(t)}| \cdot \beta,$$ (E.14)

where the first inequality is by triangle inequality; the second inequality is by the first induction hypothesis that $\overline{\rho}_{y_i,r,i}^{(t)} \geq c_1 |\rho_{y_i,r',i'}^{(t)}|$ for $r \in S_i^{(0)}$ and $0 \leq t \leq \tilde{t}-1$ and hence $|\rho_{y_i,r',i'}^{(t)}| \leq \frac{1}{c_1 |S_i^{(0)}|} \sum_{r \in S_i^{(0)}} \overline{\rho}_{y_i,r,i}^{(t)}$; the third inequality is by

$$\overline{\rho}_{y_i,r',i}^{(t)} = \frac{\eta}{nm} \sum_{s=0}^{t-1} |\ell_i'^{(s)}| \cdot \sigma'(\langle \mathbf{w}_{y_i,r'}^{(s)}, \mathbf{x}_i \rangle) \cdot \|\mathbf{x}_i\|_2^2 \leq \frac{\eta}{nm} \sum_{s=0}^{t-1} |\ell_i'^{(s)}| \cdot \|\mathbf{x}_i\|_2^2,$$

$$\overline{\rho}_{y_i,r,i}^{(t)} = \frac{\eta}{nm} \sum_{s=0}^{t-1} |\ell_i'^{(s)}| \cdot \|\mathbf{x}_i\|_2^2,$$

and hence $\overline{\rho}_{y_i,r',i}^{(t)} \leq \overline{\rho}_{y_i,r,i}^{(t)}, \forall r' \in S_i^{(t)} \setminus S_i^{(0)}, r \in S_i^{(0)}$ for $0 \leq t \leq \tilde{t}-1$; the last inequality is by $|S_i^{(t)}| \leq m \leq c'|S_i^{(0)}|$ and $c'$ can be taken as 2.5 by Lemma B.2. By plugging (E.14) back into (E.13), we can get

$$
\begin{aligned}
|\ell_i^{(t)}| &\geq \frac{1}{1 + \exp\left\{\frac{c'(1+R_{\min}^{-2}pn/c_1)}{m} \sum_{r \in S_i^{(0)}} \overline{\rho}_{y_i,r,i}^{(t)} + \frac{|S_i^{(t)}|}{m} \cdot \beta\right\}} \\
&\geq \frac{1}{1 + \exp\left\{\frac{c'(1+R_{\min}^{-2}pn/c_1)}{m} \sum_{r \in S_i^{(0)}} \overline{\rho}_{y_i,r,i}^{(t)} + \beta\right\}} \\
&\geq \frac{1}{2} \exp\left\{-\frac{c'(1+R_{\min}^{-2}pn/c_1)}{m} \sum_{r \in S_i^{(0)}} \overline{\rho}_{y_i,r,i}^{(t)} - \beta\right\}, \forall 0 \leq t \leq \tilde{t}-1.
\end{aligned}
\tag{E.15}
$$

For upper bound of $|\ell_i'^{(t)}|$, we first bound $F_{y_i}(\mathbf{W}_{y_i}^{(t)}, \mathbf{x}_i) - F_{-y_i}(\mathbf{W}_{-y_i}^{(t)}, \mathbf{x}_i)$ as follows:

$$
\begin{aligned}
&F_{y_i}(\mathbf{W}_{y_i}^{(t)}, \mathbf{x}_i) - F_{-y_i}(\mathbf{W}_{-y_i}^{(t)}, \mathbf{x}_i) \\
&\geq \frac{1}{m}\left(\sum_{r \in S_i^{(t)}} \overline{\rho}_{y_i,r,i}^{(t)} - \sum_{r \in S_i^{(t)}} \sum_{i' \neq i} |\rho_{y_i,r,i'}^{(t)}| R_{\min}^{-2} p - |S_i^{(t)}| \cdot \beta\right) - \frac{1}{m} \sum_{r=1}^{m}\left(\beta + \sum_{i' \neq i} |\rho_{-y_i,r,i'}^{(t)}| R_{\min}^{-2} p\right) \\
&\geq \frac{1}{m} \sum_{r \in S_i^{(t)}} \overline{\rho}_{y_i,r,i}^{(t)} - \frac{|S_i^{(t)}|}{c_1 m |S_i^{(0)}|} \sum_{r \in S_i^{(0)}} \overline{\rho}_{y_i,r,i}^{(t)} R_{\min}^{-2} pn - \frac{1}{c_1 |S_i^{(0)}|} \sum_{r \in S_i^{(0)}} \overline{\rho}_{y_i,r,i}^{(t)} R_{\min}^{-2} pn - 2\beta \\
&\geq \frac{1}{m} \sum_{r \in S_i^{(t)}} \overline{\rho}_{y_i,r,i}^{(t)} - \frac{2}{c_1 |S_i^{(0)}|} \sum_{r \in S_i^{(0)}} \overline{\rho}_{y_i,r,i}^{(t)} R_{\min}^{-2} pn - 2\beta \\
&\geq \frac{1}{m} \sum_{r \in S_i^{(0)}} \overline{\rho}_{y_i,r,i}^{(t)} - \frac{2c' R_{\min}^{-2} pn}{c_1 m} \sum_{r \in S_i^{(0)}} \overline{\rho}_{y_i,r,i}^{(t)} - 2\beta \\
&= \frac{1 - 2c' R_{\min}^{-2} pn/c_1}{m} \sum_{r \in S_i^{(0)}} \overline{\rho}_{y_i,r,i}^{(t)} - 2\beta,
\end{aligned}
$$

where the first inequality is by

$$
\begin{aligned}
F_{y_i}(\mathbf{W}_{y_i}^{(t)}, \mathbf{x}_i) &= \frac{1}{m} \sum_{r \in S_i^{(t)}} \langle \mathbf{w}_{y_i,r}^{(t)}, \mathbf{x}_i \rangle \\
&= \frac{1}{m} \sum_{r \in S_i^{(t)}}\left(\langle \mathbf{w}_{y_i,r}^{(0)}, \mathbf{x}_i \rangle + \overline{\rho}_{y_i,r,i}^{(t)} + \sum_{i' \neq i} \rho_{y_i,r,i'}^{(t)} \|\mathbf{x}_{i'}\|_2^{-2} \cdot \langle \mathbf{x}_{i'}, \mathbf{x}_i \rangle\right) \\
&\geq \frac{1}{m}\left(\sum_{r \in S_i^{(t)}} \overline{\rho}_{y_i,r,i}^{(t)} - \sum_{r \in S_i^{(t)}} \sum_{i' \neq i} |\rho_{y_i,r,i'}^{(t)}| R_{\min}^{-2} p - |S_i^{(t)}| \cdot \beta\right),
\end{aligned}
$$

and

$$\langle \mathbf{w}_{-y_i,r}^{(t)}, \mathbf{x}_i \rangle = \langle \mathbf{w}_{-y_i,r}^{(0)}, \mathbf{x}_i \rangle + \sum_{i'=1}^{n} \rho_{-y_i,r,i'}^{(t)} \|\mathbf{x}_{i'}\|_2^{-2} \cdot \langle \mathbf{x}_{i'}, \mathbf{x}_i \rangle$$

$$= \langle \mathbf{w}_{-y_i,r}^{(0)}, \mathbf{x}_i \rangle + \underline{\rho}_{-y_i,r,i}^{(t)} + \sum_{i' \neq i} \rho_{-y_i,r,i'}^{(t)} \|\mathbf{x}_{i'}\|_2^{-2} \cdot \langle \mathbf{x}_{i'}, \mathbf{x}_i \rangle$$

$$\leq \beta + \sum_{i' \neq i} |\rho_{-y_i,r,i'}^{(t)}| R_{\min}^{-2} p,$$

and hence

$$F_{-y_i}(\mathbf{W}_{-y_i}^{(t)}, \mathbf{x}_i) = \frac{1}{m} \sum_{r=1}^{m} \sigma(\langle \mathbf{w}_{-y_i,r}^{(t)}, \mathbf{x}_i \rangle) \leq \frac{1}{m} \sum_{r=1}^{m} \left( \beta + \sum_{i' \neq i} |\rho_{-y_i,r,i'}^{(t)}| R_{\min}^{-2} p \right); \qquad \text{(E.16)}$$

the second inequality is by the first induction hypothesis that $\overline{\rho}_{y_i,r,i}^{(t)} \geq c_1 |\rho_{y_i,r',i'}^{(t)}|, \overline{\rho}_{y_i,r,i}^{(t)} \geq c_1 |\rho_{-y_i,r',i'}^{(t)}|$ and hence $|\rho_{y_i,r',i'}^{(t)}| \leq \frac{1}{c_1 |S_i^{(0)}|} \sum_{r \in S_i^{(0)}} \overline{\rho}_{y_i,r,i}^{(t)}, |\rho_{-y_i,r',i'}^{(t)}| \leq \frac{1}{c_1 |S_i^{(0)}|} \sum_{r \in S_i^{(0)}} \overline{\rho}_{y_i,r,i}^{(t)}$ for $r \in S_i^{(0)}$ and $0 \leq t \leq \widetilde{t} - 1$; the third inequality is by $|S_i^{(t)}| \leq m$; the fourth inequality is by $m \leq c' |S_i^{(0)}|$. Therefore,

$$|\ell_i'^{(t)}| = \frac{1}{1 + \exp \left\{ F_{y_i}(\mathbf{W}_{y_i}^{(t)}, \mathbf{x}_i) - F_{-y_i}(\mathbf{W}_{-y_i}^{(t)}, \mathbf{x}_i) \right\}}$$

$$\leq \exp \left\{ -F_{y_i}(\mathbf{W}_{y_i}^{(t)}, \mathbf{x}_i) + F_{-y_i}(\mathbf{W}_{-y_i}^{(t)}, \mathbf{x}_i) \right\}$$

$$\leq \exp \left\{ -\frac{1 - 2c' R_{\min}^{-2} pn/c_1}{m} \sum_{r \in S_i^{(0)}} \overline{\rho}_{y_i,r,i}^{(t)} + 2\beta \right\}, \forall 0 \leq t \leq \widetilde{t} - 1.$$

By the induction hypothesis, we know that $S_i^{(0)} \subseteq S_i^{(t)}$ for all $0 \leq t \leq \widetilde{t} - 1$ and hence $\sigma'(\langle \mathbf{w}_{y_i,r}^{(t)}, \mathbf{x}_i \rangle) = 1$ for all $r \in S_i^{(0)}$ and $0 \leq t \leq \widetilde{t} - 1$. By (E.15) and (E.16), it follows that for all $0 \leq t \leq \widetilde{t} - 1$

$$\sum_{r \in S_i^{(0)}} \overline{\rho}_{y_i,r,i}^{(t+1)} \geq \sum_{r \in S_i^{(0)}} \overline{\rho}_{y_i,r,i}^{(t)} + \frac{\eta |S_i^{(0)}| \|\mathbf{x}_i\|_2^2 e^{-\beta}}{2nm} \cdot \exp \left\{ -\frac{c'(1 + R_{\min}^{-2} pn/c_1)}{m} \sum_{r \in S_i^{(0)}} \overline{\rho}_{y_i,r,i}^{(t)} \right\},$$

$$\text{(E.17)}$$

$$\sum_{r \in S_i^{(0)}} \overline{\rho}_{y_i,r,i}^{(t+1)} \leq \sum_{r \in S_i^{(0)}} \overline{\rho}_{y_i,r,i}^{(t)} + \frac{\eta |S_i^{(0)}| \|\mathbf{x}_i\|_2^2 e^{2\beta}}{nm} \cdot \exp \left\{ -\frac{1 - 2c' R_{\min}^{-2} pn/c_1}{m} \sum_{r \in S_i^{(0)}} \overline{\rho}_{y_i,r,i}^{(t)} \right\}.$$

$$\text{(E.18)}$$

By applying Lemma H.2 to (E.17) and taking $x_t = \frac{c'(1 + R_{\min}^{-2} pn/c_1)}{m} \sum_{r \in S_i^{(0)}} \overline{\rho}_{y_i,r,i}^{(t)}$, we can get

$$\sum_{r \in S_i^{(0)}} \overline{\rho}_{y_i,r,i}^{(t)} \geq \frac{m}{c'(1 + R_{\min}^{-2} pn/c_1)} \log \left( 1 + \frac{c'(1 + R_{\min}^{-2} pn/c_1)\eta |S_i^{(0)}| \|\mathbf{x}_i\|_2^2 e^{-\beta}}{2nm^2} \cdot t \right), \forall 0 \leq t \leq \widetilde{t}.$$

$$\text{(E.19)}$$

By applying Lemma H.1 to (E.18) and taking $x_t = \frac{1-2c'R_{\min}^{-2}pn/c_1}{m}\sum_{r\in S_i^{(0)}}\overline{\rho}_{y_i,r,i}^{(t)}$, we can get for any $0\le t\le \widetilde{t}$ that

$$\sum_{r\in S_i^{(0)}}\overline{\rho}_{y_i,r,i}^{(t)}\le \frac{m}{1-2c'R_{\min}^{-2}pn/c_1}\log\left(1+\frac{(1-2c'R_{\min}^{-2}pn/c_1)\eta|S_i^{(0)}|\|\mathbf{x}_i\|_2^2e^{2\beta}\cdot}{nm^2}\right.$$

$$\left.\exp\left(\frac{(1-2c'R_{\min}^{-2}pn/c_1)\eta|S_i^{(0)}|\|\mathbf{x}_i\|_2^2e^{2\beta}}{nm^2}\right)\cdot t\right)$$

$$\le \frac{m}{1-2c'R_{\min}^{-2}pn/c_1}\log\left(1+\frac{2(1-2c'R_{\min}^{-2}pn/c_1)\eta|S_i^{(0)}|\|\mathbf{x}_i\|_2^2e^{2\beta}}{nm^2}\cdot t\right),$$
(E.20)

where the last inequality is by $\eta\le (CR_{\max}^2/nm)^{-1}$, $C$ is a large enough constant and hence $(1-2c'R_{\min}^{-2}pn/c_1)\eta|S_i^{(0)}|\|\mathbf{x}_i\|_2^2e^{2\beta}/nm^2\le \log 2$. Since $S_i^{(0)}\subseteq S_i^{(t)}$ for all $0\le t\le \widetilde{t}-1$ and hence $\sigma'(\langle\mathbf{w}_{y_i,r}^{(t)},\mathbf{x}_i\rangle)=1$ for all $r\in S_i^{(0)}$ and $0\le t\le \widetilde{t}-1$, we have

$$\overline{\rho}_{y_i,r,i}^{(t)}=\frac{\eta}{nm}\sum_{s=0}^{t-1}|\ell_i'^{(s)}|\cdot\|\mathbf{x}_i\|_2^2,\forall 0\le t\le \widetilde{t}.$$

Accordingly, it holds that

$$\overline{\rho}_{y_i,r,i}^{(t)}=\overline{\rho}_{y_i,r',i}^{(t)},\forall r,r'\in S_i^{(0)},\forall 0\le t\le \widetilde{t}.$$

Applying this to (E.19) and (E.20), we can get

$$\overline{\rho}_{y_i,r,i}^{(t)}\ge \frac{m}{c'(1+R_{\min}^{-2}pn/c_1)|S_i^{(0)}|}\log\left(1+\frac{c'(1+R_{\min}^{-2}pn/c_1)\eta|S_i^{(0)}|\|\mathbf{x}_i\|_2^2e^{-\beta}}{2nm^2}\cdot t\right)$$

$$\ge \frac{1}{c'(1+R_{\min}^{-2}pn/c_1)}\log\left(1+\frac{c'(1+R_{\min}^{-2}pn/c_1)\eta|S_i^{(0)}|\|\mathbf{x}_i\|_2^2e^{-\beta}}{2nm^2}\cdot t\right),\forall 0\le t\le \widetilde{t},$$

$$\overline{\rho}_{y_i,r,i}^{(t)}\le \frac{m}{(1-2c'R_{\min}^{-2}pn/c_1)|S_i^{(0)}|}\log\left(1+\frac{2(1-2c'R_{\min}^{-2}pn/c_1)\eta|S_i^{(0)}|\|\mathbf{x}_i\|_2^2e^{2\beta}}{nm^2}\cdot t\right),$$

$$\le \frac{c'}{(1-2c'R_{\min}^{-2}pn/c_1)}\log\left(1+\frac{2(1-2c'R_{\min}^{-2}pn/c_1)\eta|S_i^{(0)}|\|\mathbf{x}_i\|_2^2e^{2\beta}}{nm^2}\cdot t\right),\forall 0\le t\le \widetilde{t},$$
(E.21)

By taking

$$c_2=\frac{1}{c'(1+R_{\min}^{-2}pn/c_1)},\ c_3=\frac{c'}{(1-2c'R_{\min}^{-2}pn/c_1)},$$

the above inequalities indicates that the second and third bullets hold at time $t=\widetilde{t}$. For the first bullet, it is only necessary to consider the situation where $\widetilde{t}\ge T_1=C\eta^{-1}nmR_{\max}^{-2}$. In order to apply Lemma H.4 (requiring $b>a$), we loosen the bounds in (E.21) as follows:

$$\overline{\rho}_{y_i,r,i}^{(t)}\ge c_2\log\left(1+\frac{\eta R_{\min}^2e^{-\beta}}{5nm}\cdot t\right),\forall 0\le t\le \widetilde{t},\tag{E.22}$$

$$\overline{\rho}_{y_i,r,i}^{(t)}\le c_3\log\left(1+\frac{6\eta R_{\max}^2e^{2\beta}}{5nm}\cdot t\right),\forall 0\le t\le \widetilde{t},\tag{E.23}$$

where we use $0.4m\le |S_i^{(0)}|\le 0.6m$.

By applying Lemma H.4 to (E.22) and (E.23), we can get for any $i, i' \in [n]$, $r \in S_i^{(0)}$, $r' \in S_{i'}^{(0)}$ and $T_1 \le t \le \tilde{t}$ that

$$
\frac{\overline{\rho}_{y_i,r,i}^{(t)}}{\overline{\rho}_{y_{i'},r',i'}^{(t)}} \ge \frac{c_2}{c_3} \cdot \frac{\log\left(1 + \frac{\eta R_{\min}^2 e^{-\beta}}{5nm} \cdot t\right)}{\log\left(1 + \frac{6\eta R_{\max}^2 e^{2\beta}}{5nm} \cdot t\right)}
$$

$$
\ge \frac{c_2}{c_3} \cdot \frac{\log\left(1 + \frac{\eta R_{\max}^2 e^{2\beta}}{5nm} \cdot T_1\right)}{\log\left(1 + \frac{6\eta R_{\max}^2 e^{2\beta}}{5nm} \cdot T_1\right)}
$$

$$
= \frac{c_2}{c_3} \cdot \frac{\log\left(1 + 0.2Ce^{-\beta}R_{\min}^2\right)}{\log\left(1 + 1.2Ce^{2\beta}R_{\max}^2\right)}.
$$

Notice that $S_{i'}^{(0)} \subseteq S_{i'}^{(t)}$ for all $0 \le t \le \tilde{t} - 1$ and hence $\sigma'(\langle \mathbf{w}_{y_{i'},r}^{(t)}, \mathbf{x}_{i'}\rangle) = 1$ for all $r \in S_{i'}^{(0)}$ and $0 \le t \le \tilde{t} - 1$, we have

$$
|\rho_{j,r'',i'}^{(t)}| = \frac{\eta}{nm} \sum_{s=0}^{t-1} |\ell_{i'}'^{(s)}| \cdot \sigma'(\langle \mathbf{w}_{j,r''}^{(s)}, \mathbf{x}_{i'}\rangle) \cdot \|\mathbf{x}_i\|_2^2
$$

$$
\le \frac{\eta}{nm} \sum_{s=0}^{t-1} |\ell_{i'}'^{(s)}| \cdot \|\mathbf{x}_{i'}\|_2^2, \qquad \forall j \in \{\pm 1\}, r'' \in [m], i' \in [n],
$$

$$
\overline{\rho}_{y_{i'},r',i'}^{(t)} = \frac{\eta}{nm} \sum_{s=0}^{t-1} |\ell_{i'}'^{(s)}| \cdot \|\mathbf{x}_{i'}\|_2^2, \qquad \forall r' \in S_{i'}^{(0)}, i' \in [n],
$$

and hence $|\rho_{j,r',i'}^{(t)}| \le \overline{\rho}_{y_{i'},r',i'}^{(t)}$ for $0 \le t \le \tilde{t}$. Therefore, as long as

$$
c_1 \le \frac{c_2}{c_3} \cdot \frac{\log\left(1 + 0.2Ce^{-\beta}R_{\min}^2\right)}{\log\left(1 + 1.2Ce^{2\beta}R_{\max}^2\right)},
$$

the first bullet hold for time $t = \tilde{t}$. This condition holds as long as

$$
c' = 2.5,
$$
$$
2c'c_1^{-1}R_{\min}^{-2}pn \le 0.5 \quad \Longrightarrow \quad c_2 \ge 0.37, c_3 \le 5,
$$
$$
c_1 = \frac{\log\left(1 + 0.2Ce^{-\beta}R_{\min}^2\right)}{14\log\left(1 + 1.2Ce^{2\beta}R_{\max}^2\right)}.
$$

Finally, we verify that $S_i^{(0)} \subseteq S_i^{(\tilde{t})}$. For $r \in S_i^{(0)}$, we have

$$
\langle \mathbf{w}_{y_i,r}^{(\tilde{t})}, \mathbf{x}_i\rangle = \langle \mathbf{w}_{y_i,r}^{(0)}, \mathbf{x}_i\rangle + \sum_{i'=1}^{n} \rho_{y_i,r,i'}^{(\tilde{t})} \|\mathbf{x}_{i'}\|_2^{-2} \cdot \langle \mathbf{x}_{i'}, \mathbf{x}_i\rangle
$$

$$
= \langle \mathbf{w}_{y_i,r}^{(0)}, \mathbf{x}_i\rangle + \overline{\rho}_{y_i,r,i}^{(\tilde{t})} + \sum_{i' \ne i} \rho_{y_i,r,i'}^{(\tilde{t})} \|\mathbf{x}_{i'}\|_2^{-2} \cdot \langle \mathbf{x}_{i'}, \mathbf{x}_i\rangle
$$

$$
\ge \overline{\rho}_{y_i,r,i}^{(\tilde{t})} - \sum_{i' \ne i} |\rho_{y_i,r,i'}^{(\tilde{t})}| R_{\min}^{-2} p
$$

$$
\ge \overline{\rho}_{y_i,r,i}^{(\tilde{t})} - (R_{\min}^{-2}pn/c_1)\overline{\rho}_{y_i,r,i'}^{(\tilde{t})}
$$

$$
= (1 - R_{\min}^{-2}pn/c_1) \cdot \overline{\rho}_{y_i,r,i}^{(\tilde{t})} \ge 0,
$$

where the second inequality is by $|\rho^{(\widetilde{t})}_{y_i,r,i'}| \leq \overline{\rho}^{(\widetilde{t})}_{y_i,r,i}/c_1$. This implies that $S_i^{(0)} \subseteq S_i^{(t)}$ holds for time $t = \widetilde{t}$, which completes the induction. $\qquad\square$

Next, we show that $|\underline{\rho}^{(t)}_{-y_i,r,i'}|$ will be much smaller than $|\overline{\rho}^{(t)}_{y_i,r',i}|$ as the training goes on.

**Lemma E.2.** There exists time $T_2$ and constant $c$ such that for any time $t \geq T_2$

$$|\underline{\rho}^{(t)}_{-y_i,r,i'}| \leq cR_{\min}^{-2}pn|\overline{\rho}^{(t)}_{y_i,r',i}|,$$

where $r \in [m], r' \in S_i^{(0)}$ and $i, i' \in [n]$ satisfying $-y_i = y_{i'}$.

*Proof of Lemma E.2.* First, we will prove by induction that for $r \in [m], r' \in S_i^{(0)}$ and $i, i' \in [n]$ satisfying $-y_i = y_{i'}$ it holds that

$$|\underline{\rho}^{(t)}_{-y_i,r,i'}| \leq \beta + 1 + R_{\min}^{-2}pn|\overline{\rho}^{(t)}_{y_i,r',i}|/c_1. \tag{E.24}$$

This result holds naturally when $t = 0$ since all the coefficients are zero. Suppose that there exists time $\widetilde{t}$ such that the induction hypothesis (E.24) holds for all time $t \leq \widetilde{t} - 1$. We aim to prove that (E.24) also holds for $t = \widetilde{t}$. In the following analysis, two cases will be considered: $|\underline{\rho}^{(\widetilde{t}-1)}_{y_i,r,i'}| > \beta + \sum_{k \neq i'} |\rho^{(\widetilde{t}-1)}_{y_i,r,k}| \|\mathbf{x}_k\|_2^{-2}p$ and $|\underline{\rho}^{(\widetilde{t}-1)}_{y_i,r,i'}| \leq \beta + \sum_{k \neq i'} |\rho^{(\widetilde{t}-1)}_{y_i,r,k}| \|\mathbf{x}_k\|_2^{-2}p$.

For if $|\underline{\rho}^{(\widetilde{t}-1)}_{y_i,r,i'}| > \beta + \sum_{k \neq i'} |\rho^{(\widetilde{t}-1)}_{y_i,r,k}| \|\mathbf{x}_k\|_2^{-2}p$, then by the decomposition (5.1) we have

$$\langle \mathbf{w}^{(\widetilde{t}-1)}_{y_i,r}, \mathbf{x}_{i'} \rangle = \langle \mathbf{w}^{(0)}_{y_i,r}, \mathbf{x}_{i'} \rangle + \underline{\rho}^{(\widetilde{t}-1)}_{-y_i,r,i'} + \sum_{k \neq i'} \rho^{(\widetilde{t}-1)}_{y_i,r,k} \|\mathbf{x}_k\|_2^{-2} \langle \mathbf{x}_k, \mathbf{x}_{i'} \rangle$$

$$\leq \underline{\rho}^{(\widetilde{t}-1)}_{-y_i,r,i'} + \beta + \sum_{k \neq i'} |\rho^{(\widetilde{t}-1)}_{y_i,r,k}| \|\mathbf{x}_k\|_2^{-2}p < 0.$$

and hence

$$\underline{\rho}^{(\widetilde{t})}_{-y_i,r,i'} = \underline{\rho}^{(\widetilde{t}-1)}_{-y_i,r,i'} + \frac{\eta}{nm} \cdot \ell_i'^{(\widetilde{t}-1)} \cdot \sigma'(\langle \mathbf{w}^{(\widetilde{t}-1)}_{y_i,r}, \mathbf{x}_{i'} \rangle) \cdot \|\mathbf{x}_{i'}\|_2^2 = \underline{\rho}^{(\widetilde{t}-1)}_{-y_i,r,i'}.$$

Therefore, by induction hypothesis (E.24) at time $t = \widetilde{t} - 1$, we have

$$\underline{\rho}^{(\widetilde{t})}_{-y_i,r,i'} = \underline{\rho}^{(\widetilde{t}-1)}_{-y_i,r,i'} \leq \beta + 1 + R_{\min}^{-2}pn|\overline{\rho}^{(\widetilde{t}-1)}_{y_i,r',i}|/c_1 \leq \beta + 1 + R_{\min}^{-2}pn|\overline{\rho}^{(\widetilde{t})}_{y_i,r',i}|/c_1.$$

For if $|\underline{\rho}^{(\widetilde{t}-1)}_{y_i,r,i'}| \leq \beta + \sum_{k \neq i'} |\rho^{(\widetilde{t}-1)}_{y_i,r,k}| \|\mathbf{x}_k\|_2^{-2}p$, by the first bullet in Lemma E.1, we have

$$|\underline{\rho}^{(\widetilde{t}-1)}_{-y_i,r,i'}| \leq \beta + \sum_{k \neq i'} |\overline{\rho}^{(\widetilde{t}-1)}_{y_i,r',i}| \|\mathbf{x}_k\|_2^{-2}p/c_1 \leq \beta + |\overline{\rho}^{(\widetilde{t}-1)}_{y_i,r',i}| R_{\min}^{-2}pn/c_1, \tag{E.25}$$

and thus

$$-\underline{\rho}^{(\widetilde{t})}_{-y_i,r,i'} = -\underline{\rho}^{(\widetilde{t}-1)}_{-y_i,r,i'} + \frac{\eta}{nm} \cdot |\ell_i'^{(\widetilde{t}-1)}| \cdot \sigma'(\langle \mathbf{w}^{(\widetilde{t}-1)}_{y_i,r}, \mathbf{x}_{i'} \rangle) \cdot \|\mathbf{x}_{i'}\|_2^2$$

$$\leq -\underline{\rho}^{(\widetilde{t}-1)}_{-y_i,r,i'} + \frac{\eta R_{\max}^2}{nm}$$

$$\leq \beta + 1 + |\overline{\rho}^{(t)}_{y_i,r',i}| R_{\min}^{-2}pn/c_1,$$

where the last inequality is by (E.25) and $\eta \leq (CR_{\max}^2/nm)^{-1}$ with a sufficiently large constant $C$. This demonstrates that inequality (E.24) holds for $t = \widetilde{t}$, thereby completing the induction process. By Lemma E.1, we know that there exists time $T'$ such that

$$|\overline{\rho}^{(t)}_{y_i,r',i}| \geq c_1(\beta + 1)R_{\min}^2/pn,$$

for any time $t \geq T'$. Taking $T_2 = T'$ and $c = 2/c_1$, we have

$$|\underline{\rho}^{(t)}_{-y_i,r,i'}| \leq \beta + 1 + R_{\min}^{-2}pn|\overline{\rho}^{(t)}_{y_i,r',i}|/c_1 \leq 2R_{\min}^{-2}pn|\overline{\rho}^{(t)}_{y_i,r',i}|/c_1 = cR_{\min}^{-2}pn|\overline{\rho}^{(t)}_{y_i,r',i}|,$$

which completes the proof. □

Given Lemma E.2, the following corollary can be directly obtained.

**Corollary E.3.** There exists time $T_3$ and constant $c$ such that for any time $t \geq T_3$

$$|\rho_{-y_i,r,i'}^{(t)}| \leq c R_{\min}^{-2} pn |\overline{\rho}_{y_i,r',i}^{(t)}|, \forall r \in [m], r' \in S_i^{(0)}, i, i' \in [n] \text{ with } -y_i = y_{i'}.$$

# F  Stable Rank of ReLU Network

In this section, we consider the properties of stable rank of the weight matrix $\mathbf{W}^{(t)}$ found by gradient descent at time $t$, defined as $\|\mathbf{W}^{(t)}\|_F^2 / \|\mathbf{W}^{(t)}\|_2^2$. Given Lemma E.1, we have following coefficient update rule for any $t \geq 0$, $i \in [n]$ and $r \in S_i^{(0)}$:

$$\overline{\rho}_{y_i,r,i}^{(t+1)} = \overline{\rho}_{y_i,r,i}^{(t)} + \frac{\eta}{nm} \cdot |\ell_i'^{(t)}| \cdot \|\mathbf{x}_i\|_2^2, \tag{F.1}$$

where

$$|\ell_i'^{(t)}| = \frac{1}{1 + \exp\{F_{y_i}(\mathbf{W}_{y_i}^{(t)}, \mathbf{x}_i) - F_{-y_i}(\mathbf{W}_{-y_i}^{(t)}, \mathbf{x}_i)\}}.$$

Now we are ready to prove the first bullet of Theorem 4.3.

**Lemma F.1.** For two-layer ReLU neural network defined in (E.1), under the same condition as Theorem 4.3, the stable rank of $\mathbf{W}_j^{(t)}$ satisfies the following property:

$$\limsup_{t \to \infty} \frac{\|\mathbf{W}_j^{(t)}\|_F^2}{\|\mathbf{W}_j^{(t)}\|_2^2} \leq C,$$

where $C = \Theta(1)$ is a constant.

*Proof of Lemma F.1.* By decomposition (5.1), we have

$$\mathbf{w}_{j,r}^{(t)} - \mathbf{w}_{j,r}^{(0)} = \sum_{i=1}^n \rho_{j,r,i}^{(t)} \cdot \|\mathbf{x}_i\|_2^{-2} \cdot \mathbf{x}_i = \left[\rho_{j,r,1}^{(t)}\|\mathbf{x}_1\|_2^{-2}, \cdots, \rho_{j,r,n}^{(t)}\|\mathbf{x}_n\|_2^{-2}\right] \cdot \begin{bmatrix} \mathbf{x}_1 \\ \vdots \\ \mathbf{x}_n \end{bmatrix},$$

and

$$\mathbf{W}_j^{(t)} - \mathbf{W}_j^{(0)} = \underbrace{\begin{bmatrix} \rho_{j,1,1}^{(t)}\|\mathbf{x}_1\|_2^{-2} & \rho_{j,1,2}^{(t)}\|\mathbf{x}_1\|_2^{-2} & \cdots & \rho_{j,1,n}^{(t)}\|\mathbf{x}_n\|_2^{-2} \\ \rho_{j,2,1}^{(t)}\|\mathbf{x}_1\|_2^{-2} & \rho_{j,2,2}^{(t)}\|\mathbf{x}_1\|_2^{-2} & \cdots & \rho_{j,2,n}^{(t)}\|\mathbf{x}_n\|_2^{-2} \\ \vdots & \vdots & \ddots & \vdots \\ \rho_{j,m,1}^{(t)}\|\mathbf{x}_1\|_2^{-2} & \rho_{j,m,2}^{(t)}\|\mathbf{x}_1\|_2^{-2} & \cdots & \rho_{j,m,n}^{(t)}\|\mathbf{x}_n\|_2^{-2} \end{bmatrix}}_{\mathbf{A}_t} \cdot \underbrace{\begin{bmatrix} \mathbf{x}_1 \\ \vdots \\ \mathbf{x}_n \end{bmatrix}}_{:=\mathbf{X}}.$$

Let $\mathbf{a}_i(t)^\top$ be the $i$-th column of $\mathbf{A}_t$. It follows that

$$\|\mathbf{W}_j^{(t)} - \mathbf{W}_j^{(0)}\|_F^2 = \text{Tr}(\mathbf{A}_t \mathbf{X} \mathbf{X}^\top \mathbf{A}_t^\top)$$

$$= \text{Tr}\left(\left(\sum_{i=1}^n \mathbf{a}_i(t)^\top \mathbf{x}_i\right)\left(\sum_{i=1}^n \mathbf{x}_i^\top \mathbf{a}_i(t)\right)\right)$$

$$= \text{Tr}\left(\sum_{i=1}^n \sum_{i'=1}^n \mathbf{a}_i(t)^\top \mathbf{x}_i \mathbf{x}_{i'}^\top \mathbf{a}_{i'}(t)\right)$$

$$= \sum_{i=1}^n \sum_{i'=1}^n \langle \mathbf{x}_i, \mathbf{x}_{i'}\rangle \cdot \text{Tr}(\mathbf{a}_i(t)^\top \mathbf{a}_{i'}(t))$$

$$= \sum_{i=1}^{n} \|\mathbf{x}_i\|_2^2 \cdot \text{Tr}(\mathbf{a}_i(t)^\top \mathbf{a}_i(t)) + \sum_{i \neq i'} \langle \mathbf{x}_i, \mathbf{x}_{i'} \rangle \cdot \text{Tr}(\mathbf{a}_i(t)^\top \mathbf{a}_{i'}(t))$$

$$\leq R_{\max}^2 \sum_{i=1}^{n} \text{Tr}(\mathbf{a}_i(t)^\top \mathbf{a}_i(t)) + p \sum_{i \neq i'} |\text{Tr}(\mathbf{a}_i(t)^\top \mathbf{a}_{i'}(t))|,$$

and

$$\text{Tr}(\mathbf{a}_i(t)^\top \mathbf{a}_i(t)) = \sum_{r=1}^{m}([\mathbf{a}_i(t)]_r)^2 = \sum_{r=1}^{m}(\rho_{j,r,i}^{(t)}\|\mathbf{x}_i\|_2^{-2})^2 \leq \sum_{r=1}^{m}(\rho_{j,r,i}^{(t)})^2 R_{\min}^{-4} \leq C m R_{\min}^{-4}(\log(t))^2,$$

$$|\text{Tr}(\mathbf{a}_i(t)^\top \mathbf{a}_{i'}(t))| \leq \sum_{r=1}^{m}\big|[\mathbf{a}_i(t)]_r[\mathbf{a}_{i'}(t)]_r\big| = \sum_{r=1}^{m}|\rho_{j,r,i}^{(t)}\rho_{j,r,i'}^{(t)}|\|\mathbf{x}_i\|_2^{-2}\|\mathbf{x}_{i'}\|_2^{-2} \leq C m R_{\min}^{-4}(\log(t))^2.$$

Accordingly, we have

$$\|\mathbf{W}_j^{(t)} - \mathbf{W}_j^{(0)}\|_F^2 \leq C m n R_{\max}^2 R_{\min}^{-4}(\log(t))^2 + C m n^2 p R_{\min}^{-4}(\log(t))^2$$
$$= C m n R_{\max}^2 R_{\min}^{-4}(1 + R_{\max}^{-2} n p)(\log(t))^2$$
$$\leq C' m n R_{\max}^2 R_{\min}^{-4}(\log(t))^2.$$

On the other hand, we will give an lower bound for $\|\mathbf{W}_j^{(t)} - \mathbf{W}_j^{(0)}\|_2$.

$$\|\mathbf{W}_j^{(t)} - \mathbf{W}_j^{(0)}\|_2 = \max_{\mathbf{y} \in S^{d-1}} \|(\mathbf{W}_j^{(t)} - \mathbf{W}_j^{(0)})\mathbf{y}\|_2$$
$$\geq \frac{\|(\mathbf{W}_j^{(t)} - \mathbf{W}_j^{(0)})\mathbf{X}^\top(\mathbf{X}\mathbf{X}^\top)^{-1}\mathbf{1}_n\|_2}{\|\mathbf{X}^\top(\mathbf{X}\mathbf{X}^\top)^{-1}\mathbf{1}_n\|_2}$$
$$= \frac{\|\mathbf{A}_t\mathbf{1}_n\|_2}{\|\mathbf{X}^\top(\mathbf{X}\mathbf{X}^\top)^{-1}\mathbf{1}_n\|_2}.$$

We first provide a lower bound for $\|\mathbf{A}_t\mathbf{1}_n\|_2$. Note that

$$\mathbf{A}_t\mathbf{1}_n = \begin{bmatrix} \sum_{i=1}^{n}\rho_{j,1,i}^{(t)}\|\mathbf{x}_i\|_2^{-2} \\ \vdots \\ \sum_{i=1}^{n}\rho_{j,m,i}^{(t)}\|\mathbf{x}_i\|_2^{-2} \end{bmatrix},$$

we need to bound $\sum_{i=1}^{n}\rho_{j,r,i}^{(t)}\|\mathbf{x}_i\|_2^{-2}, r \in [m]$. By Corollary E.3, there exists time $T$ such that for any $t \geq T$, $|\underline{\rho}_{y_i,r,i'}^{(t)}| \leq c R_{\min}^{-2} p n \overline{\rho}_{y_i,r,i}^{(t)}, \forall i \in S_r^{(0)}$ and $\forall i' \in [n]$ with $y_{i'} = -y_i$. Therefore, we have for $t \geq T$ that

$$\sum_{r=1}^{m}\sum_{i=1}^{n}\rho_{j,r,i}^{(t)}\|\mathbf{x}_i\|_2^{-2} = \sum_{r=1}^{m}\left(\sum_{i \in S_j}\overline{\rho}_{j,r,i}^{(t)}\|\mathbf{x}_i\|_2^{-2} + \sum_{i \in S_{-j}}\underline{\rho}_{j,r,i}^{(t)}\|\mathbf{x}_i\|_2^{-2}\right)$$

$$= \sum_{i \in S_j}\sum_{r=1}^{m}\overline{\rho}_{y_i,r,i}^{(t)}\|\mathbf{x}_i\|_2^{-2} + \sum_{i \in S_{-j}}\sum_{r=1}^{m}\underline{\rho}_{-y_i,r,i}^{(t)}\|\mathbf{x}_i\|_2^{-2}$$

$$\geq \sum_{i \in S_j}\sum_{r=1}^{m}\overline{\rho}_{y_i,r,i}^{(t)}R_{\max}^{-2} + \sum_{i \in S_{-j}}\sum_{r=1}^{m}\underline{\rho}_{-y_i,r,i}^{(t)}R_{\min}^{-2}$$

$$\geq \sum_{i \in S_j}\sum_{r \in S_i^{(0)}}\overline{\rho}_{y_i,r,i}^{(t)}R_{\max}^{-2} - \frac{m|S_{-j}|}{|S_j|}\sum_{i \in S_j}\frac{c R_{\min}^{-2}pn}{|S_i^{(0)}|}\sum_{r \in S_i^{(0)}}\overline{\rho}_{y_i,r,i}^{(t)}R_{\min}^{-2}$$

$$\geq \sum_{i \in S_j}\sum_{r \in S_i^{(0)}}\overline{\rho}_{y_i,r,i}^{(t)}R_{\max}^{-2} - \frac{m|S_{-j}| \cdot c R_{\min}^{-4}R_{\max}^2 pn}{|S_j| \cdot \min_{i \in S_j}|S_i^{(0)}|}\sum_{i \in S_j}\sum_{r \in S_i^{(0)}}\overline{\rho}_{y_i,r,i}^{(t)}R_{\max}^{-2}$$

$$\geq \frac{R_{\max}^{-2}}{2} \sum_{i \in S_j} \sum_{r \in S_i^{(0)}} \overline{\rho}_{j,r,i}^{(t)}, \tag{F.2}$$

where the second inequality is by Corollary E.3 and hence

$$|\underline{\rho}_{-y_i,r,i}^{(t)}| \leq cR_{\min}^{-2}pn\overline{\rho}_{y_{i'},r',i'}^{(t)}, \forall r, r' \in [m], \forall i, i' \in [n],$$

$$|\underline{\rho}_{-y_i,r,i}^{(t)}| \leq \frac{cR_{\min}^{-2}pn}{|S_{i'}^{(0)}|} \sum_{r \in S_{i'}^{(0)}} \overline{\rho}_{y_{i'},r,i'}^{(t)}, \forall r \in [m], \forall i, i' \in [n],$$

$$|\underline{\rho}_{-y_i,r,i}^{(t)}| \leq \frac{1}{|S_j|} \sum_{i \in S_j} \frac{cR_{\min}^{-2}pn}{|S_i^{(0)}|} \sum_{r \in S_i^{(0)}} \overline{\rho}_{y_i,r,i}^{(t)}, \forall r \in [m], \forall i \in [n],$$

the last inequality is by

$$\frac{m|S_{-j}| \cdot cR_{\min}^{-4}R_{\max}^2 pn}{|S_j| \cdot \min_{i \in S_j} |S_i^{(0)}|} \leq c'cR_{\min}^{-4}R_{\max}^2 pn \cdot \frac{|S_{-j}|}{|S_j|} \leq \frac{1}{2}.$$

Then, we have for $t \geq T$ that

$$\begin{aligned}
\|\mathbf{A}_t \mathbf{1}_n\|_2 &= \sqrt{\sum_{r=1}^m \left( \sum_{i=1}^n \rho_{j,r,i}^{(t)} \|\mathbf{x}_i\|_2^{-2} \right)^2} \\
&\geq \left| \sum_{r=1}^m \sum_{i=1}^n \rho_{j,r,i}^{(t)} \|\mathbf{x}_i\|_2^{-2} / \sqrt{m} \right| \\
&\geq \frac{R_{\max}^{-2}}{2\sqrt{m}} \sum_{i \in S_j} \sum_{r \in S_i^{(0)}} \overline{\rho}_{j,r,i}^{(t)} \\
&\geq \frac{R_{\max}^{-2}|S_j||S_i^{(0)}|}{2\sqrt{m}} \cdot \log \left( 1 + \frac{\eta R_{\min}^2 e^{-\beta}}{2c'nm} \cdot t \right) \\
&\geq CR_{\max}^{-2}\sqrt{m}n \log(t)
\end{aligned}$$

where the second inequality is is by (F.2); the third inequality is by the second bullet of Lemma E.1. For $\|\mathbf{X}^\top (\mathbf{XX}^\top)^{-1} \mathbf{1}_n\|_2$, we have

$$\begin{aligned}
\|\mathbf{X}^\top (\mathbf{XX}^\top)^{-1} \mathbf{1}_n\|_2 &= \sqrt{\mathbf{1}_n^\top (\mathbf{XX}^\top)^{-1} \mathbf{XX}^\top (\mathbf{XX}^\top)^{-1} \mathbf{1}_n} \\
&= \sqrt{\mathbf{1}_n^\top (\mathbf{XX}^\top)^{-1} \mathbf{1}_n} \\
&\leq \frac{\|\mathbf{1}_n\|_2}{\sqrt{\lambda_{\min}(\mathbf{XX}^\top)}}
\end{aligned}$$

By the Gershgorin circle theorem, we know that $\lambda_{\min}(\mathbf{XX}^\top)$ lies within at least one of the Gershgorin discs $D((\mathbf{XX}^\top)_{ii}, R_i), i \in [n]$ where $D((\mathbf{XX}^\top)_{ii}, R_i)$ is a closed disc centered at $(\mathbf{XX}^\top)_{ii} = \|\mathbf{x}_i\|_2^2$ with radius $R_i = \sum_{i' \neq i} |(\mathbf{XX}^\top)_{ii'}| = \sum_{i' \neq i} |\langle \mathbf{x}_i, \mathbf{x}_{i'} \rangle|$. Assume $\lambda_{\min}(\mathbf{XX}^\top)$ lies within $D((\mathbf{XX}^\top)_{ii}, R_i)$, then we can get following lower bound for $\lambda_{\min}(\mathbf{XX}^\top)$:

$$\lambda_{\min}(\mathbf{XX}^\top) \geq \|\mathbf{x}_i\|_2^2 - \sum_{i' \neq i} |\langle \mathbf{x}_i, \mathbf{x}_{i'} \rangle| \geq R_{\min}^2 - np = (1 - R_{\min}^{-2}np)R_{\min}^2 \geq R_{\min}^2/2.$$

Therefore, we have

$$\|\mathbf{X}^\top (\mathbf{XX}^\top)^{-1} \mathbf{1}_n\|_2 \leq \frac{\|\mathbf{1}_n\|_2}{\sqrt{\lambda_{\min}(\mathbf{XX}^\top)}} \leq \frac{\sqrt{2n}}{R_{\min}}.$$

Accordingly,

$$\|\mathbf{W}_j^{(t)} - \mathbf{W}_j^{(0)}\|_2 \geq \frac{\|\mathbf{A}_t \mathbf{1}_n\|_2}{\|\mathbf{X}^\top (\mathbf{X}\mathbf{X}^\top)^{-1}\mathbf{1}_n\|_2} \geq \frac{CR_{\max}^{-2}\sqrt{mn}\log(t)}{\sqrt{2n}/R_{\min}} = C'' R_{\max}^{-2} R_{\min} \sqrt{mn}\log(t).$$

Therefore, we have for $t \geq T$ that

$$\frac{\|\mathbf{W}_j^{(t)} - \mathbf{W}_j^{(0)}\|_F^2}{\|\mathbf{W}_j^{(t)} - \mathbf{W}_j^{(0)}\|_2^2} \leq \frac{C' mn R_{\max}^2 R_{\min}^{-4}(\log(t))^2}{C''^2 mn R_{\max}^{-4} R_{\min}^2 (\log(t))^2} \leq \frac{C' R_{\max}^6}{C''^2 R_{\min}^6},$$

which completes the proof. $\qquad\square$

Next, we will provide an example of training data satisfying the condition in Theorem 4.3 and prove that the stable rank of $\mathbf{W}_j^{(t)}$ trained by gradient descent using such data will converge to $2 \pm o(1)$. We first provide the following lemma about the increasing rate of coefficients $\rho_{j,r,i}^{(t)}$.

**Lemma F.2.** If training data $\mathbf{x}_1, \cdots, \mathbf{x}_n$ are mutually orthogonal, the activation pattern after time $T = C\eta^{-1}R_{\min}^{-2}/nm$ is determined by the activation pattern at initialization, i.e.,

$$
\begin{array}{ll}
\langle \mathbf{w}_{y_i,r}^{(t)}, \mathbf{x}_i \rangle \geq 0, & \text{if } \langle \mathbf{w}_{y_i,r}^{(0)}, \mathbf{x}_i \rangle \geq 0, \\[4pt]
\langle \mathbf{w}_{y_i,r}^{(t)}, \mathbf{x}_i \rangle < 0, & \text{if } \langle \mathbf{w}_{y_i,r}^{(0)}, \mathbf{x}_i \rangle < 0, \\[4pt]
\langle \mathbf{w}_{-y_i,r}^{(t)}, \mathbf{x}_i \rangle < 0, & \text{if } \langle \mathbf{w}_{-y_i,r}^{(0)}, \mathbf{x}_i \rangle \geq 0, \\[4pt]
\langle \mathbf{w}_{-y_i,r}^{(t)}, \mathbf{x}_i \rangle < 0, & \text{if } \langle \mathbf{w}_{-y_i,r}^{(0)}, \mathbf{x}_i \rangle < 0,
\end{array}
$$

for any time $t \geq T$. Besides, $\rho_{j,r,i}^{(t)}$ satisfy the following properties:

$$
\begin{array}{ll}
\overline{\rho}_{y_i,r,i}^{(t)} = 0, \forall t \geq 0, & \text{if } \langle \mathbf{w}_{j,r}^{(t)}, \mathbf{x}_i \rangle < 0, \\[4pt]
\overline{\rho}_{y_i,r,i}^{(t)} = \overline{\rho}_{y_i,r,i}^{(T)}, \forall t \geq T, & \text{if } \langle \mathbf{w}_{-y_i,r}^{(t)}, \mathbf{x}_i \rangle \geq 0, \\[4pt]
\lim_{t \to \infty} \overline{\rho}_{y_i,r,i}^{(t)}/\log t = m/|S_i^{(0)}|, & \text{if } \langle \mathbf{w}_{y_i,r}^{(t)}, \mathbf{x}_i \rangle \geq 0.
\end{array}
$$

*Proof of Lemma F.2.* **Part 1.** For if $\langle \mathbf{w}_{j,r}^{(0)}, \mathbf{x}_i \rangle < 0$, we first prove by induction that

$$\rho_{j,r,i}^{(t)} = 0, \langle \mathbf{w}_{j,r}^{(t)}, \mathbf{x}_i \rangle = \langle \mathbf{w}_{j,r}^{(0)}, \mathbf{x}_i \rangle < 0, \forall t \geq 0. \tag{F.3}$$

The result is obvious at $t = 0$ as all the coefficients are zero. Suppose that there exists $\widetilde{t}$ such that (F.3) holds for all time $0 \leq t \leq \widetilde{t} - 1$. We aim to prove that (F.3) also holds for $t = \widetilde{t}$. Recall that by (5.4), (5.5) and with (F.3) at time $\widetilde{t} - 1$, we have

$$\overline{\rho}_{j,r,i}^{(\widetilde{t})} = \overline{\rho}_{j,r,i}^{(\widetilde{t}-1)} - \frac{\eta}{nm} \cdot \ell_i'^{(\widetilde{t}-1)} \cdot \sigma'(\langle \mathbf{w}_{j,r}^{(\widetilde{t}-1)}, \mathbf{x}_i \rangle) \cdot \|\mathbf{x}_i\|_2^2 \cdot \mathbb{1}(y_i = j) = \overline{\rho}_{j,r,i}^{(\widetilde{t}-1)} = 0,$$

$$\underline{\rho}_{j,r,i}^{(\widetilde{t})} = \underline{\rho}_{j,r,i}^{(\widetilde{t}-1)} + \frac{\eta}{nm} \cdot \ell_i'^{(\widetilde{t}-1)} \cdot \sigma'(\langle \mathbf{w}_{j,r}^{(\widetilde{t}-1)}, \mathbf{x}_i \rangle) \cdot \|\mathbf{x}_i\|_2^2 \cdot \mathbb{1}(y_i = -j) = \underline{\rho}_{j,r,i}^{(\widetilde{t}-1)} = 0.$$

By (5.1) and the orthogonality of training data, we can get

$$\langle \mathbf{w}_{j,r}^{(\widetilde{t})}, \mathbf{x}_i \rangle = \langle \mathbf{w}_{j,r}^{(0)}, \mathbf{x}_i \rangle + \sum_{i'=1}^{n} \rho_{j,r,i'}^{(\widetilde{t})} \|\mathbf{x}_{i'}\|_2^{-2} \cdot \langle \mathbf{x}_{i'}, \mathbf{x}_i \rangle = \langle \mathbf{w}_{j,r}^{(0)}, \mathbf{x}_i \rangle + \rho_{j,r,i}^{(\widetilde{t})} = \langle \mathbf{w}_{j,r}^{(0)}, \mathbf{x}_i \rangle < 0.$$

Therefore, (F.3) holds at time $\widetilde{t}$, which completes the induction.

For if $\langle \mathbf{w}_{j,r}^{(0)}, \mathbf{x}_i \rangle \geq 0$ and $j = y_i$, we will next prove by induction that

$$\overline{\rho}_{j,r,i}^{(t)} \geq 0, \langle \mathbf{w}_{j,r}^{(t)}, \mathbf{x}_i \rangle \geq \langle \mathbf{w}_{j,r}^{(0)}, \mathbf{x}_i \rangle \geq 0, \forall t \geq 0. \tag{F.4}$$

The result is natural at $t = 0$. Suppose that there exists $\widetilde{t}$ such that (F.4) hold for all time $0 \leq t \leq \widetilde{t} - 1$. By (5.4) and (F.4) at time $\widetilde{t} - 1$, we have

$$\overline{\rho}_{j,r,i}^{(\widetilde{t})} = \overline{\rho}_{j,r,i}^{(\widetilde{t}-1)} + \frac{\eta}{nm} \cdot |\ell_i'^{(\widetilde{t}-1)}| \cdot \|\mathbf{x}_i\|_2^2 \geq \overline{\rho}_{j,r,i}^{(\widetilde{t}-1)} \geq 0$$

and hence the orthogonality of training data, we can get

$$\langle \mathbf{w}_{j,r}^{(\widetilde{t})}, \mathbf{x}_i \rangle = \langle \mathbf{w}_{j,r}^{(0)}, \mathbf{x}_i \rangle + \rho_{j,r,i}^{(\widetilde{t})} = \langle \mathbf{w}_{j,r}^{(0)}, \mathbf{x}_i \rangle \geq 0.$$

Therefore, (F.4) hold at time $\widetilde{t}$, which completes the induction.

For if $\langle \mathbf{w}_{j,r}^{(0)}, \mathbf{x}_i \rangle \geq 0$ and $j = -y_i$, we first show that under the same condition as Theorem 4.3 it holds that

$$\underline{\rho}_{-j,r,i}^{(T)} < -\langle \mathbf{w}_{j,r}^{(0)}, \mathbf{x}_i \rangle.$$

Since $T = C\eta^{-1} R_{\min}^{-2}/nm$, we have $\max_{j,r,i}\{|\rho_{j,r,i}^{(t)}|\} = O(1)$ for $t \leq T$. Therefore, we know that $F_{+1}(\mathbf{W}_{+1}^{(t)}, \mathbf{x}_i), F_{-1}(\mathbf{W}_{-1}^{(t)}, \mathbf{x}_i) = O(1)$. Thus there exists a positive constant $c$ such that $-\ell_i'^{(t)} \geq c$ for all $i \in [n]$. Here we use the method of proof by contradiction. Assume $\underline{\rho}_{-j,r,i}^{(T)} \geq -\langle \mathbf{w}_{j,r}^{(0)}, \mathbf{x}_i \rangle$. Since $\underline{\rho}_{-j,r,i}^{(T)} \leq \underline{\rho}_{-j,r,i}^{(t)}$ for $0 \leq t \leq T$ which can be seen from (5.5), we have $\underline{\rho}_{-j,r,i}^{(t)} \geq -\langle \mathbf{w}_{j,r}^{(0)}, \mathbf{x}_i \rangle$ for all $t \leq T$. Then, we can get

$$\langle \mathbf{w}_{j,r}^{(t)}, \mathbf{x}_i \rangle = \langle \mathbf{w}_{j,r}^{(0)}, \mathbf{x}_i \rangle + \rho_{j,r,i}^{(t)} \geq 0, \forall t \leq T.$$

Therefore, by the non-negativeness of $\langle \mathbf{w}_{j,r}^{(t)}, \mathbf{x}_i \rangle$ and (5.5), we can get

$$|\underline{\rho}_{-j,r,i}^{(t+1)}| = |\underline{\rho}_{-j,r,i}^{(t)}| + \frac{\eta}{nm} \cdot |\ell_i'^{(t)}| \cdot \|\mathbf{x}_i\|_2^2 \geq |\underline{\rho}_{-j,r,i}^{(t)}| + \frac{c\eta\|\mathbf{x}_i\|_2^2}{nm}$$

and hence

$$|\underline{\rho}_{-j,r,i}^{(T)}| \geq \frac{c\eta\|\mathbf{x}_i\|_2^2 T}{nm} = cC\|\mathbf{x}_i\|_2^2 R_{\min}^{-2} \geq cC \geq \beta,$$

which is a contradiction. Therefore, $\underline{\rho}_{-j,r,i}^{(T)} < -\langle \mathbf{w}_{j,r}^{(0)}, \mathbf{x}_i \rangle$. By (5.5), we have $\underline{\rho}_{-j,r,i}^{(t)} \leq \underline{\rho}_{-j,r,i}^{(T)} < -\langle \mathbf{w}_{j,r}^{(0)}, \mathbf{x}_i \rangle$ for $t \geq T$. Therefore,

$$\langle \mathbf{w}_{j,r}^{(t)}, \mathbf{x}_i \rangle = \langle \mathbf{w}_{j,r}^{(0)}, \mathbf{x}_i \rangle + \rho_{j,r,i}^{(t)} < 0, \forall t \geq T.$$

Plugging this into (5.5) gives us

$$\underline{\rho}_{-j,r,i}^{(t+1)} = \underline{\rho}_{-j,r,i}^{(t)} + \frac{\eta}{nm} \cdot \ell_i^{(t)} \cdot \sigma'(\langle \mathbf{w}_{j,r}^{(t)}, \mathbf{x}_i \rangle) \cdot \|\mathbf{x}_i\|_2^2 = \underline{\rho}_{-j,r,i}^{(t)}, \forall t \geq T.$$

This completes the proof of the first half of the lemma about the activation pattern as well as the first two properties of $\rho_{j,r,i}^{(t)}$.

**Part 2.** Now we will show that

$$\lim_{t \to \infty} \overline{\rho}_{y_i,r,i}^{(t)}/\log t = m/|S_i^{(0)}|, \tag{F.5}$$

if $\langle \mathbf{w}_{y_i,r}^{(t)}, \mathbf{x}_i \rangle \geq 0$. By the activation pattern, we can get

$$\begin{aligned}
\overline{\rho}_{y_i,r,i}^{(t+1)} &= \overline{\rho}_{y_i,r,i}^{(t)} + \frac{\eta}{nm} \cdot |\ell_i'^{(t)}| \cdot \|\mathbf{x}_i\|_2^2, &\forall t \geq 0, &\quad \text{for } \langle \mathbf{w}_{y_i,i}^{(t)}, \mathbf{x}_i \rangle \geq 0, \\
\overline{\rho}_{y_i,r,i}^{(t)} &= 0, &\forall t \geq 0, &\quad \text{for } \langle \mathbf{w}_{y_i,i}^{(t)}, \mathbf{x}_i \rangle < 0, \\
\underline{\rho}_{-y_i,r,i}^{(t)} &= \underline{\rho}_{-y_i,r,i}^{(T)}, &\forall t \geq 0, &\quad \text{for } \langle \mathbf{w}_{-y_i,i}^{(t)}, \mathbf{x}_i \rangle \geq 0, \\
\underline{\rho}_{-y_i,r,i}^{(t)} &= 0, &\forall t \geq 0, &\quad \text{for } \langle \mathbf{w}_{-y_i,i}^{(t)}, \mathbf{x}_i \rangle < 0.
\end{aligned} \tag{F.6}$$

Given this activation pattern, we can get for $t \geq 0$ that

$$
\begin{aligned}
F_{y_i}(\mathbf{W}_{y_i}^{(t)}, \mathbf{x}_i) - F_{-y_i}(\mathbf{W}_{-y_i}^{(t)}, \mathbf{x}_i) &= \frac{1}{m}\sum_{r=1}^{m}\sigma(\langle \mathbf{w}_{y_i,r}^{(t)}, \mathbf{x}_i\rangle) - \frac{1}{m}\sum_{r=1}^{m}\sigma(\langle \mathbf{w}_{-y_i,r}^{(t)}, \mathbf{x}_i\rangle) \\
&\leq \frac{1}{m}\sum_{r \in S_i^{(0)}}\langle \mathbf{w}_{y_i,r}^{(t)}, \mathbf{x}_i\rangle \\
&= \frac{1}{m}\sum_{r \in S_i^{(0)}}[\langle \mathbf{w}_{y_i,r}^{(0)}, \mathbf{x}_i\rangle + \overline{\rho}_{y_i,r,i}^{(t)}] \\
&\leq \frac{1}{m}\sum_{r \in S_i^{(0)}}\overline{\rho}_{y_i,r,i}^{(t)} + \beta,
\end{aligned}
$$

and

$$
\begin{aligned}
F_{y_i}(\mathbf{W}_{y_i}^{(t)}, \mathbf{x}_i) - F_{-y_i}(\mathbf{W}_{-y_i}^{(t)}, \mathbf{x}_i) &= \frac{1}{m}\sum_{r=1}^{m}\sigma(\langle \mathbf{w}_{y_i,r}^{(t)}, \mathbf{x}_i\rangle) - \frac{1}{m}\sum_{r=1}^{m}\sigma(\langle \mathbf{w}_{-y_i,r}^{(t)}, \mathbf{x}_i\rangle) \\
&\geq \frac{1}{m}\sum_{r \in S_i^{(0)}}\langle \mathbf{w}_{y_i,r}^{(t)}, \mathbf{x}_i\rangle - \beta \\
&= \frac{1}{m}\sum_{r \in S_i^{(0)}}[\langle \mathbf{w}_{y_i,r}^{(0)}, \mathbf{x}_i\rangle + \overline{\rho}_{y_i,r,i}^{(t)}] - \beta \\
&\geq \frac{1}{m}\sum_{r \in S_i^{(0)}}\overline{\rho}_{y_i,r,i}^{(t)} - 2\beta.
\end{aligned}
$$

Therefore, we can get following upper and lower bounds for $|\ell_i'^{(t)}|$:

$$
|\ell_i'^{(t)}| \leq \exp\left(-\frac{1}{m}\sum_{r \in S_i^{(0)}}\overline{\rho}_{y_i,r,i}^{(t)} + 2\beta\right), \forall t \geq 0,
$$

$$
|\ell_i'^{(t)}| \geq \frac{1}{2}\exp\left(-\frac{1}{m}\sum_{r \in S_i^{(0)}}\overline{\rho}_{y_i,r,i}^{(t)} - \beta\right), \forall t \geq 0.
$$

And it follows that

$$
\frac{1}{m}\sum_{r \in S_i^{(0)}}\overline{\rho}_{y_i,r,i}^{(t+1)} \leq \frac{1}{m}\sum_{r \in S_i^{(0)}}\overline{\rho}_{y_i,r,i}^{(t)} + \frac{\eta\|\mathbf{x}_i\|_2^2 e^{2\beta}}{nm}\cdot\exp\left(-\frac{1}{m}\sum_{r \in S_i^{(0)}}\overline{\rho}_{y_i,r,i}^{(t)}\right), \forall t \geq 0,
$$

$$
\frac{1}{m}\sum_{r \in S_i^{(0)}}\overline{\rho}_{y_i,r,i}^{(t+1)} \geq \frac{1}{m}\sum_{r \in S_i^{(0)}}\overline{\rho}_{y_i,r,i}^{(t)} + \frac{\eta\|\mathbf{x}_i\|_2^2 e^{-\beta}}{2nm}\cdot\exp\left(-\frac{1}{m}\sum_{r \in S_i^{(0)}}\overline{\rho}_{y_i,r,i}^{(t)}\right), \forall t \geq 0.
$$

By leveraging Lemma H.1 as well as Lemma H.2 and taking

$$
x_t = \frac{1}{m}\sum_{r \in S_i^{(0)}}\overline{\rho}_{y_i,r,i}^{(t)},
$$

we can get

$$
\frac{1}{m}\sum_{r \in S_i^{(0)}}\overline{\rho}_{y_i,r,i}^{(t)} \leq \log\left(1 + \frac{\eta\|\mathbf{x}_i\|_2^2 e^{2\beta}}{nm}\exp\left(\frac{\eta\|\mathbf{x}_i\|_2^2 e^{2\beta}}{nm}\right)\cdot t\right) \leq \log\left(1 + \frac{2\eta\|\mathbf{x}_i\|_2^2 e^{-\beta}}{nm}\cdot t\right),
$$

$$\frac{1}{m} \sum_{r \in S_i^{(0)}} \overline{\rho}_{y_i,r,i}^{(t)} \geq \log\left(1 + \frac{\eta \|\mathbf{x}_i\|_2^2 e^{-\beta}}{2nm} \cdot t\right).$$

Therefore, we have

$$\lim_{t \to \infty} \frac{1}{m} \sum_{r \in S_i^{(0)}} \overline{\rho}_{y_i,r,i}^{(t)} / \log t = 1.$$

Since $\overline{\rho}_{y_i,r,i}^{(t)} = \overline{\rho}_{y_i,r',i}^{(t)}$ for any $r \neq r \in S_i^{(0)}$, we have

$$\lim_{t \to \infty} \overline{\rho}_{y_i,r,i}^{(t)} / \log t = m/|S_i^{(0)}|,$$

which completes the proof. $\qquad\qquad\qquad\qquad\qquad\qquad\qquad\qquad\qquad\qquad\qquad\qquad\quad$ $\square$

**Lemma F.3.** For two-layer ReLU neural network defined in (E.1), there exists mutually orthogonal data $\mathbf{x}_1, \cdots, \mathbf{x}_n$ such that stable rank of $\mathbf{W}_j^{(t)}$ will converge to $2 \pm o(1)$.

*Proof of Lemma F.3.* By (5.1), we have

$$\mathbf{w}_{j,r}^{(t)} = \mathbf{w}_{j,r}^{(0)} + \underbrace{\sum_{i=1}^n \rho_{j,r,i}^{(t)} \cdot \|\mathbf{x}_i\|_2^{-2} \cdot \mathbf{x}_i}_{:=\mathbf{v}_{j,r}^{(t)}}.$$

Given the definition of $\mathbf{v}_{j,r}^{(t)}$, we have the following representation of $\mathbf{v}_{j,r}^{(t)}$ and $\mathbf{V}_j^{(t)}$.

$$\mathbf{v}_{j,r}^{(t)} = \left[\rho_{j,r,1}^{(t)} \cdot \|\mathbf{x}_1\|_2^{-2} \cdots, \rho_{j,r,n}^{(t)} \cdot \|\mathbf{x}_n\|_2^{-2}\right] \cdot \begin{bmatrix} \mathbf{x}_1 \\ \vdots \\ \mathbf{x}_n \end{bmatrix},$$

$$\mathbf{V}_j^{(t)} = \begin{bmatrix} \rho_{j,r,1}^{(t)} \cdot \|\mathbf{x}_1\|_2^{-2} & \cdots & \rho_{j,r,n}^{(t)} \cdot \|\mathbf{x}_n\|_2^{-2} \\ \vdots & \ddots & \vdots \\ \rho_{j,m,1}^{(t)} \cdot \|\mathbf{x}_1\|_2^{-2} & \cdots & \rho_{j,m,n}^{(t)} \cdot \|\mathbf{x}_n\|_2^{-2} \end{bmatrix} \cdot \begin{bmatrix} \mathbf{x}_1 \\ \vdots \\ \mathbf{x}_n \end{bmatrix}.$$

Assume $n$ is an even number and $\mathbf{x}_1, \cdots, \mathbf{x}_{n/2}$ are with label $+1$ while $\mathbf{x}_{(n/2)+1}, \cdots, \mathbf{x}_n$ are with label $-1$. And we take $\mathbf{x}_1, \cdots, \mathbf{x}_n$ as $\mathbf{e}_1, \cdots, \mathbf{e}_n$. Given Lemma F.2, $\mathbf{W}_j^{(t)} = \mathbf{W}_j^{(0)} + \mathbf{V}_j^{(t)}$ and such selection of training data, we have

$$\lim_{t \to \infty} \frac{\mathbf{W}_{+1}^{(t)}}{\log t} = [\mathbf{A}_{m \times (n/2)}, \mathbf{0}_{m \times (n/2)}] \cdot [\mathbf{I}_n, \mathbf{0}_{n \times (d-n)}] = [\mathbf{A}_{m \times (n/2)}, \mathbf{0}_{m \times (d-(n/2))}],$$

$$\lim_{t \to \infty} \frac{\mathbf{W}_{-1}^{(t)}}{\log t} = [\mathbf{0}_{m \times (n/2)}, \mathbf{B}_{m \times (n/2)}] \cdot [\mathbf{I}_n, \mathbf{0}_{n \times (d-n)}] = [\mathbf{0}_{m \times (n/2)}, \mathbf{B}_{m \times (n/2)}, \mathbf{0}_{n \times (d-n)}],$$

$$\lim_{t \to \infty} \frac{\mathbf{W}^{(t)}}{\log t} = \begin{bmatrix} \mathbf{A}_{m \times (n/2)} & \mathbf{0}_{m \times (n/2)} & \mathbf{0}_{n \times (d-n)} \\ \mathbf{0}_{m \times (n/2)} & \mathbf{B}_{m \times (n/2)} & \mathbf{0}_{n \times (d-n)} \end{bmatrix}.$$

where

$$\mathbf{A}_{m \times (n/2)} = \underbrace{\begin{bmatrix} \mathbb{1}[\langle \mathbf{w}_{+1,1}^{(0)}, \mathbf{x}_1 \rangle \geq 0] & \cdots & \mathbb{1}[\langle \mathbf{w}_{+1,1}^{(0)}, \mathbf{x}_{n/2} \rangle \geq 0] \\ \vdots & \ddots & \vdots \\ \mathbb{1}[\langle \mathbf{w}_{+1,m}^{(0)}, \mathbf{x}_1 \rangle \geq 0] & \cdots & \mathbb{1}[\langle \mathbf{w}_{+1,m}^{(0)}, \mathbf{x}_{n/2} \rangle \geq 0] \end{bmatrix}}_{:=\mathbf{C}_{m \times (n/2)}} \cdot \text{diag} \begin{bmatrix} m/|S_1^{(0)}| \\ \vdots \\ m/|S_{n/2}^{(0)}| \end{bmatrix},$$

$$\mathbf{B}_{m\times(n/2)} = \underbrace{\begin{bmatrix} \mathbb{1}[\langle \mathbf{w}_{-1,1}^{(0)}, \mathbf{x}_{(n/2)+1}\rangle \geq 0] & \cdots & \mathbb{1}[\langle \mathbf{w}_{-1,1}^{(0)}, \mathbf{x}_n\rangle \geq 0] \\ \vdots & \ddots & \vdots \\ \mathbb{1}[\langle \mathbf{w}_{-1,m}^{(0)}, \mathbf{x}_{(n/2)+1}\rangle \geq 0] & \cdots & \mathbb{1}[\langle \mathbf{w}_{-1,m}^{(0)}, \mathbf{x}_n\rangle \geq 0] \end{bmatrix}}_{:=\mathbf{D}_{m\times(n/2)}} \cdot \mathrm{diag}\begin{bmatrix} m/|S_{(n/2)+1}^{(0)}| \\ \vdots \\ m/|S_n^{(0)}| \end{bmatrix}.$$

Then, we can get the stable rank limits as follows:

$$\lim_{t\to\infty} \frac{\|\mathbf{W}_{+1}^{(t)}\|_F^2}{\|\mathbf{W}_{+1}^{(t)}\|_2^2} = \frac{\|\mathbf{A}\|_F^2}{\|\mathbf{A}\|_2^2},$$

$$\lim_{t\to\infty} \frac{\|\mathbf{W}_{-1}^{(t)}\|_F^2}{\|\mathbf{W}_{-1}^{(t)}\|_2^2} = \frac{\|\mathbf{B}\|_F^2}{\|\mathbf{B}\|_2^2},$$

$$\lim_{t\to\infty} \frac{\|\mathbf{W}^{(t)}\|_F^2}{\|\mathbf{W}^{(t)}\|_2^2} = \frac{\|\mathbf{A}\|_F^2 + \|\mathbf{B}\|_F^2}{(\max\{\|\mathbf{A}\|_2, \|\mathbf{B}\|_2\})^2}.$$

Since $\mathbf{x}_1 = \mathbf{e}_1, \cdots, \mathbf{x}_n = \mathbf{e}_n$, we can get

$$\mathbb{1}[\langle \mathbf{w}_{j,r}^{(0)}, \mathbf{x}_i\rangle \geq 0] = \mathbb{1}[[\mathbf{w}_{j,r}^{(0)}]_i \geq 0].$$

Therefore, the entries of matrix $\mathbf{C}$ and matrix $\mathbf{D}$ can be regarded as i.i.d. random variables taking 0 or 1 with equal probability. For $\|\mathbf{A}\|_F$ and $\|\mathbf{B}\|_F$, we have

$$\|\mathbf{A}\|_F^2 = \sum_{r=1}^m \sum_{i=1}^{n/2} \mathbb{1}[[\mathbf{w}_{+1,r}^{(0)}]_i \geq 0] \cdot (m/|S_i^{(0)}|)^2,$$

$$\|\mathbf{B}\|_F^2 = \sum_{r=1}^m \sum_{i=(n/2)+1}^n \mathbb{1}[[\mathbf{w}_{-1,r}^{(0)}]_i \geq 0] \cdot (m/|S_i^{(0)}|)^2.$$

By Lemma B.2, we have with probability at least $1 - \delta$ that $0.4m \leq |S_i^{(0)}| \leq 0.6m$. By Hoeffding's inequality, we have with probability at least $1 - 2\delta$ that

$$\left| \|\mathbf{A}\|_F^2 - \frac{m}{2}\sum_{i=1}^{n/2}(m/|S_i^{(0)}|)^2 \right| \leq \sqrt{\frac{m\log(2/\delta)}{2}\sum_{i=1}^{n/2}(m/|S_i^{(0)}|)^4} \leq \sqrt{\frac{625mn\log(2/\delta)}{32}},$$

$$\left| \|\mathbf{B}\|_F^2 - \frac{m}{2}\sum_{i=(n/2)+1}^n (m/|S_i^{(0)}|)^2 \right| \leq \sqrt{\frac{m\log(2/\delta)}{2}\sum_{i=(n/2)+1}^n (m/|S_i^{(0)}|)^4} \leq \sqrt{\frac{625mn\log(2/\delta)}{32}}.$$

Next, we estimate $\|\mathbf{A}\|_2$ and $\|\mathbf{B}\|_2$. Let $\mathbf{A} = \widetilde{\mathbf{A}} + \mathbb{E}[\mathbf{A}]$ and $\mathbf{B} = \widetilde{\mathbf{B}} + \mathbb{E}[\mathbf{B}]$. Assume $\mathbf{G}$ be the $m \times (n/2)$ matrix with all entries equal to $1/2$. Then,

$$\mathbb{E}[\mathbf{A}] = \mathbf{G} \cdot \mathrm{diag}\underbrace{\begin{bmatrix} m/|S_1^{(0)}| \\ \vdots \\ m/|S_{n/2}^{(0)}| \end{bmatrix}}_{:=\mathbf{a}}, \mathbb{E}[\mathbf{B}] = \mathbf{G} \cdot \mathrm{diag}\underbrace{\begin{bmatrix} m/|S_{(n/2)+1}^{(0)}| \\ \vdots \\ m/|S_n^{(0)}| \end{bmatrix}}_{:=\mathbf{b}}.$$

And the entries of matrix $\widetilde{\mathbf{A}}$ and matrix $\widetilde{\mathbf{B}}$ are independent, mean zero, sub-gaussian random variables. By Lemma H.3, we have with probability at least $1 - \delta$ that

$$\|\widetilde{\mathbf{A}}\|_2 \leq \frac{C}{2}\big(\sqrt{m} + \sqrt{n} + \sqrt{\log(2/\delta)}\big),$$

$$\|\widetilde{\mathbf{B}}\|_2 \leq \frac{C}{2}\big(\sqrt{m} + \sqrt{n} + \sqrt{\log(2/\delta)}\big),$$

where $C$ is a constant. Let $\mathbf{1}_k$ denote the row vector with $k$ entries equal to 1. Then for $\mathbb{E}[\mathbf{A}]$ and $\mathbb{E}[\mathbf{B}]$, we have

$$
\begin{aligned}
\|\mathbb{E}[\mathbf{A}]\|_2 &= \max_{\mathbf{x} \in S^{\frac{n}{2}-1}} \|\mathbf{G}\mathrm{diag}(\mathbf{a})\mathbf{x}\|_2 \\
&= \max_{\mathbf{x} \in S^{\frac{n}{2}-1}} \frac{1}{2}\|\mathbf{1}_m^\top \mathbf{1}_{\frac{n}{2}} \mathrm{diag}(\mathbf{a})\mathbf{x}\|_2 \\
&= \max_{\mathbf{x} \in S^{\frac{n}{2}-1}} \frac{\sqrt{m}}{2}|\mathbf{1}_{\frac{n}{2}} \mathrm{diag}(\mathbf{a})\mathbf{x}| \\
&= \max_{\mathbf{x} \in S^{\frac{n}{2}-1}} \frac{\sqrt{m}}{2}|\mathbf{a}^\top \mathbf{x}| \\
&= \frac{\sqrt{m}\|\mathbf{a}\|_2}{2},
\end{aligned}
$$

and

$$
\|\mathbb{E}[\mathbf{B}]\|_2 = \frac{\sqrt{m}\|\boldsymbol{b}\|_2}{2}.
$$

By triangle inequality, we have

$$
\|\mathbf{A}\|_2 \geq \|\mathbf{C}\|_2 - \|\widetilde{\mathbf{A}}\|_2 \geq (\sqrt{m}\|\mathbf{a}\|_2)/2 - \frac{C}{2}\left(\sqrt{m} + \sqrt{n} + \sqrt{\log(2/\delta)}\right),
$$

$$
\|\mathbf{A}\|_2 \leq \|\mathbf{C}\|_2 + \|\widetilde{\mathbf{A}}\|_2 \leq (\sqrt{m}\|\mathbf{a}\|_2)/2 + \frac{C}{2}\left(\sqrt{m} + \sqrt{n} + \sqrt{\log(2/\delta)}\right),
$$

$$
\|\mathbf{B}\|_2 \geq \|\mathbf{C}\|_2 - \|\widetilde{\mathbf{B}}\|_2 \geq (\sqrt{m}\|\boldsymbol{b}\|_2)/2 - \frac{C}{2}\left(\sqrt{m} + \sqrt{n} + \sqrt{\log(2/\delta)}\right),
$$

$$
\|\mathbf{B}\|_2 \leq \|\mathbf{C}\|_2 + \|\widetilde{\mathbf{B}}\|_2 \leq (\sqrt{m}\|\boldsymbol{b}\|_2)/2 + \frac{C}{2}\left(\sqrt{m} + \sqrt{n} + \sqrt{\log(2/\delta)}\right).
$$

Notice that $\|\mathbf{a}\|_2^2 = \Theta(n)$ and $\|\boldsymbol{b}\|_2^2 = \Theta(n)$, then with probability at least $1 - 2\delta$, we have

$$
\frac{\|\mathbf{A}\|_F^2}{\|\mathbf{A}\|_2^2} \leq \frac{m\|\mathbf{a}\|_2^2/2 + \sqrt{625mn\log(2/\delta)/32}}{\left(\sqrt{m}\|\mathbf{a}\|_2/2 - \frac{C}{2}\left(\sqrt{m} + \sqrt{n} + \sqrt{\log(2/\delta)}\right)\right)^2} = 2 + o(1),
$$

$$
\frac{\|\mathbf{A}\|_F^2}{\|\mathbf{A}\|_2^2} \geq \frac{m\|\mathbf{a}\|_2^2/2 - \sqrt{625mn\log(2/\delta)/32}}{\left(\sqrt{m}\|\mathbf{a}\|_2/2 + \frac{C}{2}\left(\sqrt{m} + \sqrt{n} + \sqrt{\log(2/\delta)}\right)\right)^2} = 2 - o(1),
$$

$$
\frac{\|\mathbf{B}\|_F^2}{\|\mathbf{B}\|_2^2} \geq \frac{m\|\boldsymbol{b}\|_2^2/2 + \sqrt{625mn\log(2/\delta)/32}}{\left(\sqrt{m}\|\boldsymbol{b}\|_2/2 - \frac{C}{2}\left(\sqrt{m} + \sqrt{n} + \sqrt{\log(2/\delta)}\right)\right)^2} = 2 + o(1),
$$

$$
\frac{\|\mathbf{B}\|_F^2}{\|\mathbf{B}\|_2^2} \geq \frac{m\|\boldsymbol{b}\|_2^2/2 - \sqrt{625mn\log(2/\delta)/32}}{\left(\sqrt{m}\|\boldsymbol{b}\|_2/2 + \frac{C}{2}\left(\sqrt{m} + \sqrt{n} + \sqrt{\log(2/\delta)}\right)\right)^2} = 2 - o(1).
$$

This leads to

$$
\lim_{t \to \infty} \frac{\|\mathbf{W}_{+1}^{(t)}\|_F^2}{\|\mathbf{W}_{+1}^{(t)}\|_2^2} = \frac{\|\mathbf{A}\|_F^2}{\|\mathbf{A}\|_2^2} = 2 \pm o(1),
$$

$$
\lim_{t \to \infty} \frac{\|\mathbf{W}_{-1}^{(t)}\|_F^2}{\|\mathbf{W}_{-1}^{(t)}\|_2^2} = \frac{\|\mathbf{B}\|_F^2}{\|\mathbf{B}\|_2^2} = 2 \pm o(1).
$$

For $\mathbf{W}^{(t)}$, we have the following lower bound

$$
\begin{aligned}
\frac{\|\mathbf{A}\|_F^2 + \|\mathbf{B}\|_F^2}{(\max\{\|\mathbf{A}\|_2, \|\mathbf{B}\|_2\})^2} &\geq \frac{m(\|\mathbf{a}\|_2^2 + \|\boldsymbol{b}\|_2^2)/2 - \sqrt{625mn\log(2/\delta)/8}}{\left(\sqrt{m}\max\{\|\mathbf{a}\|_2, \|\boldsymbol{b}\|_2\}/2 + \frac{C}{2}\left(\sqrt{m} + \sqrt{n} + \sqrt{\log(2/\delta)}\right)\right)^2} \\
&\geq (2 - o(1)) \cdot \frac{\|\mathbf{a}\|_2^2 + \|\boldsymbol{b}\|_2^2}{(\max\{\|\mathbf{a}\|_2, \|\boldsymbol{b}\|_2\})^2}
\end{aligned}
$$

$$\geq \frac{16}{9} - o(1),$$

where the third inequality is by $(5/3)\sqrt{n} \leq \|\mathbf{a}\|_2 \leq (5/2)\sqrt{n}$ and $(5/3)\sqrt{n} \leq \|\boldsymbol{b}\|_2 \leq (5/2)\sqrt{n}$ due to $0.4m \leq |S_i^{(0)}| \leq 0.6m$. And

$$\frac{\|\mathbf{A}\|_F^2 + \|\mathbf{B}\|_F^2}{(\max\{\|\mathbf{A}\|_2, \|\mathbf{B}\|_2\})^2} \leq \frac{m(\|\mathbf{a}\|_2^2 + \|\boldsymbol{b}\|_2^2)/2 + \sqrt{625mn\log(2/\delta)/8}}{\left(\sqrt{m}\max\{\|\mathbf{a}\|_2, \|\boldsymbol{b}\|_2\}/2 - \frac{C}{2}\left(\sqrt{m} + \sqrt{n} + \sqrt{\log(2/\delta)}\right)\right)^2}$$

$$\leq (2 + o(1)) \cdot \frac{\|\mathbf{a}\|_2^2 + \|\boldsymbol{b}\|_2^2}{(\max\{\|\mathbf{a}\|_2, \|\boldsymbol{b}\|_2\})^2}$$

$$\leq 9 + o(1),$$

where the third inequality is by $(5/3)\sqrt{n} \leq \|\mathbf{a}\|_2 \leq (5/2)\sqrt{n}$ and $(5/3)\sqrt{n} \leq \|\boldsymbol{b}\|_2 \leq (5/2)\sqrt{n}$ due to $0.4m \leq |S_i^{(0)}| \leq 0.6m$. Therefore,

$$\lim_{t\to\infty} \frac{\|\mathbf{W}\|_F^2}{\|\mathbf{W}\|_2^2} = \frac{\|\mathbf{A}\|_F^2 + \|\mathbf{B}\|_F^2}{(\max\{\|\mathbf{A}\|_2, \|\mathbf{B}\|_2\})^2} \in [16/9 - o(1), 9 + o(1)].$$

$\square$

## G  Margin Results and Loss Convergence

In this section, we prove the convergence rate of training loss as well as the increasing rate of margin for both two-layer ReLU and leaky ReLU networks defined as

$$f(\mathbf{W}^{(t)}, \mathbf{x}) = F_{+1}(\mathbf{W}_{+1}^{(t)}, \mathbf{x}) - F_{-1}(\mathbf{W}_{-1}^{(t)}, \mathbf{x})$$

$$= \frac{1}{m} \sum_{r=1}^{m} \sigma(\langle \mathbf{w}_{+1,r}^{(t)}, \mathbf{x} \rangle) - \frac{1}{m} \sum_{r=1}^{m} \sigma(\langle \mathbf{w}_{-1,r}^{(t)}, \mathbf{x} \rangle), \tag{G.1}$$

$$\sigma \in \{\text{ReLU}, \text{leaky ReLU}\}.$$

We first prove the following auxiliary lemma.

**Lemma G.1.** For both two-layer leaky ReLU and ReLU neural networks defined in (G.1), the following properties hold for any $t \geq 0$:

- $y_i f(\mathbf{W}^{(t)}, \mathbf{x}_i) \geq -c$ for any $i \in [n]$ where $c$ is a positive constant.
- $y_i f(\mathbf{W}^{(t)}, \mathbf{x}_i) - y_k f(\mathbf{W}^{(t)}, \mathbf{x}_k) \leq C_1$ for any $i, k \in [n]$ where $C_1$ is a constant.
- $\ell_i'^{(t)}/\ell_k'^{(t)} \leq C_2$ for any $i, k \in [n]$ where $C_2$ is a constant.
- $S_i^{(t)} \subseteq S_i^{(t+1)}$ for any $i \in [n]$, where $S_i^{(t)} := \{r \in [m] : \langle \mathbf{w}_{y_i,r}^{(t)}, \mathbf{x}_i \rangle \geq 0\}$.

*Proof of Lemma G.1.* We prove this lemma by induction. When $t = 0$, since

$$|y_i f(\mathbf{W}^{(0)}, \mathbf{x}_i)| = \left| \sum_j j y_i F_j(\mathbf{W}_j^{(0)}, \mathbf{x}_i) \right|$$

$$= \left| \sum_j j y_i \cdot \frac{1}{m} \sum_{r=1}^{m} \sigma(\langle \mathbf{w}_{j,r}^{(0)}, \mathbf{x}_i \rangle) \right|$$

$$\leq \sum_j \frac{1}{m} \sum_{r=1}^{m} |\sigma(\langle \mathbf{w}_{j,r}^{(0)}, \mathbf{x}_i \rangle)|$$

$$\leq \sum_j \frac{1}{m} \sum_{r=1}^{m} |\langle \mathbf{w}_{j,r}^{(0)}, \mathbf{x}_i \rangle|$$

$$\leq 2\beta,$$

the first bullet holds as long as $c \geq 2\beta$. We also have

$$y_i f(\mathbf{W}^{(0)}, \mathbf{x}_i) - y_k f(\mathbf{W}^{(0)}, \mathbf{x}_k) \leq |y_i f(\mathbf{W}^{(t)}, \mathbf{x}_i)| + |y_k f(\mathbf{W}^{(0)}, \mathbf{x}_k)| \leq 4\beta,$$

which verifies the second bullet at time $t = 0$ as long as $C_1 \geq 4\beta$. This leads to

$$
\begin{aligned}
\frac{\ell_i'^{(0)}}{\ell_k'^{(0)}} &= \frac{1 + \exp(y_k f(\mathbf{W}^{(0)}, \mathbf{x}_k))}{1 + \exp(y_i f(\mathbf{W}^{(0)}, \mathbf{x}_i))} \\
&\leq 2\big(1 + \exp(y_k f(\mathbf{W}^{(0)}, \mathbf{x}_k) - y_i f(\mathbf{W}^{(0)}, \mathbf{x}_i))\big) \\
&\leq 2(1 + \exp(C_1)) \\
&\leq C_2,
\end{aligned}
$$

as long as $C_2 \geq 2(1 + \exp(C_1))$. For any $r \in S_i^{(0)}$, we have

$$
\begin{aligned}
\langle \mathbf{w}_{y_i,r}^{(1)}, \mathbf{x}_i \rangle &= \langle \mathbf{w}_{y_i,r}^{(0)}, \mathbf{x}_i \rangle + \frac{\eta}{nm} \sum_{i'=1}^{n} |\ell_{i'}'^{(0)}| \cdot \sigma'(\langle \mathbf{w}_{y_i,r}^{(0)}, \mathbf{x}_i \rangle) \cdot \langle y_{i'} \mathbf{x}_{i'}, y_i \mathbf{x}_i \rangle \\
&= \langle \mathbf{w}_{y_i,r}^{(0)}, \mathbf{x}_i \rangle + \frac{\eta}{nm} \cdot |\ell_i'^{(0)}| \cdot \|\mathbf{x}_i\|_2^2 + \frac{\eta}{nm} \sum_{i' \neq i} |\ell_{i'}'^{(0)}| \cdot \sigma'(\langle \mathbf{w}_{y_i,r}^{(0)}, \mathbf{x}_i \rangle) \cdot \langle y_{i'} \mathbf{x}_{i'}, y_i \mathbf{x}_i \rangle \\
&\geq \langle \mathbf{w}_{y_i,r}^{(0)}, \mathbf{x}_i \rangle + \frac{\eta}{nm} \cdot |\ell_i'^{(0)}| \cdot \|\mathbf{x}_i\|_2^2 - \frac{\eta}{nm} \sum_{i' \neq i} |\ell_{i'}'^{(0)}| \cdot |\langle \mathbf{x}_{i'}, \mathbf{x}_i \rangle| \\
&\geq \langle \mathbf{w}_{y_i,r}^{(0)}, \mathbf{x}_i \rangle \geq 0,
\end{aligned}
$$

where the second equality is by $\langle \mathbf{w}_{y_i,r}^{(0)}, \mathbf{x}_i \rangle \geq 0$; the first inequality is by triangle inequality; the second inequality is by $|\ell_{i'}'^{(0)}|/|\ell_i'^{(0)}| \leq C_2$ and the condition that $R_{\min}^2 \geq Cnp$, $C$ is a sufficiently large constant. This verifies the fourth bullet at time $t = 0$.

Now suppose there exists time $\widetilde{t} \geq 0$ such that these four hypotheses hold for any $0 \leq t \leq \widetilde{t}$. We aim to prove that these conditions also hold for $t = \widetilde{t} + 1$. We first prove that $y_i f(\mathbf{W}^{(\widetilde{t}+1)}, \mathbf{x}_i) \geq y_i f(\mathbf{W}^{(\widetilde{t})}, \mathbf{x}_i)$. We have

$$
\begin{aligned}
&y_i f(\mathbf{W}^{(\widetilde{t}+1)}, \mathbf{x}_i) - y_i f(\mathbf{W}^{(\widetilde{t})}, \mathbf{x}_i) \\
&= \sum_j y_i j \Big( F_j(\mathbf{W}_j^{(\widetilde{t}+1)}, \mathbf{x}_i) - F_j(\mathbf{W}_j^{(\widetilde{t})}, \mathbf{x}_i) \Big) \\
&= \sum_j y_i j \cdot \frac{1}{m} \sum_{r=1}^{m} \Big( \sigma(\langle \mathbf{w}_{j,r}^{(\widetilde{t}+1)}, \mathbf{x}_i \rangle) - \sigma(\langle \mathbf{w}_{j,r}^{(\widetilde{t})}, \mathbf{x}_i \rangle) \Big) \\
&= \sum_j y_i j \cdot \frac{1}{m} \sum_{r=1}^{m} \frac{\sigma(\langle \mathbf{w}_{j,r}^{(\widetilde{t}+1)}, \mathbf{x}_i \rangle) - \sigma(\langle \mathbf{w}_{j,r}^{(\widetilde{t})}, \mathbf{x}_i \rangle)}{\langle \mathbf{w}_{j,r}^{(\widetilde{t}+1)}, \mathbf{x}_i \rangle - \langle \mathbf{w}_{j,r}^{(\widetilde{t})}, \mathbf{x}_i \rangle} \cdot \langle -\eta \cdot \nabla_{\mathbf{w}_{j,r}} L_S(\mathbf{W}^{(\widetilde{t})}), \mathbf{x}_i \rangle \\
&= \sum_j y_i j \cdot \frac{1}{m} \sum_{r=1}^{m} \frac{\sigma(\langle \mathbf{w}_{j,r}^{(\widetilde{t}+1)}, \mathbf{x}_i \rangle) - \sigma(\langle \mathbf{w}_{j,r}^{(\widetilde{t})}, \mathbf{x}_i \rangle)}{\langle \mathbf{w}_{j,r}^{(\widetilde{t}+1)}, \mathbf{x}_i \rangle - \langle \mathbf{w}_{j,r}^{(\widetilde{t})}, \mathbf{x}_i \rangle} \cdot \Big\langle \frac{\eta}{nm} \sum_{i'=1}^{n} |\ell_{i'}'^{(\widetilde{t})}| \cdot \sigma'(\langle \mathbf{w}_{j,r}^{(\widetilde{t})}, \mathbf{x}_{i'} \rangle) \cdot j y_{i'} \mathbf{x}_{i'}, \mathbf{x}_i \Big\rangle \\
&= \sum_j \frac{1}{m} \sum_{r=1}^{m} \frac{\sigma(\langle \mathbf{w}_{j,r}^{(\widetilde{t}+1)}, \mathbf{x}_i \rangle) - \sigma(\langle \mathbf{w}_{j,r}^{(\widetilde{t})}, \mathbf{x}_i \rangle)}{\langle \mathbf{w}_{j,r}^{(\widetilde{t}+1)}, \mathbf{x}_i \rangle - \langle \mathbf{w}_{j,r}^{(\widetilde{t})}, \mathbf{x}_i \rangle} \cdot \frac{\eta}{nm} |\ell_i'^{(\widetilde{t})}| \cdot \sigma'(\langle \mathbf{w}_{j,r}^{(\widetilde{t})}, \mathbf{x}_i \rangle) \cdot \|\mathbf{x}_i\|_2^2 \\
&\quad + \sum_j \frac{1}{m} \sum_{r=1}^{m} \frac{\sigma(\langle \mathbf{w}_{j,r}^{(\widetilde{t}+1)}, \mathbf{x}_i \rangle) - \sigma(\langle \mathbf{w}_{j,r}^{(\widetilde{t})}, \mathbf{x}_i \rangle)}{\langle \mathbf{w}_{j,r}^{(\widetilde{t}+1)}, \mathbf{x}_i \rangle - \langle \mathbf{w}_{j,r}^{(\widetilde{t})}, \mathbf{x}_i \rangle} \cdot \frac{\eta}{nm} \sum_{i' \neq i} |\ell_{i'}'^{(\widetilde{t})}| \cdot \sigma'(\langle \mathbf{w}_{j,r}^{(\widetilde{t})}, \mathbf{x}_{i'} \rangle) \cdot \langle y_{i'} \mathbf{x}_{i'}, y_i \mathbf{x}_i \rangle.
\end{aligned}
$$

By the fourth induction hypothesis at time $t = \widetilde{t}$, we have $S_i^{(\widetilde{t}+1)} \subseteq S_i^{(\widetilde{t})}$ and hence

$$\frac{\sigma(\langle \mathbf{w}_{j,r}^{(\widetilde{t}+1)}, \mathbf{x}_i \rangle) - \sigma(\langle \mathbf{w}_{j,r}^{(\widetilde{t})}, \mathbf{x}_i \rangle)}{\langle \mathbf{w}_{j,r}^{(\widetilde{t}+1)}, \mathbf{x}_i \rangle - \langle \mathbf{w}_{j,r}^{(\widetilde{t})}, \mathbf{x}_i \rangle} = \frac{\langle \mathbf{w}_{j,r}^{(\widetilde{t}+1)}, \mathbf{x}_i \rangle - \langle \mathbf{w}_{j,r}^{(\widetilde{t})}, \mathbf{x}_i \rangle}{\langle \mathbf{w}_{j,r}^{(\widetilde{t}+1)}, \mathbf{x}_i \rangle - \langle \mathbf{w}_{j,r}^{(\widetilde{t})}, \mathbf{x}_i \rangle} = 1, \tag{G.2}$$

for $j = y_i$ and $r \in S_i^{(t)}$. For $\sigma \in \{\text{ReLU}, \text{leaky ReLU}\}$, $\sigma$ is non-decreasing and 1-Lipschitz continuous, which gives

$$0 \leq \frac{\sigma(\langle \mathbf{w}_{j,r}^{(\widetilde{t}+1)}, \mathbf{x}_i \rangle) - \sigma(\langle \mathbf{w}_{j,r}^{(\widetilde{t})}, \mathbf{x}_i \rangle)}{\langle \mathbf{w}_{j,r}^{(\widetilde{t}+1)}, \mathbf{x}_i \rangle - \langle \mathbf{w}_{j,r}^{(\widetilde{t})}, \mathbf{x}_i \rangle} \leq 1. \tag{G.3}$$

Then, we have

$$y_i f(\mathbf{W}^{(\widetilde{t}+1)}, \mathbf{x}_i) - y_i f(\mathbf{W}^{(\widetilde{t})}, \mathbf{x}_i) \geq \frac{\eta}{nm^2} \sum_{r \in S_i^{(\widetilde{t})}} |\ell_i'^{(\widetilde{t})}| \cdot \|\mathbf{x}_i\|_2^2 - \frac{\eta}{nm^2} \sum_j \sum_{r=1}^m \sum_{i' \neq i} |\ell_{i'}'^{(\widetilde{t})}| \cdot |\langle \mathbf{x}_{i'}, \mathbf{x}_i \rangle|$$

$$\geq \frac{\eta}{2nm^2} \sum_{r \in S_i^{(\widetilde{t})}} |\ell_i'^{(\widetilde{t})}| \cdot \|\mathbf{x}_i\|_2^2$$

$$= \frac{\eta |S_i^{(\widetilde{t})}|}{2nm^2} \cdot |\ell_i'^{(\widetilde{t})}| \cdot \|\mathbf{x}_i\|_2^2$$

$$\geq \frac{\eta}{5nm} \cdot |\ell_i'^{(\widetilde{t})}| \cdot \|\mathbf{x}_i\|_2^2,$$

where the first inequality is by (G.2), (G.3), $\sigma' \in [0, 1]$ and triangle inequality; the second inequality is by $|\ell_{i'}'^{(\widetilde{t})}|/|\ell_i'^{(\widetilde{t})}| \leq C_2$, $|S_i^{(\widetilde{t})}| \geq |S_i^{(0)}| \geq 0.4m$ and the condition that $R_{\min}^2 \geq Cnp$, $C$ is a sufficiently large constant. And

$$y_i f(\mathbf{W}^{(\widetilde{t}+1)}, \mathbf{x}_i) - y_i f(\mathbf{W}^{(\widetilde{t})}, \mathbf{x}_i)$$

$$\leq \sum_j \sum_{r=1}^m \frac{\eta}{nm^2} \sum_{r \in S_i^{(\widetilde{t})}} |\ell_i'^{(\widetilde{t})}| \cdot \|\mathbf{x}_i\|_2^2 + \frac{\eta}{nm^2} \sum_j \sum_{r=1}^m \sum_{i' \neq i} |\ell_{i'}'^{(\widetilde{t})}| \cdot |\langle \mathbf{x}_{i'}, \mathbf{x}_i \rangle|$$

$$\leq \frac{3\eta}{2nm^2} \sum_j \sum_{r=1}^m |\ell_i'^{(\widetilde{t})}| \cdot \|\mathbf{x}_i\|_2^2$$

$$= \frac{3\eta}{nm} \cdot |\ell_i'^{(\widetilde{t})}| \cdot \|\mathbf{x}_i\|_2^2,$$

where the first inequality is by (G.3), $\sigma' \in [0, 1]$ and triangle inequality; the second inequality is by $|\ell_{i'}'^{(\widetilde{t})}|/|\ell_i'^{(\widetilde{t})}| \leq C_2$, $|S_i^{(\widetilde{t})}| \geq |S_i^{(0)}| \geq 0.4m$ and the condition that $R_{\min}^2 \geq Cnp$, $C$ is a sufficiently large constant. Now, we obtain

$$y_i f(\mathbf{W}^{(\widetilde{t}+1)}, \mathbf{x}_i) \geq y_i f(\mathbf{W}^{(\widetilde{t})}, \mathbf{x}_i) + \frac{\eta}{5nm} \cdot |\ell_i'^{(\widetilde{t})}| \cdot \|\mathbf{x}_i\|_2^2, \tag{G.4}$$

$$y_i f(\mathbf{W}^{(\widetilde{t}+1)}, \mathbf{x}_i) \leq y_i f(\mathbf{W}^{(\widetilde{t})}, \mathbf{x}_i) + \frac{3\eta}{nm} \cdot |\ell_i'^{(\widetilde{t})}| \cdot \|\mathbf{x}_i\|_2^2, \tag{G.5}$$

which implies that $y_i f(\mathbf{W}^{(\widetilde{t}+1)}, \mathbf{x}_i) \geq y_i f(\mathbf{W}^{(\widetilde{t})}, \mathbf{x}_i) \geq -c$. This verifies the first bullet at time $t = \widetilde{t} + 1$. By subtracting (G.5) from (G.4), we have

$$y_k f(\mathbf{W}^{(\widetilde{t}+1)}, \mathbf{x}_k) - y_i f(\mathbf{W}^{(\widetilde{t}+1)}, \mathbf{x}_i)$$

$$\leq y_k f(\mathbf{W}^{(\widetilde{t})}, \mathbf{x}_k) - y_i f(\mathbf{W}^{(\widetilde{t})}, \mathbf{x}_i) + \frac{3\eta}{nm} \cdot |\ell_k'^{(\widetilde{t})}| \cdot \|\mathbf{x}_k\|_2^2 - \frac{\eta}{5nm} \cdot |\ell_i'^{(\widetilde{t})}| \cdot \|\mathbf{x}_i\|_2^2.$$

If $|\ell_i'^{(\tilde{t})}|/|\ell_k'^{(\tilde{t})}| \geq 15R^2$, then $\frac{3\eta}{nm} \cdot |\ell_k'^{(\tilde{t})}| \cdot \|\mathbf{x}_k\|_2^2 \leq \frac{\eta}{5nm} \cdot |\ell_i'^{(\tilde{t})}| \cdot \|\mathbf{x}_i\|_2^2$ and hence

$$y_k f(\mathbf{W}^{(\tilde{t}+1)}, \mathbf{x}_k) - y_i f(\mathbf{W}^{(\tilde{t}+1)}, \mathbf{x}_i) \leq y_k f(\mathbf{W}^{(\tilde{t})}, \mathbf{x}_k) - y_i f(\mathbf{W}^{(\tilde{t})}, \mathbf{x}_i) \leq C_1.$$

If $|\ell_i'^{(\tilde{t})}|/|\ell_k'^{(\tilde{t})}| < 15R^2$, then by Lemma H.5

$$y_k f(\mathbf{W}^{(\tilde{t})}, \mathbf{x}_k) - y_i f(\mathbf{W}^{(\tilde{t})}, \mathbf{x}_i) \leq \log(4|\ell_i'^{(\tilde{t})}|/|\ell_k'^{(\tilde{t})}|) < \log(60R^2),$$

and hence

$$y_k f(\mathbf{W}^{(\tilde{t}+1)}, \mathbf{x}_k) - y_i f(\mathbf{W}^{(\tilde{t}+1)}, \mathbf{x}_i) \leq y_k f(\mathbf{W}^{(\tilde{t})}, \mathbf{x}_k) - y_i f(\mathbf{W}^{(\tilde{t})}, \mathbf{x}_i) + \frac{3\eta}{nm} \cdot |\ell_k'^{(\tilde{t})}| \cdot \|\mathbf{x}_k\|_2^2$$
$$\leq \log(60R^2) + 1.$$

Combining the two cases, we have

$$y_k f(\mathbf{W}^{(\tilde{t}+1)}, \mathbf{x}_k) - y_i f(\mathbf{W}^{(\tilde{t}+1)}, \mathbf{x}_i) \leq C_1,$$

as long as $C_1 \geq \max\{4\beta, \log(60R^2) + 1\}$, which verifies the fourth bullet at time $t = \tilde{t} + 1$. By Lemma H.5, this leads to

$$\frac{\ell_i'^{(\tilde{t}+1)}}{\ell_k'^{(\tilde{t}+1)}} = \frac{1 + \exp(y_k f(\mathbf{W}^{(\tilde{t}+1)}, \mathbf{x}_k))}{1 + \exp(y_i f(\mathbf{W}^{(\tilde{t}+1)}, \mathbf{x}_i))}$$
$$\leq 2\big(1 + \exp(y_k f(\mathbf{W}^{(\tilde{t}+1)}, \mathbf{x}_k) - y_i f(\mathbf{W}^{(\tilde{t}+1)}, \mathbf{x}_i))\big)$$
$$\leq 2(1 + \exp(C_1))$$
$$\leq C_2,$$

as long as $C_2 \geq 2(1 + \exp(C_1))$. For any $r \in S_i^{(\tilde{t}+1)}$, we have

$$\langle \mathbf{w}_{y_i,r}^{(\tilde{t}+2)}, \mathbf{x}_i \rangle = \langle \mathbf{w}_{y_i,r}^{(\tilde{t}+1)}, \mathbf{x}_i \rangle + \frac{\eta}{nm} \sum_{i'=1}^{n} |\ell_{i'}'^{(\tilde{t}+1)}| \cdot \sigma'(\langle \mathbf{w}_{y_i,r}^{(\tilde{t}+1)}, \mathbf{x}_i \rangle) \cdot \langle y_{i'} \mathbf{x}_{i'}, y_i \mathbf{x}_i \rangle$$
$$= \langle \mathbf{w}_{y_i,r}^{(\tilde{t}+1)}, \mathbf{x}_i \rangle + \frac{\eta}{nm} \cdot |\ell_i'^{(\tilde{t}+1)}| \cdot \|\mathbf{x}_i\|_2^2$$
$$\quad + \frac{\eta}{nm} \sum_{i' \neq i} |\ell_{i'}'^{(\tilde{t}+1)}| \cdot \sigma'(\langle \mathbf{w}_{y_i,r}^{(\tilde{t}+1)}, \mathbf{x}_i \rangle) \cdot \langle y_{i'} \mathbf{x}_{i'}, y_i \mathbf{x}_i \rangle$$
$$\geq \langle \mathbf{w}_{y_i,r}^{(\tilde{t}+1)}, \mathbf{x}_i \rangle + \frac{\eta}{nm} \cdot |\ell_i'^{(\tilde{t}+1)}| \cdot \|\mathbf{x}_i\|_2^2 - \frac{\eta}{nm} \sum_{i' \neq i} |\ell_{i'}'^{(\tilde{t}+1)}| \cdot |\langle \mathbf{x}_{i'}, \mathbf{x}_i \rangle|$$
$$\geq \langle \mathbf{w}_{y_i,r}^{(\tilde{t}+1)}, \mathbf{x}_i \rangle \geq 0,$$

where the second equality is by $\langle \mathbf{w}_{y_i,r}^{(\tilde{t}+1)}, \mathbf{x}_i \rangle \geq 0$; the first inequality is by triangle inequality; the second inequality is by $|\ell_{i'}'^{(\tilde{t}+1)}|/|\ell_i'^{(\tilde{t}+1)}| \leq C_2$ and the condition that $R_{\min}^2 \geq Cnp$, $C$ is a sufficiently large constant. This verifies the fourth bullet at time $t = \tilde{t} + 1$. $\qquad\square$

Notice that $\|\mathbf{W}^{(t)}\|_F = \Theta(\log t)$ and considering the fact that the difference between any two margins can be bounded by a constant, the difference between any two margins can be bounded by a constant, we can derive the following lemma, which demonstrates that the normalized margin of all the training data points will converge to the same value.

**Lemma G.2.** For both two-layer ReLU and leaky ReLU neural networks, gradient descent will asymptotically find a neural network such that all the training data points possess the same normalized margin, i.e.,

$$\lim_{t \to \infty} \left| y_i f\Big(\frac{\mathbf{W}^{(t)}}{\|\mathbf{W}^{(t)}\|_F}, \mathbf{x}_i\Big) - y_k f\Big(\frac{\mathbf{W}^{(t)}}{\|\mathbf{W}^{(t)}\|_F}, \mathbf{x}_k\Big) \right| = 0,$$

for any $i, k \in [n]$.

By Lemma G.1, we can establish the subsequent lemma regarding the logarithmic rate of increase in margin. This lemma will be beneficial in demonstrating the convergence rate of the training loss in subsequent proofs.

**Lemma G.3.** There exists time $T = \Theta(\eta^{-1} R_{\min}^{-2} nm)$ such that the following increasing rate of margin $y_i f(\mathbf{W}^{(t)}, \mathbf{x}_i)$ holds:

$$\left| y_i f(\mathbf{W}^{(t)}, \mathbf{x}_i) - \log t - \log(\eta \|\mathbf{x}_i\|_2^2 / nm) \right| \leq C_3,$$

$$C_4 \cdot \left( \frac{\eta \|\mathbf{x}_i\|_2^2}{nm} \cdot t \right)^{-1} \leq |\ell_i'^{(t)}| \leq C_5 \cdot \left( \frac{\eta \|\mathbf{x}_i\|_2^2}{nm} \cdot t \right)^{-1},$$

for any $i \in [n]$ and $t \geq T$, where $C_3, C_4, C_5$ are constants.

*Proof of Lemma G.3.* To prove this, we want to leverage Lemma H.1 and Lemma H.2. To achieve this, we need approximate $|\ell_i'^{(t)}|$ by $y_i f(\mathbf{W}^{(t)}, \mathbf{x}_i)$. We have

$$|\ell_i'^{(t)}| = \frac{1}{1 + \exp\left( y_i f(\mathbf{W}_{y_i}^{(t)}, \mathbf{x}_i) \right)} \leq \exp\left( -y_i f(\mathbf{W}_{y_i}^{(t)}, \mathbf{x}_i) \right), \tag{G.6}$$

and

$$|\ell_i'^{(t)}| = \frac{1}{1 + \exp\left( y_i f(\mathbf{W}_{y_i}^{(t)}, \mathbf{x}_i) \right)} \geq \frac{1}{1 + e^c} \cdot \exp\left( -y_i f(\mathbf{W}_{y_i}^{(t)}, \mathbf{x}_i) \right) \tag{G.7}$$

by $y_i f(\mathbf{W}_{y_i}^{(t)}, \mathbf{x}_i) \geq -c$. Plugging the upper and lower bounds of $|\ell_i'^{(t)}|$ into (G.4) and (G.5), we obtain

$$y_i f(\mathbf{W}^{(t+1)}, \mathbf{x}_i) - y_i f(\mathbf{W}^{(t)}, \mathbf{x}_i) \leq \frac{3\eta \|\mathbf{x}_i\|_2^2}{nm} \cdot \exp\left( -y_i f(\mathbf{W}_{y_i}^{(t)}, \mathbf{x}_i) \right), \tag{G.8}$$

$$y_i f(\mathbf{W}^{(t+1)}, \mathbf{x}_i) - y_i f(\mathbf{W}^{(t)}, \mathbf{x}_i) \geq \frac{\eta \|\mathbf{x}_i\|_2^2}{5(1 + e^c)nm} \cdot \exp\left( -y_i f(\mathbf{W}_{y_i}^{(t)}, \mathbf{x}_i) \right). \tag{G.9}$$

By taking $x_t = y_i f(\mathbf{W}^{(t)}, \mathbf{x}_i) - y_i f(\mathbf{W}^{(0)}, \mathbf{x}_i)$ and applying Lemma H.1 to (G.8), we can get

$$y_i f(\mathbf{W}^{(t)}, \mathbf{x}_i) \leq y_i f(\mathbf{W}^{(0)}, \mathbf{x}_i) + \log \left( 1 + \frac{3\eta \|\mathbf{x}_i\|_2^2}{e^{y_i f(\mathbf{W}^{(0)}, \mathbf{x}_i)} nm} \exp \left( \frac{3\eta \|\mathbf{x}_i\|_2^2}{e^{y_i f(\mathbf{W}^{(0)}, \mathbf{x}_i)} nm} \right) \cdot t \right).$$

As long as $t \geq \frac{e^{y_i f(\mathbf{W}^{(0)}, \mathbf{x}_i)} nm}{3\eta \|\mathbf{x}_i\|_2^2} \exp\left( -\frac{3\eta \|\mathbf{x}_i\|_2^2}{e^{y_i f(\mathbf{W}^{(0)}, \mathbf{x}_i)} nm} \right) = \Theta(\eta^{-1} R_{\min}^{-2} nm)$, we have

$$y_i f(\mathbf{W}^{(t)}, \mathbf{x}_i) \leq y_i f(\mathbf{W}^{(0)}, \mathbf{x}_i) + \log \left( \frac{6\eta \|\mathbf{x}_i\|_2^2}{e^{y_i f(\mathbf{W}^{(0)}, \mathbf{x}_i)} nm} \exp \left( \frac{3\eta \|\mathbf{x}_i\|_2^2}{e^{y_i f(\mathbf{W}^{(0)}, \mathbf{x}_i)} nm} \right) \cdot t \right)$$

$$= \log t + \log \left( \frac{\eta \|\mathbf{x}_i\|_2^2}{nm} \right) + \log 6 + \frac{3\eta \|\mathbf{x}_i\|_2^2}{e^{y_i f(\mathbf{W}^{(0)}, \mathbf{x}_i)} nm}$$

$$\leq \log t + \log \left( \frac{\eta \|\mathbf{x}_i\|_2^2}{nm} \right) + \log 6 + 1,$$

where the last inequality is by $y_i f(\mathbf{W}^{(0)}, \mathbf{x}_i) \leq 2\beta$ and $\eta \leq (C R_{\max}^2 / nm)^{-1}$, $C$ is a sufficiently large constant. By taking $x_t = y_i f(\mathbf{W}^{(t)}, \mathbf{x}_i) - y_i f(\mathbf{W}^{(0)}, \mathbf{x}_i)$ and applying Lemma H.2 to (G.9), we can get

$$y_i f(\mathbf{W}^{(t)}, \mathbf{x}_i) \geq y_i f(\mathbf{W}^{(0)}, \mathbf{x}_i) + \log \left( 1 + \frac{\eta \|\mathbf{x}_i\|_2^2}{5(1 + e^c) e^{y_i f(\mathbf{W}^{(0)}, \mathbf{x}_i)} nm} \cdot t \right)$$

$$\geq y_i f(\mathbf{W}^{(0)}, \mathbf{x}_i) + \log \left( \frac{\eta \|\mathbf{x}_i\|_2^2}{5(1 + e^c) e^{y_i f(\mathbf{W}^{(0)}, \mathbf{x}_i)} nm} \cdot t \right)$$

$$= \log t + \log \left( \frac{\eta \|\mathbf{x}_i\|_2^2}{nm} \right) - \log \left( 5(1 + e^c) \right).$$

Since

$$\frac{e^{y_i f(\mathbf{W}^{(0)}, \mathbf{x}_i)} nm}{3\eta\|\mathbf{x}_i\|_2^2} \exp\left(-\frac{3\eta\|\mathbf{x}_i\|_2^2}{e^{y_i f(\mathbf{W}^{(0)}, \mathbf{x}_i)} nm}\right) \le \frac{e^{y_i f(\mathbf{W}^{(0)}, \mathbf{x}_i)} nm}{3\eta\|\mathbf{x}_i\|_2^2} \le \frac{e^{2\beta}}{3}\eta^{-1} R_{\min}^{-2} nm,$$

Taking

$$T = \left\lceil \frac{e^{2\beta}}{3}\eta^{-1} R_{\min}^{-2} nm \right\rceil, \quad C_3 = \max\left\{\log 6 + 1, \log\left(5(1 + e^c)\right)\right\}$$

completes the proof of the first equation. By plugging the margin upper bound into (G.6), we can get

$$\begin{aligned}
|\ell_i'^{(t)}| &\le \exp\left(-y_i f(\mathbf{W}_{y_i}^{(t)}, \mathbf{x}_i)\right) \\
&\le \exp\left(-\log t - \log(\eta\|\mathbf{x}_i\|_2^2 / nm) + C_3\right) \\
&\le \exp(C_3) \cdot \left(\frac{\eta\|\mathbf{x}_i\|_2^2}{nm} \cdot t\right)^{-1}.
\end{aligned}$$

By plugging the margin lower bound into (G.7), we can get

$$\begin{aligned}
|\ell_i'^{(t)}| &\ge \frac{1}{1 + e^c} \cdot \exp\left(-y_i f(\mathbf{W}_{y_i}^{(t)}, \mathbf{x}_i)\right) \\
&\ge \frac{1}{1 + e^c} \cdot \exp\left(-\log t - \log(\eta\|\mathbf{x}_i\|_2^2 / nm) - C_3\right) \\
&\ge \frac{1}{e^{C_3}(1 + e^c)} \cdot \left(\frac{\eta\|\mathbf{x}_i\|_2^2}{nm} \cdot t\right)^{-1}.
\end{aligned}$$

Therefore, taking $C_4 = 1/e^{C_3}(1 + e^c)$ and $C_5 = \exp(C_3)$ completes the proof. $\quad\square$

Now we give the following lemma about the convergence rate of training loss.

**Lemma G.4.** For both two-layer ReLU and leaky ReLU networks defined in (G.1), we have the following convergence rate of training loss

$$L_S(\mathbf{W}^{(t)}) = \Theta(t^{-1}).$$

*Proof.* Having obtained a lower bound for the margin in Lemma G.3, we can now use it to derive an upper bound for the loss function as follows:

$$\begin{aligned}
L_S(\mathbf{W}^{(t)}) &= \frac{1}{n}\sum_{i=1}^{n} \ell(y_i f(\mathbf{W}^{(t)}, \mathbf{x}_i)) \\
&= \frac{1}{n}\sum_{i=1}^{n} \log\left(1 + \exp\left(-y_i f(\mathbf{W}^{(t)}, \mathbf{x}_i)\right)\right) \\
&\le \frac{1}{n}\sum_{i=1}^{n} \exp\left(-y_i f(\mathbf{W}^{(t)}, \mathbf{x}_i)\right) \\
&\le \frac{1}{n}\sum_{i=1}^{n} \exp\left(-\log t - \log(\eta\|\mathbf{x}_i\|_2^2 / nm) + C_3\right) \\
&= \frac{1}{n}\sum_{i=1}^{n} \exp(C_3) \cdot \left(\frac{\eta\|\mathbf{x}_i\|_2^2}{nm} \cdot t\right)^{-1} \\
&= O(t^{-1}).
\end{aligned}$$

where the first inequality is by $\log(1 + z) \le z$; the second inequality is by Lemma G.3.
Having obtained an upper bound for the margin in Lemma G.3, we can now use it to derive a lower bound for the loss function as follows:

$$L_S(\mathbf{W}^{(t)}) = \frac{1}{n}\sum_{i=1}^{n} \ell(y_i f(\mathbf{W}^{(t)}, \mathbf{x}_i))$$

$$= \frac{1}{n} \sum_{i=1}^{n} \log \left( 1 + \exp \left( - y_i f(\mathbf{W}^{(t)}, \mathbf{x}_i) \right) \right)$$

$$\geq \frac{1}{n} \sum_{i=1}^{n} \exp \left( - y_i f(\mathbf{W}^{(t)}, \mathbf{x}_i) \right) - \exp \left( - 2 y_i f(\mathbf{W}^{(t)}, \mathbf{x}_i) \right)$$

$$\geq \frac{1}{n} \sum_{i=1}^{n} \exp \left( - \log t - \log(\eta \|\mathbf{x}_i\|_2^2 / nm) - C_3 \right)$$

$$\quad - \exp \left( - 2(\log t + \log(\eta \|\mathbf{x}_i\|_2^2 / nm) - C_3) \right)$$

$$= \frac{1}{n} \sum_{i=1}^{n} \exp(-C_3) \cdot \left( \frac{\eta \|\mathbf{x}_i\|_2^2}{nm} \cdot t \right)^{-1} - \exp(2C_3) \cdot \left( \frac{\eta \|\mathbf{x}_i\|_2^2}{nm} \cdot t \right)^{-2}$$

$$= \Omega(t^{-1}).$$

where the first inequality is by $\log(1 + z) \geq z - z^2/2$ for $z \geq 0$; the second inequality is by Lemma G.3. This completes the proof. $\square$

In addition to the aforementioned lemmas, in the case of leaky ReLU, assuming convergence in direction, we can demonstrate that the directional limit corresponds to a Karush-Kuhn-Tucker (KKT) point of the max-margin problem. This result is presented in the following lemma.

**Lemma G.5.** For two-layer leaky ReLU network defined in (G.1), assume that $\mathbf{W}^{(t)}$ converges in direction, i.e. the limit of $\mathbf{W}^{(t)} / \|\mathbf{W}^{(t)}\|_F$ exists. Denote $\lim_{t \to \infty} \mathbf{W}^{(t)} / \|\mathbf{W}^{(t)}\|_F$ as $\bar{\mathbf{W}}$. There exists a scaling factor $\alpha > 0$ such that $\alpha \bar{\mathbf{W}}$ satisfies Karush-Kuhn-Tucker (KKT) conditions of the following max-margin problem:

$$\min_{\mathbf{W}} \frac{1}{2} \|\mathbf{W}\|_F^2, \qquad s.t. \qquad y_i f(\mathbf{W}, \mathbf{x}_i) \geq 1, \forall i \in [n]. \tag{G.10}$$

*Proof.* We need to prove that there exists $\lambda_1, \cdots, \lambda_n \geq 0$ such that for every $j \in \{\pm 1\}$ and $r \in [m]$ we have

$$\bar{\mathbf{w}}_{j,r} = \sum_{i=1}^{n} \lambda_i \nabla_{\mathbf{w}_{j,r}} \left( y_i f(\bar{\mathbf{W}}, \mathbf{x}_i) \right) = \sum_{i=1}^{n} \lambda_i y_i j \cdot \sigma'(\langle \bar{\mathbf{w}}_{j,r}, \mathbf{x}_i \rangle) \cdot \mathbf{x}_i. \tag{G.11}$$

By (5.1), we know that

$$\bar{\mathbf{w}}_{j,r} = \lim_{t \to \infty} \frac{\mathbf{w}_{j,r}^{(t)}}{\|\mathbf{W}^{(t)}\|_F}$$

$$= \lim_{t \to \infty} \frac{\mathbf{w}_{j,r}^{(0)}}{\|\mathbf{W}^{(t)}\|_F} + \sum_{i=1}^{n} \frac{\rho_{j,r,i}^{(t)}}{\|\mathbf{W}^{(t)}\|_F} \cdot \|\mathbf{x}_i\|_2^{-2} \cdot \mathbf{x}_i$$

$$= \lim_{t \to \infty} \sum_{i=1}^{n} \frac{\rho_{j,r,i}^{(t)}}{\|\mathbf{W}^{(t)}\|_F} \cdot \|\mathbf{x}_i\|_2^{-2} \cdot \mathbf{x}_i$$

$$= \sum_{i=1}^{n} \lim_{t \to \infty} \frac{\rho_{j,r,i}^{(t)}}{\|\mathbf{W}^{(t)}\|_F} \cdot \|\mathbf{x}_i\|_2^{-2} \cdot \mathbf{x}_i,$$

where the second equality is by $\|\mathbf{W}^{(t)}\|_F = \Theta(\log t)$ and the last equality is by the existence of $\lim_{t \to \infty} \mathbf{W}^{(t)} / \|\mathbf{W}^{(t)}\|_F$ as well as the uniqueness of data-correlated decomposition. By Lemma D.3 and $\|\mathbf{W}^{(t)}\|_F = \Theta(\log t)$, we can obtain that

$$\lim_{t \to \infty} \frac{\rho_{j,r,i}^{(t)}}{\|\mathbf{W}^{(t)}\|_F} = \lim_{t \to \infty} \frac{\rho_{j,r',i}^{(t)}}{\|\mathbf{W}^{(t)}\|_F}, \tag{G.12}$$

for any $j \in \{\pm 1\}$, $r, r' \in [m]$, $i \in [n]$, and

$$\lim_{t \to \infty} \frac{\rho_{j,r,i}^{(t)}}{\|\mathbf{W}^{(t)}\|_F} = -\gamma^{-1} \cdot \lim_{t \to \infty} \frac{\rho_{j,r',i'}^{(t)}}{\|\mathbf{W}^{(t)}\|_F}, \tag{G.13}$$

for any $j \in \{\pm 1\}$, $i, i' \in [n]$ with $j = y_i$, $j = -y_{i'}$ and $r, r' \in [m]$. Define

$$S_j = \{i \in [n] : y_i = j\}, \quad \lambda_i' := \lim_{t \to \infty} \frac{\rho_{y_i,r,i}^{(t)}}{\|\mathbf{W}^{(t)}\|_F}.$$

By (G.12), we know $\lambda_i'$ is well defined and $\lambda_i' \geq 0$. And by (G.13), we know that for any $r \in [m]$, $i \in [n]$,

$$\lim_{t \to \infty} \frac{\rho_{-y_i,r,i}^{(t)}}{\|\mathbf{W}^{(t)}\|_F} = -\gamma \lambda_i'.$$

Then, we have

$$\bar{\mathbf{w}}_{j,r} = \sum_{i=1}^{n} \lim_{t \to \infty} \frac{\rho_{j,r,i}^{(t)}}{\|\mathbf{W}^{(t)}\|_F} \cdot \|\mathbf{x}_i\|_2^{-2} \cdot \mathbf{x}_i$$

$$= \sum_{i \in S_j} \lim_{t \to \infty} \frac{\rho_{j,r,i}^{(t)}}{\|\mathbf{W}^{(t)}\|_F} \cdot \|\mathbf{x}_i\|_2^{-2} \cdot \mathbf{x}_i + \sum_{i \in S_{-j}} \lim_{t \to \infty} \frac{\rho_{j,r,i}^{(t)}}{\|\mathbf{W}^{(t)}\|_F} \cdot \|\mathbf{x}_i\|_2^{-2} \cdot \mathbf{x}_i$$

$$= \sum_{i \in S_j} \lambda_i' \cdot \|\mathbf{x}_i\|_2^{-2} \cdot \mathbf{x}_i - \sum_{i \in S_{-j}} \gamma \lambda_i' \cdot \|\mathbf{x}_i\|_2^{-2} \cdot \mathbf{x}_i.$$

By Lemma C.5, it holds for any $t \geq T_1$ that

$$\sigma'(\langle \mathbf{w}_{j,r}^{(t)}, \mathbf{x}_i \rangle) = 1, \qquad \forall j \in \{\pm 1\}, \, i \in S_j,$$
$$\sigma'(\langle \mathbf{w}_{j,r}^{(t)}, \mathbf{x}_i \rangle) = \gamma, \qquad \forall j \in \{\pm 1\}, \, i \in S_{-j}.$$

This leads to

$$\sigma'(\langle \bar{\mathbf{w}}_{j,r}, \mathbf{x}_i \rangle) = 1, \qquad \forall j \in \{\pm 1\}, \, i \in S_j,$$
$$\sigma'(\langle \bar{\mathbf{w}}_{j,r}, \mathbf{x}_i \rangle) = \gamma, \qquad \forall j \in \{\pm 1\}, \, i \in S_{-j}.$$

Thus, we can get

$$\bar{\mathbf{w}}_{j,r} = \sum_{i \in S_j} \lambda_i' \cdot \sigma'(\langle \bar{\mathbf{w}}_{j,r}, \mathbf{x}_i \rangle) \cdot \|\mathbf{x}_i\|_2^{-2} \cdot \mathbf{x}_i - \sum_{i \in S_{-j}} \lambda_i' \cdot \sigma'(\langle \bar{\mathbf{w}}_{j,r}, \mathbf{x}_i \rangle) \cdot \|\mathbf{x}_i\|_2^{-2} \cdot \mathbf{x}_i$$

$$= \sum_{i=1}^{n} \lambda_i' y_i j \cdot \sigma'(\langle \bar{\mathbf{w}}_{j,r}, \mathbf{x}_i \rangle) \cdot \|\mathbf{x}_i\|_2^{-2} \cdot \mathbf{x}_i.$$

Taking $\lambda_i = \lambda_i' \|\mathbf{x}_i\|_2^{-2}$ completes the proof of (G.11). On the other hand, by Lemma G.2 and the assumption of the existence of $\mathbf{W}^{(t)} / \|\mathbf{W}^{(t)}\|_F$, we can get

$$y_i f(\bar{\mathbf{W}}, \mathbf{x}_i) = y_k f(\bar{\mathbf{W}}, \mathbf{x}_k),$$

for any $i, k \in [n]$. Taking $\alpha = 1/y_i f(\bar{\mathbf{W}}, \mathbf{x}_i)$, we have

$$y_i f(\alpha \bar{\mathbf{W}}, \mathbf{x}_i) = 1,$$

for any $i \in [n]$, which completes the proof. $\qquad \square$

# H  Auxiliary Lemmas

**Lemma H.1.** Let $\{x_t\}_{t=0}^{\infty}$ be an non-negative sequence satisfying the following inequality:

$$x_{t+1} - x_t \leq C \cdot e^{-x_t}, \forall t \geq 0$$

then we have
$$x_t \leq \log(e^{x_0} + Ce^C \cdot t).$$

*Proof of Lemma H.1.* Given the inequality $x_{t+1} - x_t \leq C \cdot e^{-x_t}$ for all $t \geq 0$, we want to prove that $x_T \leq \log(e^{x_0} + Ce^C \cdot T)$ for $T \geq 0$. We start by manipulating the inequality as follows:

$$x_{t+1} - x_t \leq C \cdot e^{-x_t}$$
$$\implies \quad x_{t+1} - x_t \leq C \cdot e^{-x_{t+1} + C} \qquad \text{(using } x_{t+1} \text{ instead of } x_t\text{)}$$
$$\implies \quad e^{x_{t+1}}(x_{t+1} - x_t) \leq Ce^C \qquad \text{(multiplying both sides by } e^{x_{t+1}}\text{).}$$

Summing the inequality from $t = 0$ to $t = T - 1$, we get:

$$\sum_{t=0}^{T-1} e^{x_{t+1}}(x_{t+1} - x_t) \leq Ce^C \cdot T.$$

Since $e^x$ is a monotone increasing function, we can approximate the above sum with an integral:

$$\int_{x_0}^{x_T} e^x dx \leq Ce^C \cdot T.$$

Evaluating the integral, we get:

$$e^{x_T} - e^{x_0} \leq Ce^C \cdot T.$$

Rearranging the inequality, we get:

$$e^{x_T} \leq e^{x_0} + Ce^C \cdot T.$$

Taking the natural logarithm of both sides, we get:

$$x_T \leq \log(e^{x_0} + Ce^C \cdot T).$$

Therefore, we have shown that $x_T \leq \log(e^{x_0} + Ce^C \cdot T)$, as required. $\qquad \square$

**Lemma H.2.** Let $\{x_t\}_{t=0}^{\infty}$ be an sequence satisfying the following inequality:

$$x_{t+1} - x_t \geq C \cdot e^{-x_t}, \forall t \geq 0$$

then we have

$$x_t \geq \log(e^{x_0} + C \cdot t).$$

*Proof of Lemma H.2.* Given the inequality $x_{t+1} - x_t \geq C \cdot e^{-x_t}$ for all $t \geq 0$, we want to prove that $x_T \geq \log(e^{x_0} + C \cdot T)$ for $T \geq 0$. We start by manipulating the inequality as follows:

$$x_{t+1} - x_t \geq C \cdot e^{-x_t}$$
$$\implies \quad e^{x_t}(x_{t+1} - x_t) \geq C \qquad \text{(multiplying both sides by } e^{x_t}\text{).}$$

Summing the inequality from $t = 0$ to $t = T - 1$, we get:

$$\sum_{t=0}^{T-1} e^{x_t}(x_{t+1} - x_t) \geq C \cdot T.$$

Since $e^x$ is a monotone increasing function, we can approximate the above sum with an integral:

$$\int_{x_0}^{x_T} e^x dx \geq C \cdot T.$$

Evaluating the integral, we get:

$$e^{x_T} - e^{x_0} \geq C \cdot T.$$

Rearranging the inequality, we get:

$$e^{x_T} \geq e^{x_0} + C \cdot T.$$

Taking the natural logarithm of both sides, we get:

$$x_T \geq \log(e^{x_0} + C \cdot T).$$

Therefore, we have shown that $x_T \geq \log(e^{x_0} + C \cdot T)$, as required. $\square$

**Lemma H.3** (Theorem 4.4.5 in Vershynin (2018)). Let $\mathbf{A}$ be an $m \times n$ random matrix whose entries $a_{ij}$ are independent, mean zero, sub-gaussian random variables. Then for any $t > 0$ we have

$$\|\mathbf{A}\|_2 \leq CK(\sqrt{m} + \sqrt{n} + t)$$

with probability at least $1 - 2\exp(-t^2)$. Here $K = \max_{i,j} \|a_{ij}\|_{\phi_2}$ where $\|\cdot\|_{\phi_2}$ is the sub-gaussian norm.

**Lemma H.4.** For $t \geq s > 0$, we have

$$\frac{\log(1 + at)}{\log(1 + bt)} \geq \frac{\log(1 + as)}{\log(1 + bs)},$$

if $b > a > 0$.

*Proof of Lemma H.4.* Let $f(t) = \log(1 + at) / \log(1 + bt)$, and we want to prove that $f'(t) > 0$ for all $t > 0$. To find the derivative of $f(t)$, we use the quotient rule:

$$
\begin{aligned}
f'(t) &= \frac{(\log(1 + bt))\frac{d}{dt}(\log(1 + at)) - (\log(1 + at))\frac{d}{dt}(\log(1 + bt))}{(\log(1 + bt))^2} \\
&= \frac{(\log(1 + bt))\frac{a}{1+at} - (\log(1 + at))\frac{b}{1+bt}}{(\log(1 + bt))^2} \\
&= \frac{a(1 + bt)\log(1 + bt) - b(1 + at)\log(1 + at)}{(1 + at)(1 + bt)(\log(1 + bt))^2}.
\end{aligned}
$$

Next, we define the function $g(t) = \left(\frac{1}{b} + t\right)\log(1 + bt) - \left(\frac{1}{a} + t\right)\log(1 + at)$, and we aim to show that $g'(t) > 0$ for all $t > 0$. We start by computing the derivative of $g(t)$:

$$g'(t) = \log(1 + bt) - \log(1 + at).$$

Since $b > a$ and $t > 0$, we have $1 + bt > 1 + at$, which implies that $\log(1 + bt) > \log(1 + at)$. Therefore, we have $g'(t) > 0$ for all $t > 0$. Note that $g(0) = 0$, we then have $g(t) > 0$ for all $t > 0$. Therefore, we have $a(1 + bt)\log(1 + bt) - b(1 + at)\log(1 + at) > 0$ for all $t > 0$, which in turn implies that $f'(t) > 0$ for all $t > 0$. Thus, we have shown that $f(t)$ is increasing for $t > 0$ and hence $f(t) > f(s)$, which completes the proof. $\square$

**Lemma H.5.** Let $g(z) = \ell'(z) = -1/(1 + \exp(z))$, then we have that

$$\frac{g(z_2)}{g(z_1)} \leq 2\big(1 + \exp(z_1 - z_2)\big), \forall z_1, z_2 \in \mathbb{R},$$

and

$$\frac{g(z_2)}{g(z_1)} \geq \frac{1}{4}\exp(z_1 - z_2), \forall z_1 \in \mathbb{R}, z_2 \geq -1.$$

*Proof of Lemma H.5.* We first prove the first inequality. For if $z_1 \leq 0$, we have

$$\frac{g(z_2)}{g(z_1)} = \frac{1 + \exp(z_1)}{1 + \exp(z_2)} \leq \frac{2}{1 + \exp(z_2)} \leq 2.$$

For if $z_1 > 0$, we have

$$\frac{g(z_2)}{g(z_1)} = \frac{1 + \exp(z_1)}{1 + \exp(z_2)} \leq \frac{2\exp(z_1)}{\exp(z_2)} = 2\exp(z_1 - z_2).$$

Thus, for any $z_1 \in \mathbb{R}$, we have

$$\frac{g(z_2)}{g(z_1)} \leq 2 + 2\exp(z_1 - z_2).$$

Now we prove the second inequality. We have

$$\frac{g(z_2)}{g(z_1)} = \frac{1 + \exp(z_1)}{1 + \exp(z_2)} = \frac{\exp(-z_2) + \exp(z_1 - z_2)}{\exp(-z_2) + 1} \geq \frac{\exp(z_1 - z_2)}{\exp(1) + 1} \geq \frac{1}{4}\exp(z_1 - z_2),$$

which completes the proof. $\qquad\square$

