# OpenReview forum: "Implicit Bias of Gradient Descent for Two-layer ReLU and Leaky ReLU Networks on Nearly-orthogonal Data"
_NeurIPS.cc/2023/Conference — NeurIPS 2023 poster_

### Official Review · Reviewer_6oj7 · 2023-06-11

**Soundness:** 4 excellent
**Presentation:** 3 good
**Contribution:** 3 good
**Rating:** 7
**Confidence:** 4

**Summary:**

The paper studies convergence and implicit bias in two-layer (leaky) ReLU networks trained with gradient descent. They focus on the case where the initialization scale and the step size of GD are sufficiently small, and the training data are nearly orthogonal. They show two main results:
-  For leaky ReLU networks, they prove convergence to stable-rank 1 of the weight matrices (they consider the matrix of the positive neurons and the matrix of the negative neurons separately), and also give a convergence rate analysis;
-  For ReLU networks they bound the stable ranks by a constant, which in some special cases can be roughly 2. They analyze the convergence rate also here.


**Strengths:**

I think that it is a nice paper. The near-orthgonality assumption is quite strong, but I believe that it is a natural assumption, and that understanding the behavior of GD in this setting is helpful. For this reason, this assumption appears (either explicitly or implicitly) in several works in recent years.

This work extends the results from Frei et al. 2022b in several ways:
- Frei et al. 2022b showed convergence to rank 1 of the matrices when using gradient flow and constant stable rank when using GD, and here it is extended to stable rank 1 using GD;
- Frei et al. 2022b showed convergence to a constant stable rank when using GD with smooth leaky ReLU activation, and here it is extendedto leaky ReLU and to ReLU;
- The convergence rate here is tighter than Frei et al. 2022b.

I think that the question considered in the paper is interesting and the contribution is significant and non-trivial. Also, the paper is well-written.

**Weaknesses:**

I don’t see major weaknesses. Below I discuss some comments/questions that I have.

- In the introduction the authors argue that “This finding suggests that ReLU networks possess superior learning ability compared to leaky ReLU networks”. I don’t see how this claim follows from the results. This sentence appears after mentioning the example where the stable rank becomes approximately 2 in ReLU networks, but this occurs also in leaky ReLU networks for this example.
- Related to the above comment: Relating low stable rank to learning ability is not a trivial argument. While the exact rank can be used for bounding the complexity of the hypothesis class, this is not true in general for the stable rank. So I don’t see a direct argument that gives learning guarantees by only using the stable rank. (Still, I think that the stable rank is a natural notion and hence studying it is of interest).
- The experiments on MNIST are conceptually similar to the experiments on CIFAR from Frei et al. 2022b. Please explain what is the contribution of this experiment compared to this prior work.
- Typos:
    - In Lemma 5.4 (line 215): should be $i,i’ \in [n]$ instead of $r,r’ \in [m]$.
    - Line 286: “RelU”
    - Line 759 (in Lemma F.2):
        - In the first property should be $\rho_{j,r,i}$
        - The second property should be without the \bar{}
        - All conditions should consiter $w^{(0)}$ rather than $w^{(t)}$
    - Last equation in page 42: should be $0_{m \times(d-n)}$. Also in the equation before that.


**Questions:**

See the “weaknesses” section.

**Limitations:**

The authors discussed the limitations.

---

> ### Author Rebuttal · Authors · 2023-08-10
>
> Thank you for your positive feedback and thoughtful comments! Below are our answers to your comments and questions.
>
> ----
>
> **Q1**. In the introduction, the authors argue that “This finding suggests that ReLU networks possess superior learning ability compared to leaky ReLU networks”. This sentence appears after mentioning the example where the stable rank becomes approximately 2 in ReLU networks, but this occurs also in leaky ReLU networks for this example.
>
> **A1**. In the context of leaky ReLU networks, we prove that the stable rank of the weight matrix converges to 1 (rather than 2) when the data are nearly orthogonal. For ReLU networks, we provide an example concerning fully orthogonal data and prove that the stable rank of the weight matrix trained using such data converges to a value approximately equal to 2. Since the stable rank achieved by ReLU networks is larger than the stable rank achieved by leaky ReLU networks,  this suggests that ReLU networks possess superior learning ability than leaky ReLU.
>
> ----
>
> **Q2**.  Relating low stable rank to learning ability is not a trivial argument. While the exact rank can be used for bounding the complexity of the hypothesis class, this is not true in general for the stable rank.
>
> **A2**. Thank you for raising this point. We agree that, in general, using stable rank to characterize the complexity of the hypothesis class needs a more careful argument. Yet in our results, the stable rank of leaky ReLU network is asymptotically 1, which means the exact rank equals to 1, and the stable rank of ReLU network is approximately 2 for learning completely orthogonal data, which means the rank is strictly greater than 1. So our argument about the learning ability of leaky ReLU vs ReLU still holds through the lens of exact rank.
>
> ----
>
> **Q3**. Explain what is the contribution of this experiment compared to Frei et al. 2022b.
>
> **A3**. While our experiments on MNIST bear conceptual similarities to those of Frei et al. 2022b on CIFAR10, they offer additional insights:
>
> - We conducted our validation on the MNIST dataset, in addition to Frei et al.'s choice of CIFAR10. This allowed us to test the robustness of our findings across different datasets.
>
> - We presented visualizations of stable rank changes for both leaky ReLU and ReLU under various weight variances. In contrast, Frei et al. 2022b only visualized ReLU.
>
> - We provide additional experiments in Appendix I (Figures 4 and 5) to show how the stable rank of networks for both ReLU and leaky ReLU is influenced by network width. Our findings indicate that, for networks of adequate width, further increase in network width does not correlate with stable rank increments. Therefore, the stable rank emerges as an inherent characteristic of a specific learning problem, which is invariant in the network width.
>
> In essence, our work not only supplements but also broadens the experimental scope initiated by Frei et al. 2022b, delivering a more comprehensive understanding of stable ranks across various settings.
>
> ----
>
> **Q4**. Typos.
>
> **A4**. Thank you for pointing them out! We will fix them in the revision.

---

> > ### Comment · Reviewer_6oj7 · 2023-08-12
> >
> > Thanks for the response. I will stick with my high score.
> > Regarding Q1, I still don't understand this issue. Why do you claim that a higher rank is related to "superior learning ability"? It seems to me that a lower rank might be better in terms of generalization because it implies lower complexity of the hypothesis class.

---

> > > ### Author Response · Authors · 2023-08-12
> > >
> > > Thank you for your follow-up question. Here “superior learning ability” means “superior expressive power.”  A nearly stable rank 1 weight matrix implies that all the weights have nearly the same directions, indicating the low expressive power of the learned neural networks
> > >
> > > Since we don’t have any assumptions about the data distribution, it's hard to consider generalization ability. However, we acknowledge that superior expressive power could potentially lead to a poorer generalization. To clarify, we will replace the term "superior learning ability" with "superior expressive power" in the revision.

---

> > > > ### Comment · Reviewer_6oj7 · 2023-08-13
> > > >
> > > > I am also not sure that it is related to superior expressive power. I think that it just means that the implicit bias in each architecture is different.

---

> > > > > ### Author Response · Authors · 2023-08-13
> > > > >
> > > > > Thank you for your suggestion, which makes our claim more accurate. We will modify our claim to “This finding suggests that the implicit bias of ReLU networks is different from that of leaky ReLU networks.”

---

### Official Review · Reviewer_BcDb · 2023-06-15

**Soundness:** 4 excellent
**Presentation:** 3 good
**Contribution:** 3 good
**Rating:** 7
**Confidence:** 3

**Summary:**

The paper investigates implicit bias, especially in terms of the stable rank of the weight matrix (which is the square of the ratio between the Frobenius and spectral norms), of gradient descent on the logistic loss for binary classification tasks and two-layer neural networks, where the first layer is trained and the second layer is fixed, the activation function is either ReLU or Leaky ReLU, and the training data are nearly orthogonal.  The main results are that, provided the learning rate and the initialisation scale are sufficiently small, as the number of training steps tends to infinity, the stable rank tends to 1 in the Leaky ReLU case, and is bounded by a constant in the ReLU case.  The authors also prove upper bounds on the convergence rate of the empirical loss, and tight bounds on the convergence rate of the norms of individual neurons in the Leaky ReLU case.  The theoretical results are supplemented by experiments on synthetic data generated by adding Gaussian noise, and on MNIST.

**Strengths:**

The paper tackles one of the greatest current challenges in the theory of learning, namely understanding the implicit bias of gradient-based algorithms and its consequences for generalisation.  In contrast to most related works, it considers gradient descent rather than gradient flow, and is thus more readily applicable to practical settings.  The activation functions that it focuses on, namely ReLU and Leaky ReLU, are among the most popular.  The theoretical results are proved in detail in the extensive appendix, and sufficient details as well as the code for the experiments are provided.

**Weaknesses:**

The paper is a follow-up to "Implicit Bias in Leaky ReLU Networks Trained on High-Dimensional Data" by Frei, Vardi, Bartlett, Srebro, and Hu (ICLR 2023), and shares most of the assumptions with the gradient descent part of that work, with the important difference that in the present paper the activation functions are ReLU and Leaky ReLU rather than a smooth variant of Leaky ReLU as in that work.  It is not very clear to me what difficulties exactly needed to be overcome to step up from that previous work to this paper, or in other words to what extent and in what ways the present paper is not a straightforward increment.  Another aspect that is similarly unclear to me is how essentially different are the ReLU and Leaky ReLU cases.  The theorems and proofs for them are presented in the main and in the appendix mostly separately, yet seem to follow similar patterns, so it would be good to clarify where the main underlying differences are.  A minor comment is that there is a typo saying that the signal-noise decomposition technique is introduced in the Preliminaries.

**Questions:**

What can you say about what happens if the second layer is trained as well, or if the network is deeper?

**Limitations:**

Training only the first layer significantly simplifies the dynamics.

---

> ### Author Rebuttal · Authors · 2023-08-10
>
> We appreciate your valuable comments and feedback! We address your comments and questions below.
>
> ----
>
> **Q1**. It is not very clear to me what difficulties exactly needed to be overcome to step up from that previous work to this paper.
>
> **A1**. The technique employed in this paper differs significantly from that used in [1]. The argument presented in [1] (similar to the approach in [2]) heavily relies on the smoothness assumption of both the activation function and the neural network. They utilize this assumption to approximate $f(W^{(t+1)},x)=f(W^{(t)}-\eta\nabla L(W^{(t)}),x)$ by $f(W^{(t)},x)-\eta\langle\nabla f(W^{(t)},x),\nabla L(W^{(t)})\rangle$ to establish the balance of loss ratio for different training examples, which lays the foundation for proving their Theorem 4.2.
>
> In sharp contrast, our paper adopts a significantly different technique, which transforms the analysis of $W^{(t)}$ into the dynamic and increasing pattern of decomposition coefficients $\rho_{j,r,i}^{(t)}$. We then use these coefficients to approximate $f(W^{(t)},x)$ and prove our theory. This approach allows us to get rid of the smoothness assumption, making it suitable for analyzing Leaky ReLU/ReLU (rather than their smoothed version) networks.
>
> [1] Frei et al. "Implicit bias in leaky relu networks trained on high-dimensional data." arXiv preprint arXiv:2210.07082 (2022).
>
> [2] Frei et al. "Benign overfitting without linearity: Neural network classifiers trained by gradient descent for noisy linear data." Conference on Learning Theory. PMLR, 2022.
>
> ----
>
> **Q2**. How essentially different are the ReLU and Leaky ReLU cases.
>
> **A2**. For the case of leaky ReLU activation functions, we can accurately describe the activation behavior once a certain threshold of training time has been reached. To elaborate further, this behavior entails that beyond that specific time point, all neurons of class $y_i$ can be activated by the input $x_i$, while none of the neurons of class $-y_i$ can be activated by $x_i$. This observed pattern forms a pivotal basis for establishing the stable rank of the weight matrix. This is due to the fact that, given this pattern, the increments in $\rho_{j,r,i}^{(t)}$ are consistent across different values of $r$, assuming a fixed pairing of $j$ and $i$.
>
> However, for the ReLU case, such a property does not hold. It is rather challenging to comprehensively characterize the activation pattern, as there might be instances where neurons of class $y_i$ are never activated by input $x_i$. Additionally, the activation pattern of neurons belonging to class $-y_i$ with respect to input $x_i$ could be consistently activated, or even flip infinitely throughout the training process. Consequently, the increments in $\rho_{j,r,i}^{(t)}$ might exhibit inconsistencies across different $r$ values, assuming a fixed pairing of $j$ and $i$. This leads us to only be able to establish a constant upper limit for the stable rank of the weight matrix. As a result, the arguments for both leaky ReLU and ReLU, presented in both the main paper and the appendix, are treated separately.
>
> ----
>
> **Q3**. A minor comment is that there is a typo saying that the signal-noise decomposition technique is introduced in the Preliminaries.
>
> **A3**. We’re sorry for the typo. We will remove the “ signal-noise decomposition technique” here.
>
> ----
>
> **Q4**. What can you say about what happens if the second layer is trained as well, or if the network is deeper?
>
> **A4**. Our proof begins with decomposing the weight by the direction of the data input, and demonstrating the coefficients satisfy an ‘automatic balanced’ property. When working with deep neural networks, or two-layer networks with the second layer being trained, we need to first decompose the weight by the direction of the layer inputs. For example, we can decompose the weight of the first layer by the direction of the data inputs, but for subsequent layers, like the second layer, we must decompose the weights with respect to the direction of the second layer inputs (i.e., the data representation after the first layer). We can then prove that the coefficients remain auto-balanced. However, a significant challenge arises when the direction of the data representation changes during training. To address this challenge, we need to prove that the direction of the representation will converge, which requires more involved analysis. We will explore this direction in the future.

---

> > ### Comment · Reviewer_BcDb · 2023-08-10
> >
> > Thank you for these helpful answers.  In their light, and in that of the rest of the rebuttal, I shall raise my score to 7.

---

> > > ### Author Response · Authors · 2023-08-10
> > > **Thank you!**
> > >
> > > Thank you for raising the score and for your insightful comments!

---

### Official Review · Reviewer_4Kvz · 2023-07-04

**Soundness:** 3 good
**Presentation:** 3 good
**Contribution:** 3 good
**Rating:** 6
**Confidence:** 4

**Summary:**

This paper studies training two-layer leaky-ReLU/ReLU networks with gradient descent. The authors show that for approximately orthogonal data, GD converges and the weight matrices in the leaky-ReLU network have a stable rank of one asymptotically under GD, while those in the ReLU network can have a stable rank larger than one.

**Strengths:**

1. The convergence analysis of GD on two-layer neural networks seems novel
2. The characterization of the stable rank of the weight matrices is interesting

**Weaknesses:**

1. There is little discussion on the upper bound $c$ in Theorem 4.3. How large it can be? How does it depend on the training data?
2. For leaky-ReLU, the theorem only says it has a stable-rank one, meaning $W_j$ is approximately rank-one, is there a characterization of the left and right dominant singular vectors?

**Questions:**

See "Weakness"

**Limitations:**

yes

---

> ### Author Rebuttal · Authors · 2023-08-10
>
> Thank you for your insightful feedback! Here's our response to your comments and questions.
>
> ----
>
> **Q1**. There is little discussion on the upper bound c in Theorem 4.3. How large it can be? How does it depend on the training data?
>
> **A1**. The upper bound c in Theorem 4.3 can be considered an absolute constant. The relationship between c and the training data is determined solely by two quantities: $np/R_{min}^{2}$ and $R=R_{max}/R_{min}$. However, we make the assumption that both of these quantities can be bounded above by a constant. As a result, the upper bound c can also be treated as a constant. And the exact form of upper bound $c$ is given by $CR^6(1-R_{min}^{-2}np)$ where $C$ represents an absolute constant unrelated to the training data. The terms $R$, $R_{max}$, $R_{min}$ and $p$ are defined in lines 108, 109, 110.
>
> ----
>
> **Q2**. Is there a characterization of the left and right dominant singular vectors?
>
> **A2**. The convergence direction of network parameters as singular vectors of a tensor has been examined in linear neural networks trained via gradient flow, as described in [1]. However, for non-linear and non-smooth neural networks trained using gradient descent, the directional convergence of the weight matrix remains an unresolved issue. As we have not yet demonstrated the convergence of the weight matrix, we are currently unable to characterize the dominant left and right singular vectors. We will comment on this point as a limitation of our method in the final version.
>
> [1] Yun et al. "A unifying view on implicit bias in training linear neural networks." arXiv preprint arXiv:2010.02501 (2020).

---

### Official Review · Reviewer_c8EH · 2023-07-07

**Soundness:** 3 good
**Presentation:** 3 good
**Contribution:** 3 good
**Rating:** 6
**Confidence:** 3

**Summary:**

This study analyzes two-layer ReLU and leaky ReLU networks trained by gradient descent in a classification problem. It is shown that when training data is mutually nearly-orthogonal, by taking an initialization scale and a learning rate sufficiently small, a leaky ReLU network converges to a parameter with stable rank 1. They also show that a ReLU network converges to a parameter with a bounded stable rank, and it will equal 2 when the input is completely orthogonal. A similar analysis has already been provided by Frei et al. 2022 (with gradient flow and Leaky ReLU), this work takes another approach apart from the KKT condition, and a possibly faster convergence rate is obtained.

**Strengths:**

- Frei et al. (2022) have already shown a certain result on this topic, but it requires several non-practical modifications of the algorithm
(such as considering gradient flow or smoothed leaky ReLU).
This study overcomes this shortcoming with a novel approach and obtains further results (extension for the ReLU activation and gradient descent without smoothing), which has an important novelty.


**Weaknesses:**

- The near-orthogonality assumption $R_{\min}^2\ge CR^2\gamma^{-4}np$ seems to be strong (while Frei et al. (2022) also impose the same condition, but I realize that they assume high-dimensional settings).

- It will be more interesting if the authors can characterize the direction to which the weight matrix converges.

- The authors refer to the works on benign overfitting in the related work section, but the connection of this study to them needs to be made clearer. Does this study focus on high-dimensional settings? Or is the double descent curve provably obtained by this study?

- Some terms are used without definition, such as $W_{y_i}^{(t)}$ in (5.8) and $w_{y_i,r}^{(0}$ in Lemma 5.5, and so on.

**Questions:**

- The authors conduct numerical experiments and observe the stable rank of trained networks, but I'm wondering about it precisely reflects the theoretical result. That is, whether the stable rank of the leaky ReLU network with large $\gamma$ converges to 1 or not, and the stable rank of the ReLU network could converge to 2 by using fully-orthogonal training data.

---

> ### Author Rebuttal · Authors · 2023-08-10
>
> Thank you for your positive feedback and valuable comments! We address your comments and questions as follows.
>
> ----
>
> **Q1**. The near-orthogonality assumption seems to be strong.
>
> **A1**. We have based our work on the assumption of nearly-orthogonal data following Vardi et al., (2022) and Frei et al., (2022). This assumption becomes milder in the high-dimensional setting. We will add more discussion about that in the revision. Although we are not aware of any weaker assumption to be used here, we will pursue it as an important future direction.
>
> ----
>
> **Q2**. It will be more interesting if the authors can characterize the direction to which the weight matrix converges.
>
> **A2**. Thank you for your suggestion. There exist several challenges that need to be addressed to establish the directional convergence of the weight matrix. Firstly, in the context of leaky ReLU, even though we can demonstrate that neurons within the same class align with a common direction, we haven't yet devised a method to prove the convergence of $\rho_{j,r,i}^{(t)}/\\|W_j\\|_{F}^{2}$. In the case of ReLU, prior to delving into directional convergence, we need to comprehend the activation pattern of all neurons. Regrettably, in the ReLU scenario, we can only characterize a portion of the activation pattern. As the training progresses, unlike the leaky ReLU case, instances might arise where neurons belonging to class $y_i$ remain deactivated by $\mathbf{x}_i$, and neurons of class $-y_i$ are consistently activated by $\mathbf{x}_i$ or even alternate their activation states infinitely. Consequently, establishing directional convergence proves to be intricate at present. We plan to delve further into these intricacies in the future.
>
> ----
>
> **Q3**. Does this study focus on high-dimensional settings? Or is the double descent curve probably obtained by this study?
>
> **A3**. Yes, both our paper and the papers studying the benign overfitting phenomenon focus on high-dimensional settings. We both need to prove that the training loss will converge to zero for the non-convex loss. However, implicit bias is a property of the learning algorithm on the training data, while benign overfitting/double descent is a property of the learning algorithm on both training and test data. Since we don’t make any assumption on the test data distribution, we are not able to prove double descent or benign overfitting in our study.
>
> ----
>
> **Q4**. Some terms are used without definition.
>
> **A4**. In line 95, $W_j$ is defined as the collection of model weights associated with $F_j (j\in{\pm 1})$. Additionally, $W_{y_i}$ signifies the collection of model weights belonging to class $y_i$, whereas $W_{y_i}^{(t)}$ pertains to the collection of model weights of class $y_i$ obtained via gradient descent at iteration $t$. Proceeding to line 96, the term $w_{j,r}$ is clarified as the weight vector corresponding to the r-th neuron in $W_j$, and $w_{y_i,r}^{(t)}$ denotes the weight vector corresponding to the r-th neuron within $W_{y_i}$ obtained by gradient descent at iteration t.
>
> ----
>
> **Q5**. Wonder whether experiments reflect the theoretical result. That is, whether the stable rank of the leaky ReLU network with large $\gamma$ converges to $1$ or not, and the stable rank of the ReLU network could converge to $2$ by using fully-orthogonal training data.
>
> **A5**. Thank you for your suggestion! We have conducted additional experiments to validate our theoretical results for long epochs (1000). The stable rank for the leaky ReLU network with large slopes will converge to $1$. In comparison, the stable rank for the ReLU network will not converge to $1$.
>
> We also conduct the experiments on the fully-orthogonal training data and find that the stable rank for the large-width ReLU network will converge to $2$ instead of $1$.  The detailed proof for the behavior of a large-width ReLU network on the fully-orthogonal training data can be found in Appendix Lemma F.3 from Line 796 to 828.
>
> These additional experiments can be found in the uploaded PDF.

---

> > ### Comment · Reviewer_c8EH · 2023-08-18
> >
> > Thanks for the author's reply. My concerns are adequately addressed.

---

### Author Rebuttal · Authors · 2023-08-10

We want to thank all the reviewers for their positive and valuable feedback. In the uploaded pdf, we include additional empirical results to address reviewer c8EH’s concern. We have conducted experiments for long epochs (1000). We find that the stable rank for the leaky ReLU network with large slopes will converge to $1$. In comparison, the stable rank for the ReLU network will not converge to $1$.  We also conduct the experiments on the fully-orthogonal training data and find that the stable rank for the large-width ReLU network will converge to $2$ instead of $1$. The additional empirical findings strongly support our main Theorems 4.1 and 4.3.

---

### Decision · Program_Chairs · 2023-09-21

**Decision:**

Accept (poster)

**Comment:**

This paper studies the low-rank implicit bias of GD in two-layer ReLU and leaky ReLU networks in near-orthogonal data settings, strengthening a previous paper by Frei et al. The reviewers unanimously agree that this is a good paper, and therefore I recommend accepting it.

I suggest changing to a less broad title, such as including "low-rank" and "near-orthogonal."